# Propellanes as Rigid Scaffolds for the Stereodefined Attachment of σ-Pharmacophoric Structural Elements to Achieve σ Affinity

**DOI:** 10.3390/ijms22115685

**Published:** 2021-05-26

**Authors:** Héctor Torres-Gómez, Constantin Daniliuc, Dirk Schepmann, Erik Laurini, Sabrina Pricl, Bernhard Wünsch

**Affiliations:** 1Department of Chemistry and Pharamcy, Institut für Pharmazeutische und Medizinische Chemie, Westfälische Wilhelms-Universität Münster, Corrensstraße 48, D-48149 Münster, Germany; tghm@uaem.mx (H.T.-G.); dirk.schepmann@uni-muenster.de (D.S.); 2Department of Chemistry and Pharamcy, Organisch-Chemisches Institut, Westfälische Wilhelms-Universität Münster, Corrensstraße 40, D-48149 Münster, Germany; constantin.daniliuc@uni-muenster.de; 3Molecular Biology and Nanotechnology Laboratory (MolBNL@UniTS), Department of Engineering and Architecture, University of Trieste, 34127 Trieste, Italy; SABRINA.PRICL@dia.units.it (S.P.); ERIK.LAURINI@dia.units.it (E.L.); 4Department of General Biophysics, Faculty of Biology and Environmental Protection, University of Lodz, 90-236 Lodz, Poland; 5GRK 2515, Chemical Biology of Ion Channels (Chembion), Westfälische Wilhelms-Universität Münster, D-48149 Münster, Germany

**Keywords:** σ receptors, rigidity, propellanes, azapropellanes, stereochemistry, X-ray crystal structures, molecular dynamics, selectivity, molecular interactions

## Abstract

Following the concept of conformationally restriction of ligands to achieve high receptor affinity, we exploited the propellane system as rigid scaffold allowing the stereodefined attachment of various substituents. Three types of ligands were designed, synthesized and pharmacologically evaluated as σ_1_ receptor ligands. Propellanes with (1) a 2-methoxy-5-methylphenylcarbamate group at the “left” five-membered ring and various amino groups on the “right” side; (2) benzylamino or analogous amino moieties on the “right” side and various substituents at the left five-membered ring and (3) various urea derivatives at one five-membered ring were investigated. The benzylamino substituted carbamate *syn,syn*-**4a** showed the highest σ_1_ affinity within the group of four stereoisomers emphasizing the importance of the stereochemistry. The cyclohexylmethylamine **18** without further substituents at the propellane scaffold revealed unexpectedly high σ_1_ affinity (*K*_i_ = 34 nM) confirming the relevance of the bioisosteric replacement of the benzylamino moiety by the cyclohexylmethylamino moiety. Reduction of the distance between the basic amino moiety and the “left” hydrophobic region by incorporation of the amino moiety into the propellane scaffold resulted in azapropellanes with particular high σ_1_ affinity. As shown for the propellanamine **18**, removal of the carbamate moiety increased the σ_1_ affinity of **9a** (*K*_i_ = 17 nM) considerably. Replacement of the basic amino moiety by H-bond forming urea did not lead to potent σ ligands. According to molecular dynamics simulations, both azapropellanes *anti*-**5** and **9a** as well as propellane **18** adopt binding poses at the σ_1_ receptor, which result in energetic values correlating well with their different σ_1_ affinities. The affinity of the ligands is enthalpy driven. The additional interactions of the carbamate moiety of *anti*-**5** with the σ_1_ receptor protein cannot compensate the suboptimal orientations of the rigid propellane and its *N*-benzyl moiety within the σ_1_ receptor-binding pocket, which explains the higher σ_1_ affinity of the unsubstituted azapropellane **9a**.

## 1. Introduction

In 1976, Martin and coworkers [1] postulated σ receptors as the third type of opioid receptors. The name σ receptor was derived from the benzomorphan SKF-10,047, which caused a unique pharmacological profile in animal studies. Twenty years later, the σ_1_ receptor was cloned from different tissues of different speces [2,3,4,5,6]. Subsequently, several models of the structure of the σ_1_ receptor were reported, until it was crystallized in 2016, confirming its unique structure [7,8]. The identification of the σ_2_ receptor took an even longer time. In 2017, the identity of the σ_2_ receptor and the endoplasmic reticulum (ER)-resident transmembrane protein 97 (TMEM97) was shown [9,10]. Very recently, the first structure of the human σ_2_ receptor was reported. [11]

The σ_1_ receptor is involved in various neuropsychiatric disorders, such as schizophrenia and depression [12,13,14,15,16]. Several clinically used antidepressants show medium to high σ_1_ receptor affinity in addition to their main mechanism of action [17,18,19,20]. σ_1_ receptors also play a role in drug/alcohol dependence and neurodegenerative disorders (e.g., Alzheimer’s disease) [21,22,23]. The σ_1_ receptor antagonist S1RA has been successfully tested in phase II clinical studies for the treatment of neuropathic pain [24,25]. Since the exact signal transduction path of σ_1_ receptors is not fully understood so far, analgesic activity in neuropathic pain mouse models is the best method to discriminate σ_1_ receptor antagonists from agonists [26,27]. Several human tumors, including prostate, breast and bladder tumors, express a high density of σ_1_ receptors. Strong metastasis and poor prognosis were associated with high expression levels of σ_1_ receptors. Antagonists at σ_1_ receptors were able to reduce tumor cell proliferation [28,29]. Several human tumor cells derived from various tissues (e.g., prostate, breast, colon and lung) overexpress σ_2_ receptors. Agonists at the σ_2_ receptor are capable of killing tumor cells via apoptotic and non-apoptotic machanisms [30,31,32,33,34].

The structures of σ_1_ and σ_2_ receptor ligands are quite diverse. Some prototypical σ ligands containing highly flexible structural elements are displayed in Figure 1. Binding of flexible ligands to a biological target is associated with an entropic penalty, since the binding site of the target forces the flexible ligand into a particular conformation leading to loss of conformational freedom of the ligand.

We are interested in rather rigid ligands with a defined three-dimensional structure fitting exactly into the binding pocket of the target protein. In this respect, we reported on spiro- and bicyclic ligands with high affinity and selectivity for σ_1_ receptors. As an example, the spirocyclic piperidine derivative (*S*)-fluspidine (**1**) is depicted in Figure 2 [35,36,37,38]. (*S*)-Fluspidine (**1**) interacts with low nanomolar affinity with σ_1_ receptors and shows 300-fold selectivity against the σ_2_ subtype. The 18-F-labeled analog [^18^F]**1** is currently investigated as the PET tracer for imaging of σ_1_ receptors in the brain of patients suffering from major depression [39]. The piperazine derivative **2** rigidified by a propano bridge exhibits even higher σ_1_ affinity than **1** and more than 40-fold selectivity over the σ_2_ receptor [40]. On the other hand, the rigid granatane derivative **3** reported by Mach and coworkers [41] represents a ligand with a 30-fold preference for the σ_2_ receptor. (Figure 2)

Inspired by these conformationally restricted spiro- and bicyclic σ ligands **1–3** we introduced the propellane as novel rigid scaffold to achieve high σ_1_ and/or σ_2_ receptor affinity. With K_i_ values of 77 nM and 82 nM the [4.3.3]propellane *syn,syn*-**4a** [42] and the 3-aza[4.4.3]propellane *anti*-**5** [43] represent promising σ_1_ receptor ligands. (Figure 2)

Herein, we started further exploiting the propellane system as rigid scaffold to attach various functional groups and substituents, designed to address σ_1_ and/or σ_2_ receptors. (Figure 3) Due to the rigid structure of the propellane scaffold, all substituents adopt an exact orientation. The manuscript contains three parts. In the first part (compounds of type **A**), the carbamate at the “left” part of the propellane system was kept constant and the substituent at the second cyclopentane ring was modified (compare substituents of granatane **3**). The second part deals predominantly with propellanes of type **B** containing an arylmethylamino moiety (and analogous amino groups) at the “right” side of the propellane system and variations of the “left” side. The third part investigated, whether the amino moiety on the “right” side could be replaced bioisosterically by a urea moiety as shown for compounds of type **C**. The urea is not basic, but represents a strong H-bond donor and acceptor.

## 2. Chemistry

The synthesis of the carbamates **4** started with the mixture of the diastereomeric hydroxyketones *anti*-**6** and *syn*-**6** [42], which was reacted with 2-methoxy-5-methylphenyl isocyanate in the presence of,Bu_2_Sn(OAc)_2_. The carbamates *anti*-**7** and *syn*-**7** were isolated in 36% and 31% yield, respectively. (Scheme 1) The X-ray crystal structure of *syn*-**7** confirmed the successful formation of the carbamate and its *syn*-configuration at 8-position (Figure 4). Moreover, the very long conjoining C–C bond, which belongs to all three rings of the propellane, was confirmed by the crystal structure (C1–C6 = 1.562(2) Å). Final reductive amination of the ketones *anti*-**7** and *syn*-**7** with primary amines and NaBH(OAc)_3_ [44] provided the amines 8-*anti*-**4** and 8-*syn*-**4**. Both type of compounds were obtained as mixture of diastereomers. The ratio of diastereomers was approximately 1:1, respectively.

The affinity of this first set of propellanylcarbamates **4** and **5** towards σ_1_ and σ_2_ receptors was determined in receptor binding studies. In these experiments, the test compounds compete with a radioligand for a limited amount of receptors. The tritium-labeled radioligands [^3^H]-(+)-pentazocine and [^3^H]-di-o-tolylguanidine were used in the σ_1_ and σ_2_ assay, respectively [45,46,47]. The affinity data of the propellanylcarbamates are summarized in Table 1.

The group of diastereomeric benzylamines **4a** show nicely the relationship between the stereochemistry and the σ_1_ and σ_2_ receptor affinity. Whereas the 8-*anti*-derivatives *anti,anti*-**4a** and *anti,syn*-**4a** exhibit very low σ_1_ affinity, the 8-*syn*-derivatives are much more potent and *syn,syn*-**4a** shows the highest σ_1_ affinity within the four diastereomeres (*K*_i_ = 77 nM). Although the granatane derivative **3** reveals high σ_2_ affinity the diastereomeric propellane derivatives **4a** with the same substitution pattern did not interact with σ_2_ receptors up to a concentration of 1 µM [42].

Introduction of various substituents (OCH_3_, Cl, CH_3_, CF_3_, NO_2_, and NMe_2_) at various positions of the benzyl moiety (8-*anti*-**1b**-**1h**) did not lead to remarkable σ_1_ or σ_2_ affinity. The furan-2-yl and indol-3-yl derivatives 8-*anti*-**4i** and 8-*anti*-**4k** show σ_1_ affinity in the range of K_i_ = 1 µM. The corresponding 8-*syn*-derivatives 8-*syn*-**4i** and 8-*syn*-**4k** reveal higher σ_1_ affinity than their 8-*anti*-analogs. The highest σ_1_ affinity of propellanes bearing a hetarylmethyl moiety was found for the furan-2-yl derivative 8-*syn*-**4i** (*K*_i_ = 276 nM).

Removal or extension of the benzyl-CH_2_ moiety provided (substituted) aniline 8-*anti*-**4l** -**4o** or homologous phenylethylamine and phenylpropylamine derivatives 8-*anti*-**4p** and 8-*anti*-**4q**. However, neither removal nor extension of the benzyl-CH_2_ moiety led to considerable σ_1_ or σ_2_ affinity. Only the phenylethylamine derivative 8-*anti*-**4p** shows σ_1_ affinity in the high nanomolar range (*K*_i_ = 668 nM).

With exception of the 5-hydroxypentyl derivative 8-*anti*-**4t** (K_i_(σ_1_) = 454 nM), propellane derivatives with a substituted alkylamino moiety in 11-position (8-*anti*-**4r**-**4u**) show only negligible affinity towards both σ receptor subtypes.

Introduction of the basic amino moiety into the propellane scaffold led to a reduced distance between the basic amino moiety and the carbamate group of the aza-propellanes **5**. The aza-propellane *anti*-**5** displays σ_1_ affinity, which is comparable with the σ_1_ affinity of *syn,syn*-**4a**.^43^ It has to be noted that the stereodescriptors change due to a change of the hierarchy of the three rings by expansion of one cyclopentane moiety into a piperidine ring. Thus, the three-dimensional structure of *syn,syn*-**4a** corresponds to that of *anti*-**5**, which explains the similar σ_1_ affinity (K_i_ = 77 nM and 82 nM) [43].

In the second part of the manuscript, the focus lies on propellanes containing arylmethylamino moieties on the “right” side and various substituents on the “left” side. Although the synthesis of benzylamine **8** (Figure 5) has already been reported in literature [48], its affinity towards σ receptors is missing. The 3-azapropellanes **9** (Figure 5) show promising σ_1_ affinity, which depends on the substitution pattern and the configuration.^43^ Therefore, we decided to investigate the σ receptor affinity of the benzylamine **8** and modify the substituents in the 11-position.

At first, the ketones **10**–**12** were reductively aminated with benzylamine and NaBH(OAc)_3_^44^ to yield the benzylamines **8**, **13** and **14**. The pure diastereomeric alcohols *anti*-**11** and *syn*-**11** were converted separately into the benzylamines 11-*anti*-**13** and 11-*syn*-**13**, respectively, which were not further separated. Whereas the *syn*- and *anti*-configured benzylamines **8** were also not separated, the diastereomeric 1,3-dioxolanes **14** were separated by flash chromatography to obtain diastereomerically pure 8-*anti*-**14** and 8-*syn*-**14**. (Scheme 2).

Furthermore, the unsubstituted ketone **10** was reductively aminated with tryptamine to yield the indolylethylamine **15**. The mixture of diastereomeric benzylamines *anti*-**8** and *syn*-**8** was treated with ammonium formate and Pd(OH)_2_ removing reductively the benzyl moiety. Upon treatment with aldehydes and NaBH(OAc)_3_ [44], the resulting primary amine **16** was further transformed into the dimethylamino substituted benzylamine **17** and the cyclohexylmethylamine **18**. (Scheme 2).

The σ_1_ affinity of the 3-azapropellane *anti*-**9b** with an OH moiety at 12-position is three-fold higher than the σ_1_ affinity of its *syn*-diastereomer *syn*-**9b**. (Table 2) However, the low nanomolar σ_1_ affinity of the unsubstituted 3-azapropellane **9a** (*K*_i_ = 17 nM) was unexpected.^42,43,48^ Compared to the naked azapropellane **9a**, the σ_1_ affinity of the naked propellanamine **8** is 35-fold reduced. The 3-azapropellanes **9a** and *anti*-**9b** and the naked propellanamine **8** show at least 10-fold selectivity for the σ_1_ receptor over the σ_2_ receptor. The least potent σ_1_ ligand *syn*-**9b** has only a slight preference for the σ_1_ receptor over the σ_2_ subtype (Table 2).

Modifications of the 11-substituent and the *N*-substituent led to propellanamines **13–17** with very low σ_1_ and σ_2_ affinity. However, the cyclohexylmethyl moiety increased the σ_1_ and the σ_2_ affinity remarkably. With *K*_i_ values of 24 nM (σ_1_ affinity) and 101 nM (σ_2_ affinity), **18** represents the most potent σ ligand of this series of compounds. The increase of both σ_1_ and σ_2_ affinities by introduction of the cyclohexylmethyl moiety instead of the benzyl moiety has already been observed for some other classes of σ ligands [49,50].

In the third part of this project, the amino moiety on the “right” side was replaced by a urea to investigate, whether this H-bond donor and H-bond acceptor group could mimic the basic amino moiety. For this purpose, the propellane-8,11-dione **19** was reductively aminated with benzylamine. The resulting secondary amines *syn*-**20** and *anti*-**20** were separated by flash column chromatography and subsequently reacted with 2-methoxy-5-methylphenyl isocyanate to obtain the urea derivatives *syn*-**21** and *anti*-**21**, respectively. Final reduction of the ketones *syn*-**21** and *anti*-**21** with NaBH_4_ provided the four diastereomeric *N*-benzylurea derivatives *syn,anti*-**22**, *syn,syn*-**22**, *anti,anti*-**22** and *anti,syn*-**22** bearing a secondary alcohol in 11-position (Scheme 3).

The *N*-benzylurea derivatives *anti*-**21**, *syn,anti*-**22** and *anti,syn*-**22** were crystallized from EtOAc, leading to crystals suitable for X-ray crystal structure analysis. (Figure 6, Figure 7 and Figure 8).

The crystal structure of *anti*-**21** nicely confirms the anticonfiguration of the urea at the propellane system. The conjoining bond C1–C6 is rather long (1.550(3) Å). The cyclohexane ring adopts a chair conformation and the cyclopentane ring bearing the urea adopts an envelope conformation, which leads to an outward orientation of the large urea (Figure 6).

The X-ray crystal structures of the diastereomeric *N*-benzylurea derivatives *syn,anti*-**22** and *anti,syn*-**22** containing an additional OH moiety in 11-position are shown in Figure 7 and Figure 8. Both structures prove the *syn,anti*- and *anti,syn*-configuration, respectively. The six-membered ring of the propellane scaffold of *syn,anti*-**22** adopts a chair conformation. However, in the diastereomer *anti,syn*-**22** a boat-like conformation was found for the cyclohexane ring. This boat-like conformation leads to an extraordinarily long conjoining bond (C1–C6 = 1.590(2) Å). Both five-membered rings of urea derivative *anti*,*syn*-**22** adopt unusual envelope conformations and all three rings of the propellane scaffold flap in the same direction (Figure 8). This pattern was only reported for heterocyclic 8,11-dioxa[4.3.3]propellanes [51].

To obtain urea derivatives without the *N*-benzyl substituent, the primary amine **16** was reacted with various isocyanates. Since the primary amine **16** was employed as 1.1-mixture of *anti*- and *syn*-diastereomers, the urea derivatives **23a**–**d** were obtained as 1:1-mixture of *anti*- and *syn*-diastereomers as well (Scheme 4).

Finally, the dioxolane substituted benzylamine **14** (mixture of *syn*- and *anti*-diastereomers) was debenzylated by a transfer hydrogenolysis using NH_4_HCO_2_ in the presence of Pd(OH)_2_. The mixture of diastereomeric primary amines **24** was isolated in 75% yield and converted into urea upon treatment with 3,4-difluorophenyl isocyanate. After hydrolysis of the dioxolane, the diastereomeric difluorophenylurea derivatives *syn*-**25** and *anti*-**25** were isolated in 42% and 35% yield, respectively. Reduction of the ketones *syn*-**25** and *anti*-**25** with NaBH_4_ provided diastereomeric alcohols **26**. Whereas the mixture of *syn,anti*-**26** and *syn,syn*-**26** was obtained as 1:1-mixture of diastereomers, *anti,anti*-**26** and *anti,syn*-**26** were separated by flash chromatography (Scheme 5).

The *syn*-configuration of *syn*-**25** was confirmed unequivocally by X-ray crystal structure analysis. In addition to the *syn*-configuration the chair conformation of the cyclohexane ring and the long conjoining bond C1–C6 = 1.546(3) Å was demonstrated (Figure 9).

The σ_1_ and σ_2_ affinities of the urea derivatives were determined in receptor binding studies. Unfortunately, the synthesized urea were not able to compete with the radioligands proving that the urea could not replace bioisosterically the basic amino moiety (Table 3).

## 3. Computational Studies

Within the class of propellanamines and azapropellanes some promising σ_1_ receptor ligands were identified. Therefore, molecular dynamics (MD) simulations were performed to shed light on their mechanism of binding. The starting structure for the σ_1_ receptor was obtained from the RCSB Protein Data Bank (PDB ID 5HK1) [7]. Following a consolidated computational protocol [49,52], the binding modes of compounds **9a** (Figure 10A,C) and *anti*-**5** (Figure 10B,D) were initially recognized. A MM/PBSA (molecular mechanics/Poisson–Boltzmann surface area) approach [53] provided the binding free energy (ΔG) of the complexes of both compounds with the σ_1_ receptor. The obtained energetic values are in good agreement with their different σ_1_ affinity. The following ΔG values were obtained: −10.02 kcal/mol for **9a** (*K*_i_(σ_1_) = 17 nM) and −8.87 kcal/mol for anti**-5** (*K*_i_(σ_1_) = 82 nM).

As expected, both σ_1_ ligands share the same thermodynamics pattern; their binding is enthalpy driven characterized by favorable van der Waals and electrostatic interactions. Specifically, our analysis resulted in an enthalpy contribution (ΔH) of −18.47 kcal/mol and −17.58 kcal/mol for **9a** and anti-**5**, respectively. Instead, the entropic components (−TΔS) penalize the binding with the σ_1_ receptor with the corresponding values of 8.45 kcal/mol and 8.71 kcal/mol for **9a** and anti-**5**, respectively.

Through the per-residue binding free energy deconvolution (PRBFED) of the enthalpic terms (ΔH_res_), the main amino acid residues of the σ_1_ receptor involved in ligand binding were identified. Basically, by elucidation of the specific ligand/protein interactions, the PRBFED analysis (Figure 10E) allowed to better understand the preference of the σ_1_ receptor for the smaller azapropellane **9a**. Specifically, azapropellane **9a** is provided with the basic chemical features of a prototypical σ_1_ receptor ligand: the rings of the propellane system can perform stabilizing hydrophobic and van der Waals interactions with the σ_1_ residues W89, F107, Y120 and W164 (ΔH_res_ = −3.27 kcal/mol, Figure 10E) while the *N*-benzyl group is perfectly encased in an hydrophobic cavity underlying σ_1_ residues I124,V152, and H154 (ΔH_res_ = −1.64 kcal/mol). However, the most peculiar interaction is definitively performed by its charged N-atom, involved in a permanent ionic interaction with the carboxylic group of E172 (ΔH_res_ = −4.93 kcal/mol) with a detected average dynamics length (ADL) of 1.75 ± 0.12 Å (Figure 10A,F). Moreover, the optimal orientation of this interaction is guaranteed by a stable, internal hydrogen bond between the other O-atom of E172 and the OH moiety of Y103 (ADL = 1.73 ± 0.12 Å, Figure 10A,G). In the same series, the propellanamine derivative **18** exhibited a good σ_1_ affinity (*K*_i_(σ_1_) = 34 nM). Our MD study confirmed a very similar binding mode and interaction spectra as observed for the azapropellane **9a** (Appendix A, Appendix A). Indeed, the charged secondary amine maintained the ionic interaction with E172 (ΔH_res_ = −4.44 kcal/mol, Appendix A) and the cyclohexyl moiety of **18** is able to positively interact in the hydrophobic cavity with σ_1_ residues I124, V152 and H154 (ΔH_res_ = −1.72 kcal/mol). Accordingly, the σ_1_/**18** complex is provided with a favorable ΔG value of −9.78 kcal/mol.

Compared to **9a**, the enthalpic contribution of *anti*-**5** was almost 1 kcal/mol lower, although its 2-methoxy-5-methylphenyl carbamate moiety could provide additional interactions in the σ_1_ receptor cavity (Figure 10B,D). However, these additional interactions of *anti*-**5** with residues L95, L182 and Y206 (ΔH_res_ = −1.68 kcal/mol, Figure 10E) were not specific and could not compensate the decrease of the other interactions due to a not optimal orientation in the binding site. Indeed, *anti*-**5** cannot establish an optimal H-bond with the carboxylate side chain of E172 (ΔH_res_ = −3.65 kcal/mol) as demonstrated by the less stable ADL detected in our MD simulation (ADL = 2.12 ± 0.35 Å, Figure 10B,F). Furthermore, the assumed binding pose of *anti*-**5** does not allow an optimal position of its *N*-benzyl moiety in the hydrophobic cavity underlying I124, V152 and H154 (ΔH_res_ = −0.53 kcal/mol, Figure 10E). Moreover, optimal stabilizing interactions of the propellane system of *anti*-**5** with W89, F107, Y120 and W164 (ΔH_res_ = −2.58 kcal/mol) were not reached.

## 4. Conclusions

In this manuscript, the rigid propellane system was used to attach σ_1_ and σ_2_ pharmacophoric substituents in a defined three-dimensional orientation. It was shown, that propellanes with both substituents at the five-membered rings adopting *syn*-configuration (e.g., *syn*,*syn*-**4a**) exhibited high σ_1_ affinity underlining the importance of the stereochemistry. Urea instead of the amino moiety led to the loss of σ_1_ and σ_2_ affinity. High σ_1_ affinity was achieved by incorporation of the basic amino moiety into the propellane scaffold. The azapropellanes *anti*-**5** and **9a** demonstrated high σ_1_ affinity, which is enthalpy driven. Although the carbamate moiety of *anti*-**5** contributed to the binding free energy of *anti*-**5** within the σ_1_ receptor binding pocket, it forces the complete propellane scaffold and its benzyl moiety into a less favorable orientation in the binding pocket. As a result, the σ_1_ affinity of *anti*-**5** is lower than the σ_1_ affinity of **9a**.

## 5. Experimental

### 5.1. Chemistry, General Methods

Unless otherwise mentioned, THF was dried with sodium/benzophenone and was freshly distilled before use. Thin layer chromatography (tlc): Silica gel 60 F_254_ plates (Merck). Preparative thin layer chromatography (ptlc): Silica gel 60 F_254_, plates (Merck) 20 × 20 cm, layer thickness 2 mm flash chromatography (fc): Silica gel 60, 40–64 μm (Merck); parentheses include: diameter of the column, length of column, fraction size, eluent, R*_f_* value. Melting point: Melting point apparatus SMP 3 (Stuart Scientific), uncorrected. IR: IR spectrophotometer IRAffinity with MIRacle 10 accessory FT-ATR-IR (Shimadzu). ^1^H NMR (400 MHz), ^13^C NMR (100 MHz): Mercury plus 400 spectrometer (Varian); δ in ppm relative to tetramethylsilane; coupling constants are given with 0:5 Hz resolution. Where necessary, the assignment of the signals in the ^1^H NMR and ^13^C NMR spectra was performed using ^1^H-^1^H COSY, ^1^H-^13^C HSQC NMR spectra, the stereochemistry was assigned using NOESY NMR spectra. MS: EI = electron impact and ESI = electrospray ionization: MicroTof (Bruker Daltronics, Bremen, Germany), calibration with sodium formate clusters before measurement. HPLC method for determination of the product purity: Merck Hitachi Equipment(Darmstadt, Germany); UV detector: L-7400; autosampler: L-7200; pump: L-7100; degasser: L-7614; Method: column: LiChrospher^®^ 60 RP-select B (5 mm), 250–4 mm cartridge; flow rate: 1:00 mLmin1; injection volume: 5:0 mL; detection at l = 210 nm; solvents: A: water with 0:05% (*v*/*v*) trifluoroacetic acid; B: acetonitrile with 0:05% (*v*/*v*) trifluoroacetic acid: gradient elution: (A%): 0–4 min: 90%, 4–29 min: gradient from 90% to 0%, 29–31 min: 0%, 31–31:5 min: gradient from 0% to 90% and 31:5–40 min: 90%.

### 5.2. General Procedure A for the Synthesis of Carbamates and Ureas

Under N_2_, propellanol of propellanamine (1 equi.), the respective isocyanate (1.2 eq.) and the catalyst Bu_2_Sn(OAc)_2_ (0.2 eq.) were dissolved in THF (5 mL per 100 mg of propellanamine) and the mixture was stirred at rt for 24–48 h. Water (5 mL) was added and the mixture was stirred vigorously for 20 min. The mixture was extracted with EtOAc (3×), the combined EtOAc layers were washed with brine (1×), dried (Na_2_SO_4_), filtered, the filtrate was concentrated in vacuo and the residue was purified by fc.

### 5.3. Synthetic Procedures

#### 5.3.1. (*anti*-11-Oxo[4.3.3]propellan-8-yl) *N*-(2-methoxy-5-methylphenyl)carbamate (*anti*-7) and (*syn*-11-Oxo[4.3.3]propellan-8-yl) *N*-(2-methoxy-5-methylphenyl)carbamate (*syn*-7)

According to General Procedure A, stereoisomeric hydroxyketones *anti*-**6** and *syn*-**6** (1.3 g, 6.69 mmol), 2-methoxy-5-methylphenyl isocyanate (1.4 g, 8.71 mmol) and the catalyst Bu_2_Sn(OAc)_2_ (0.36 mL, 1.34 mmol) were reacted in THF (25 mL) and worked-up. The residue was purified by fc (5 cm, cyclohexane:ethyl acetate = 1:0–7:3, 50 mL).

*anti*-**7** (R_f_ = 0.52): Pale yellow solid, mp 143–144 °C, yield 0.86 g (36%), C_21_H_27_NO_4_ (357.2). MS (EI): *m/z* (%) = 357 [M^+^], 181 [M-C_12_H_17_O]^+^, 137 [M-C_13_H_17_O_3_]^+^, 122 [M-C_13_H_18_NO_3_]^+^. Exact mass (APCI): *m/z* = 358.1979 (calcd. 358.2013 for C_21_H_28_NO_4_ [M+H]^+^). FT-IR (ATR, film): ν (cm^−1^) = 3292 (ν *N-H*), 2931 (ν *C-H* aliphatic), 1732 (ν *C=O*_ketone_), 1697 (ν *C=O*_carbamate_), 1597 (δ *N-H*). ^1^H NMR (CDCl_3_): δ (ppm) = 1.32–1.45 (m, 8H, 2-CH_2_, 3-CH_2_, 4-CH_2_, 5-CH_2_), 1.75 (dd, *J* = 14.9/4.1Hz, 2H, 7-H_anti_, 9-H_anti_), 2.22 (s, 3H, C*H_3_*) 2.24 (d, *J* = 19 Hz, 2H, 10-H_syn_, 12-H_syn_,), 2.35 (d, *J* = 19 Hz, 2H, 10-H_anti_, 12-H_anti_), 2.36 (dd, *J* = 14.8/8.8 Hz, 2H, 7-H_syn_, 9-H_syn_), 3.77 (s, 3H, OC*H*_3_), 5.32 (tt, *J* = 8.4/4.4 Hz, 1H, 8-H), 6.67 (d, *J* = 8.3 Hz, 1H, 3-H_N-Ar_), 6.71 (dd, *J* = 1.6/8.3 Hz, 1H, 4-H_N-Ar_), 7.06 (s, broad, 1H, N*H* _carbamate_), 7.83 (s, broad, 1H, 6-H_N-Ar_). ^13^C NMR (CDCl_3_): δ (ppm) = 21.2 (1C, *C*H_3_), 21.4 (2C, C-3, C-4), 32.1 (2C, C-2, C-5), 43.9 (2C, C-7, C-9), 47.7, (2C, C1, C-6), 49.9 (2C, C-10, C-12), 55.9 (1C, O*C*H_3_), 75.6 (1C, C-8), 110.0 (1C, C-3_N-Ar_), 118.9 (1C, C- _N-Ar_), 123.2 (1C, C-4_N-Ar_), 127.4 (1C, C-1_N-Ar_), 130.7 (1C, C-5_N-Ar_), 145.7 (1C, C-2_N-Ar_), 153.3 (1C, NH(*C*O)O), 219.2 (1C, *C*=O_ketone_). Purity (HPLC): 95.9% (*t_R_* = 21.40 min).

*syn*-**7** (R_f_ = 0.48): Pale yellow solid, mp 115–118 °C, yield 0.74 g (31%), C_21_H_27_NO_4_ (357.2). MS (EI): *m/z* (%) = 357 [M^+^], 181 [M-C_12_H_17_O]^+^, 137 [M-C_13_H_17_O_3_]^+^, 122 [M-C_13_H_18_NO_3_]^+^. Exact mass (APCI): *m/z* = 358.1991 (calcd. 358.2013 for C_21_H_28_NO_4_ [M+H]^+^ ). FT-IR (ATR, film): ν (cm^−1^) = 3425 (ν *N-H*), 2931 (ν *C-H* aliphatic), 1737 (ν *C=O*_ketone_), 1720 (ν *C=O*_carbamate_), 1597 (δ *N-H*). ^1^H NMR (CDCl_3_): δ (ppm) = 1.33–1.43 (m, 4H, 2-H_eq_, 3-H_eq_, 4-H_eq_, 5-H_eq_), 1.47–1.51 (m, 2H, 2-H_ax_, 4-H_ax_), 1.62–1.68 (m, 2H, 3-H_ax_, 5-H_ax_), 1.97 (dd, *J* = 15.2/3.6 Hz, 2H, 7-H_syn_, 9-H_syn_), 2.10 (d, *J* = 19.2 Hz, 2H, 10-H_anti_, 12-H_anti_,), 2.15 (dd, *J* = 15.2/8.4 Hz, 2H, 7-H_anti_, 9-H_anti_), 2.21 (d, *J* = 19.2 Hz, 2H, 10-H_syn_, 12-H_syn_,), 2.23 (s, 3H, C*H*_3_), 3.79 (s, 3H, OC*H*_3_), 5.27 (tt, *J* = 8.7/3.7 Hz, 1H, 8-H), 6.68 (d, *J* = 8.4 Hz, 1H, 3-H_N-Ar_), 6.72 (dd, *J* = 2/8.4 Hz, 1H, 4-H_N-Ar_), 7.11 (s, broad, 1H, NH), 7.85 (s, broad, 1H, 6-H_N-Ar_). ^13^C NMR (CDCl_3_): δ (ppm) = 21.2 (1C, CH_3_), 21.7 (2C, C-3, C-4), 31.9 (2C, C-2, C-5), 43.9 (2C, C-7, C-9), 47.6 (2C, C-1, C-6), 49.7 (2C, C-10, C-12), 56.0 (1C, O*C*H_3_), 75.6 (1C, C-8), 110.1 (1C, C-3_N-Ar_), 119.0 (1C, C-6_N-Ar_), 123.1 (1C, C-4_N-Ar_), 127.4 (1C, C-1_N-Ar_), 130.8 (1C, C-5_N-Ar_), 145.7 (1C, C-2_N-Ar_), 153.4 (1C, NH(*C*O)O), 218.7 (1C, *C*=O_ketone_). Purity (HPLC): 95.5% (*t_R_* = 21.79 min).

#### 5.3.2. [(8-*anti*-11-*anti* and 8-*anti*-11-*syn*)-11-(3,4-Dimethoxbenzylamino)[4.3.3]propellan-8-yl] *N*-(2-methoxy-5-methylphenyl)carbamate (8-*anti*-4b)

NaBH(OAc)_3_ (0.47 g, 2.24 mmol) was added to a solution of ketone *anti*-**7** (0.20 g, 0.56 mmol), 3,4-dimethoxybenzylamine (0.11 g, 0.67 mmol) and acetic acid (32 μL, 0.56 mmol) in 1,2-dichloroethane (10 mL, dried over molecular sieves 4 Å). The mixture was stirred at rt for 48 h. Then NaOH (1 M) was added (pH 8–10), the mixture was extracted with CH_2_Cl_2_ (3×) and the combined organic layers were washed with brine (1×), dried (Na_2_SO_4_), filtered, the filtrate was concentrated in vacuo and the residue was purified by fc (3 cm, cyclohexane:ethyl acetate = 7:3 to 5:5, 10 mL, R*_f_* = 0.25, cyclohexane:ethyl acetate = 7:3) to obtain a mixture of diastereoisomeric aminocarbamates *anti,anti-***4b** and *anti,syn*-**4b** as brown oil, yield 0.24 g (85%). C_30_H_40_N_2_O_5_ (508.6). Exact mass (APCI): *m/z* = 509.2982 (calcd.509.3010 for C_30_H_41_N_2_O_5_ [M+H]^+^). FT-IR (ATR, film): ν (cm^−1^) = 3329 (ν *N-H*), 2927 (ν *C-H* aliphatic), 1724 (ν *C=O*), 1597 (δ *N-H*). ^1^H NMR (400 MHz, CDCl_3_): δ (ppm) = 1.29–1.49 (m, 8H, 2-CH_2_, 3-CH_2_, 4-CH_2_, 5-CH_2_), 1.61 (dd, *J* = 13.4/6.6 Hz, 2 × 0.5H, 7-H, 9-H), 1.67–1.75 (m, 4 × 0.5H, 7-H, 9-H, 10-H, 12-H), 1.92 (dd, *J* = 14.4/4.8 Hz, 2 × 0.5H, 10-H, 12-H), 1.98–2.10 (m, 2H, 7-H, 9-H), 2.14–2.26 (m, 2H, 10-H, 12-H), 2.28 (s, 3H, CH_3_), 3.37–3.51 (m, 2 × 0.5H, 11-H), 3.70 (s, 2H, NCH_2_Ar), 3.82 (s, 3 × 0.5H, *p*-OCH_3_), 3.84 (s, 3 × 0.5H, *p*-OCH_3_), 3.86 (s, 3 × 0.5H, OCH_3Arylcarbamate_), 3.88 (s, 3 × 0.5H, *m*-OCH_3_), 3.89 (s, 3 × 0.5H, *m*-OCH_3_), 5.19–5.31 (m, 2 × 0.5H, 8-H), 6.73 (dd, *J* = 8.3/3.5 Hz, 1H, 6-H_Bn_), 6.77 (m, 1H, 5-H_Bn_), 6.81 (dd, *J* = 8.1/2.1 Hz, 1H, 4-H_Ar_), 6.85 (d, *J* = 8.4 Hz, 1H, 3-H_Ar_), 6.90 (t, *J* = 2.1 Hz, 1H, 2-H_Bn_), 7.13 (s, 0.5H, NH), 7.18 (s, 0.5H, NH), 7.93 (s, 1H, 6-H_Ar_). A signal for the NH proton is not seen in the spectrum. ^13^C NMR (100 MHz, CDCl_3_): δ (ppm) = 20.8, 20.9 (2C, C-3, C-4), 21.2 (1C, *C*H_3_), 32.4, 32.7 (2C, C-2, C-5), 44.5, 44.8 (2C, C-10, C-12), 45.0 (2C, C-7, C-9), 49.7 (2C, C-1, C-6), 55.2 (1C, NH*C*H_2_Ph), 55.7 (1C, C-11), 55.9, 59.1 (3C, 3 × O*C*H_3_), 76.2 (1C, C-8), 110.2 (1C, C-3_Ar_), 111.2 (1C, C-2_Bn_), 112.1 (1C, C-5_Bn_), 119.0 (1C, C-6_Ar_), 121.1 (1C, C-6_Bn_), 122.9 (1C, C-4_Ar_), 127.8 (1C, C-1_Ar_), 129.5 (1C, C-1_Bn_), 130.8 (1C, C-5_Ar_), 145.7 (1C, C-2_Ar_), 148.1, 148.8 (2C, C-3_Bn_, C-4_Bn_) 153.6 (1C, C=O). *anti,anti*-**4b**:*anti,syn*-**4b** = 1:1. Purity (HPLC): 96.0% (*t_R_* = 20.31 min).

#### 5.3.3. [(8-anti-11-*anti* and 8-*anti*-11-*syn*)-11-(4-Chlorobenzylamino)[4.3.3]propellan-8-yl] *N*-(2-methoxy-5-methylphenyl)carbamate (8-*anti*-4c)

NaBH(OAc)_3_ (0.47 g, 2.24 mmol) was added to a solution of ketone *anti*-**7** (0.20 g, 0.56 mmol), 4-chlorobenzylamine (95 mg, 0.67 mmol) and acetic acid (32 μL, 0.56 mmol) in 1,2-dichloroethane (10 mL, dried over molecular sieves 4 Å). The mixture was stirred at rt for 6 days. Then NaOH (1 M) was added (pH 8–10), the mixture was extracted with CH_2_Cl_2_ (3×) and the combined organic layers were washed with brine (1×), dried (Na_2_SO_4_), filtered, the filtrate was concentrated in vacuo and the residue was purified by fc (3 cm, cyclohexane:ethyl acetate:methanol = 5:3.5:1.5, 10 mL, R*_f_* = 0.40) to obtain a mixture of diastereoisomeric aminocarbamates *anti,anti-***4c** and *anti,syn*-**4c** as yellow oil, yield 0.12 g (45%). C_28_H_35_ClN_2_O_3_ (482.2). Exact mass (APCI): *m/z* = 483.2387 (calcd.483.2409 for C_28_H_36_ClN_2_O_3_ [M+H]^+^). FT-IR (ATR, film): ν (cm^−1^) = 3329 (ν *N-H*), 2927 (ν *C-H* aliphatic), 1724 (ν *C=O*), 1597 (δ *N-H*). ^1^H NMR (400 MHz, CDCl_3_): δ (ppm) = 1.30–1.53 (m, 8H, 2-CH_2_, 3-CH_2_, 4-CH_2_, 5-CH_2_), 1.59–1.76 (m, 2H, 7-H, 9-H), 1.89–2.09 (m, 4H, 10-CH_2_, 12-CH_2_), 2.14–2.27 (m, 2H, 7-H, 9-H), 2.29 (s, 3H, CH_3_), 3.35–3.48 (m, 2 × 0.5H, 11-H), 3.73 (s, 2H, NCH_2_Ph), 3.85 (s, 3H, OCH_3_), 5.18–5.31 (m, 2 × 0.5H, 8-H), 6.73 (dd, *J* = 8.3/2.1 Hz, 1H, 4-H_Ar_), 6.77 (d, *J* = 8.3 Hz, 1H, 3-H_Ar_), 7.13 (s, 0.5H, NH), 7.18 (s, 0.5H, NH), 7.25–7.36 (m, 4H, 4-chlorophenyl), 7.93 (s, 1H, 6-H_Ar_). A signal for the NH proton is not seen in the spectrum. ^13^C NMR (100 MHz, CDCl_3_): δ (ppm) = 21.0, 21.1 (2C, C-3, C-4), 21.5 (1C, CH_3_), 32.4, 32.8 (2C, C-2, C-5), 44.7, 44.8, 45.0, 45.2 (4C, C-7, C-9, C-10, C-12), 49.7 (2C, C-1, C-6), 50.5 (1C, NCH_2_Ph), 55.7, 55.9 (2 × 0.5C, OCH_3_), 57.0 (1C, C-11), 76.0, 76.5 (2 × 0.5C, C-8), 109.9 (1C, C-3_Ar_), 118.9 (1C, C-6_Ar_), 122.9 (1C, C-4_Ar_), 127.6, 128.6 (2C, C-3_Bn_, C-5_Bn_), 129.6, 129.8 (2C, C-2_Bn_, C-6_Bn_), 130.7 (1C, C-5_Ar_), 138.7 (1C, C-1_Bn_) 145.6 (1C, C-2_Ar_), 153.6 (1C, C=O). *anti,anti*-**4c**:*anti,syn*-**4c** = 1:1. Purity (HPLC): 92.6% (*t_R_* = 20.85 min).

#### 5.3.4. [(8-*anti*-11-*anti* and 8-*anti*-11-*syn*)-11-(3,4-Dichlorobenzylamino)[4.3.3]propellan-8-yl] *N*-(2-methoxy-5-methylphenyl)carbamate (8-*anti*-4d)

NaBH(OAc)_3_ (0.47 g, 2.24 mmol) was added to a solution of ketone *anti*-**7** (0.20 g, 0.56 mmol), 3,4-dichlorobenzylamine (0.12 g, 0.67 mmol) and acetic acid (32 μL, 0.56 mmol) in 1,2-dichloroethane (10 mL, dried over molecular sieves 4 Å). The mixture was stirred at rt for 6 d. Then NaOH (1 M) was added (pH 8–10), the mixture was extracted with CH_2_Cl_2_ (3×) and the combined organic layers were washed with brine (1×), dried (Na_2_SO_4_), filtered, the filtrate was concentrated in vacuo and the residue was purified by fc (3 cm, cyclohexane:ethyl acetate:methanol = 5:3.5:1.5, 10 mL, R*_f_* = 0.49) to obtain a mixture of diastereoisomeric aminocarbamates *anti,anti-***4d** and *anti,syn*-**4d** as yellow oil, yield 0.17 g (59%). C_28_H_34_Cl_2_N_2_O_3_ (517.5). Exact mass (APCI): *m/z* = 517.2015 (calcd.517.2019 for C_28_H_35_Cl_2_N_2_O_3_ [M+H]^+^). FT-IR (ATR, film): ν (cm^−1^) = 3329 (ν *N-H*), 2927 (ν *C-H* aliphatic), 1724 (ν *C=O*), 1597 (δ *N-H*). ^1^H NMR (400 MHz, CDCl_3_): δ (ppm) = 1.28–1.55 (m, 8H, 2-CH_2_, 3-CH_2_, 4-CH_2_, 5-CH_2_), 1.60–1.75 (m, 2 × 1.5H, 7-H, 9-H), 1.88–2.10 (m, 4H, 10-CH_2_, 12-CH_2_), 2.14–2.25 (m, 2 × 0.5H, 7-H, 9-H), 2.28 (s, 3H, CH_3_), 3.34–3.48 (m, 2 × 0.5H, 11-H), 3.72 (s, 2H, NCH_2_Ph), 3.83 (s, 3H, OCH_3_), 5.17–32 (m, 2 × 0.5H, 8-H), 6.73 (dd, *J* = 8.2/1.8 Hz, 1H, 4-H_Ar_), 6.77 (d, *J* = 8.7 Hz, 1H, 3-H_Ar_), 7.13 (s, 1H, NH), 7.20 (dd, *J* = 7.4/3.2 Hz, 1H, 5-H_3,4-dichlorophenyl_), 7.38 (dd, *J* = 8.2/5.3 Hz, 1H, 6-H_3,4-dichlorophenyl_), 7.46 (dd, *J* = 5.0/2.1 Hz, 1H, 2-H_3,4-dichlorophenyl_), 7.92 (s, 1H, 6-H_Ar_). A signal for the NH proton is not seen in the spectrum. ^13^C NMR (100 MHz, CDCl_3_): δ (ppm) = 21.0 (1C, CH_3_), 21.1, 21.5 (2C, C-3, C-4), 32.4, 32.8 (2C, C-2, C-5), 44.6, 44.8, 45.0, 45.1 (4C, C-7, C-9, C10, C-12), 49.8, 50.5 (2C, C1, C-6), 51.7 (1C, NCH_2_Ph), 55.9 (1C, OCH_3_), 57.1 (1C, C-11), 76.0, 76.4 (2 × 0.5C, C-8), 110.0(1C, C-3_Ar_), 118.9 (1C, C-6_Ar_), 122.9 (1C, C-4_Ar_),125. 6, 127.8 (2C, C-2_3,4-dichlorophenyl_, C-6_3,4-dichlorophenyl_), 103.3, 130.4, 130.5, 130.7, 130.8 (5C, C-3_3,4-dichlorophenyl_, C-4_3,4-dichlorophenyl_, C-5_3,4-dichlorophenyl_, C-1_Ar_, C-5Ar), 132.5 (1C, C-1_3,4dichlorophenyl_) 146.0 (1C, C-2_Ar_), 153.6 (1C, C=O). *anti,anti*-**4d**:*anti,syn*-**4d** = 1:1. Purity (HPLC): 88.6% (*t_R_* = 21.17 min).

#### 5.3.5. [(8-*anti*-11-*anti* and 8-*anti*-11-*syn*)-11-(2,4-Dimethylbenzylamino)[4.3.3]propellan-8-yl] *N*-(2-methoxy-5-methylphenyl)carbamate (8-*anti*-4e)

NaBH(OAc)_3_ (0.3 g, 1.42 mmol) was added to a solution of ketone *anti*-**7** (0.10 g, 0.28 mmol), 3,4-dimethylbenzylamine (45 mg, 0.34 mmol) and acetic acid (16 μL, 0.28 mmol) in 1,2-dichloroethane (5 mL, dried over molecular sieves 4 Å). The mixture was stirred at rt for 72 h. Then NaOH (1 M) was added (pH 8–10), the mixture was extracted with CH_2_Cl_2_ (3×) and the combined organic layers were washed with brine (1×), dried (Na_2_SO_4_), filtered, the filtrate was concentrated in vacuo and the residue was purified by fc (3 cm, cyclohexane:ethyl acetate:methanol = 5.5:3.5:1, 20 mL, R*_f_* = 0.21) to obtain a mixture of diastereoisomeric aminocarbamates *anti,anti-***4e** and *anti,syn*-**4e** as brown oil, yield 0.12 g (93%). C_30_H_40_N_2_O_3_ (476.7). Exact mass (APCI): *m/z* = 477.3159 (calcd. 477.3112 for C_30_H_41_N_2_O_3_ [M+H]^+^). FT-IR (ATR, film): ν (cm^−1^) = 3329 (ν *N-H*), 2927 (ν *C-H* aliphatic), 1724 (ν *C=O*), 1597 (δ *N-H*). ^1^H NMR (600 MHz, CDCl_3_): δ (ppm) = 1.24–1.70 (m, 9H, 2-CH_2_, 3-CH_2_, 4-CH_2_, 5-CH_2_, 7-CH_2_(0.5H), 9-CH_2_(0.5H)), 1.97–2.12 (m, 3H, 7-CH_2_(0.5H), 9-CH_2_(0.5H)), 10-CH_2_(1H), 12-CH_2_(1H)), 2.14–2.37 (m, 13H, 7-CH_2_(1H), 9-CH_2_(1H), 10-CH_2_(1H), 12-CH_2_(1H), CH_3_), 3.45–3.51 (m, 0.5H, 11-H), 3.51–3.58 (m, 0.5H, 11-H), 3.76 (s, 2H, NCH_2_Ph), 3.82 (s, 3 × 0.5H, OCH_3_), 3.85 (s, 3 × 0.5H, OCH_3_), 5.20 (tt, *J* = 8.4/4.0 Hz 0.5H, 8-H), 5.28 (tt, *J* = 8.4/5.0 Hz, 0.5H, 8-H), 6.71–6.80 (m, 2H, 3-H_Ar_, 4-H_Ar_), 6.95–7.03 (m, 2H, 5-H_2,4-diMePhenyl_, 6-H_2,4-diMePhenyl_), 7.11 (d, *J* = 3.8 Hz, 1H, 3-H_2,4-diMePhenyl_), 7.36 (s, 1H, NH), 7.92 (s, 1H, 6-H_Ar_). A signal for the NH proton is not seen in the spectrum. ^13^C NMR (150 MHz, CDCl_3_): δ (ppm) = 19.0 (1C, CH_3_), 20.8, 21.1, 21.4 (4C, C-3, C-4, 2 × CH_3_), 32.5, 32.8 (2C, C-2, C-5), 41.8, 44.6, 45.0 (4C, C-7, C-9, C-10, C-12), 49.7, 50.3 (2C, C-1, C-6), 55.7 (0.5C, C-11), 55.9 (1C, OCH_3_), 57.2 (0.5C, C-11), 75.8 (0.5C, C-8), 76.4 (0.5C, C-8), 110.0 (1C, C-3_Ar_), 119.1 (1C, C-6_Ar_), 122.8 (1C, C-4_Ar_), 127.0 (1C, C-5_2,4-diMePhenyl_), 127.5, 127.7 (2 × 0.5C, C-1_Ar_), 129.0 (1C, C-3_2,4-diMePhenyl_), 130.6, 130.7 (2 × 0.5C, C-5_Ar_), 131.5 (2C, C-1_2,4-diMePhenyl_, C-6_2,4-diMePhenyl_), 136.5 (1C, C-2_2,4-diMePhenyl_), 137.8 (1C, C-4_2,4-diMePhenyl_), 145.6 (1C, C-2_Ar_), 153.7 (1C, C=O). *anti,anti*-**4e**:*anti,syn*-**4e** = 1:1. Purity (HPLC): 91.3% (*t_R_* = 21.61 min).

#### 5.3.6. [(8-*anti*-11-*anti* and 8-*anti*-11-*syn*)-11-[(3,5-Bis(trifluoromethyl)benzylamino]-[4.3.3]propellan-8-yl] *N*-(2-methoxy-5-methylphenyl)carbamate (8*-anti*-4f)

NaBH(OAc)_3_ (0.3 g, 1.42 mmol) was added to a solution of ketone *anti*-**7** (0.10 g, 0.28 mmol), 3,5-bis(trifluoromethyl)benzylamine (82 mg, 0.34 mmol) and acetic acid (16 μL, 0.28 mmol) in 1,2-dichloroethane (5 mL, dried over molecular sieves 4 Å). The mixture was stirred at rt for 72 h. Then NaOH (1 M) was added (pH 8–10), the mixture was extracted with CH_2_Cl_2_ (3×) and the combined organic layers were washed with brine (1×), dried (Na_2_SO_4_), filtered, the filtrate was concentrated in vacuo and the residue was purified by fc (3 cm, cyclohexane:ethyl acetate = 7:3–1:1, 20 mL, R*_f_* = 0.14, cyclohexane:ethyl acetate = 7:3) to obtain a mixture of diastereoisomeric aminocarbamates *anti,anti-***4f** and *anti,syn*-**4f** as brown oil, yield 95 mg (59%). C_30_H_34_F_6_N_2_O_3_ (584.6). Exact mass (APCI): *m/z* = 585.2596 (calcd. 585.2546 for C_30_H_34_F_6_N_2_O_3_ [M+H]^+^). FT-IR (ATR, film): ν (cm^−1^) = 3433 (ν *N-H*), 2931 (ν *C-H* aliphatic), 1724 (ν *C=O*), 1597 (δ *N-H*). ^1^H NMR (400 MHz, CDCl_3_): δ (ppm) = 1.31–1.52 (m, 10H, 2-CH_2_, 3-CH_2_, 4-CH_2_, 5-CH_2_, 7-CH_2_(1H), 9-CH_2_(1H)), 1.70 (dd, *J* = 14.0/5.4 Hz, 2 × 0.5H, 7-H, 9-H), 1.93 (dd, *J* = 14.4/4.9 Hz, 2 × 0.5H, /-H, 9-H), 2.02–2.13 (m, 2H, 10-H, 12-H), 2.16–2.27 (m, 2H, 10-H, 12-H), 2.28 (m, 3H, CH_3_), 3.38–3.52 (m, 2 × 0.5H, 11-H), 3.83 (s, 3H, OCH_3_), 3.88 (s, 2H, NCH_2_Ph), 5.19–5.33 (m, 2 × 0.5H, 8-H), 6.73 (d, *J* = 8.3 Hz, 1H, 3-H_Ar_), 6.77 (dd, *J* = 8.8/3.8 Hz, 1H, 4-H_Ar_), 7.13 (s, 1H, NH), 7.77 (s, 1H, 4-H_3,5-diCF3Ph_), 7.85 (d, *J* = 4.9 Hz, 2H, 2-H_3,5-diCF3Ph_, 6-H_3,5-diCF3Ph_), 7.92 (s, 1H, 6-H_Ar_). A signal for the NH protons is not seen in the spectrum. ^13^C NMR (100 MHz, CDCl_3_): δ (ppm) = 21.0 (1C, CH_3_), 21.1, 21.4 (2C, C-3, C-4), 32.4, 32.8 (2C, C-2, C-5), 44.9 (4C, C-7, C-9, C-10, C-12), 49.8, 50.4 (2C, C-1, C-6), 57.5, 56.3 (2 × 0.5C, C-11), 75.9, 76.4 (2 × 0.5C, C-8), 110.0 (1C, C-3_Ar_), 118.9 (1C, C-6_Ar_), 121.3 (d, *J* = 7.53 Hz, 1C, C-4_3,5-diCF3Ph_), 122.9 (1C, C-4_Ar_), 126.2 (q, *J* = 262.3 Hz, 2C, CF_3_), 128.6 (2C, C-2_3,5-diCF3Ph_, C-6_3,5-diCF3Ph_), 130.7 (2C, C-1_Ar_, C-5_Ar_), 130.9 (2C, C-3_3,5-diCF3Ph_, C-5_3,5-diCF3Ph_), 141.4 (1C, C-1_3,5-diCF3Ph_), 145.6 (1C, C-2_Ar_), 153.6 (1C, C=O). *anti,anti*-**4f**:*anti,syn*-**4f** = 1:1. Purity (HPLC): 92.0% (*t_R_* = 22.40 min).

#### 5.3.7. [(8-*anti*-11-*anti* and 8-*anti-*11-*syn*)-11-(4-Nitrobenzylamino)[4.3.3]propellan-8-yl] *N*-(2-methoxy-5-methylphenyl)carbamate (8-*anti-*4g)

NaBH(OAc)_3_ (0.3 g, 1.42 mmol) was added to a solution of ketone *anti*-**7** (0.10 g, 0.28 mmol), 4-nitrobenzylamine hydrochloride (70 mg, 0.34 mmol) and NEt_3_ (58 μL, 0.42 mmol) in 1,2-dichloroethane (5 mL, dried over molecular sieves 4 Å). The mixture was stirred at rt for 24 h. Then acetic acid (16 μL, 0.28 mmol) was added and the mixture was stirred for additional 24 h. Then NaOH (1 M) was added (pH 8–10), the mixture was extracted with CH_2_Cl_2_ (3×) and the combined organic layers were washed with brine (1×), dried (Na_2_SO_4_), filtered, the filtrate was concentrated in vacuo and the residue was purified by fc (1 cm, cyclohexane:ethyl acetate:methanol = 5.5:3.5:1, 10 mL, R*_f_* = 0.29) to obtain a mixture of diastereoisomeric aminocarbamates *anti,anti-***4g** and *anti,syn*-**4g** as yellow oil, yield 90 mg (65%). C_28_H_35_N_3_O_5_ (493.6). MS (ESI): *m/z* = 494 [M+H]^+^. Exact mass (APCI): *m/z* = 494.2680 (494.2649 calcd. for C_28_H_36_N_3_O_5_ [M+H]^+^). FT-IR (ATR, film): ν (cm^−1^) = 3425 (ν *N-H*), 2931 (ν *C-H* aliphatic), 1724 (ν *C=O*), 1597 (δ *N-H*), 1519 (ν *N-O*), 1342 (ν *N-O*). ^1^H NMR (600 MHz, CDCl_3_): δ (ppm) = 1.36–1.55 (m, 8H, 2-CH_2_, 3-CH_2_, 4-CH_2_, 5-CH_2_), 1.68 (m, 4H, 7-H, 9-H, 10-H, 12-H), 1.90–2.11 (m, 2H, 10-H, 12-H), 2.17–2.27 (m, 2H, 7-H, 9-H), 2.28 (s, 3H, CH_3_), 3.38–3.43 (m, 0.4H, 11-H), 3.46 (tt, *J* = 7.4, 5.8 Hz, 0.6H), 3.82 (s, 3 × 0.4H, OCH_3_), 3.83 (s, 3 × 0.6H, OCH_3_), 3.87 (s, 2H, NCH_2_Ph), 5.22 (tt, *J* = 8.2, 5.3 Hz, 0.6H, 8-H), 5.28 (tt, *J* = 8.9/4.8 Hz, 0.4H, 8-H), 6.74 (dd, *J* = 8.3/1.3 Hz, 1H, 3-H_Ar_), 6.77 (dd, *J* = 8.4/2.1 Hz, 1H, 4-H_Ar_), 7.10 (s, 1H, NH), 7.51–7.56 (m, 2H, 2-H_4-NO2Ph_, 6-H_4-NO2Ph_), 7.91 (s, 1H, 6-H_Ar_), 8.15–8.19 (m, 2H, 3-H_4-NO2Ph_, 5-H_4-NO2Ph_). A signal for the NH proton is not seen in the spectrum. Signals for the OH and NH protons are not seen in the spectrum. ^13^C NMR (150 MHz, CDCl_3_): δ (ppm) = 21.0, 21.1, 21.5 (3C, C-3, C-4, CH_3_), 32.4, 32.8 (2C, C-2, C-5), 44.9, 45.0 (4C, C-7, C-9, C-10, C-12), 49.8, 50.5 (2C, C-1, C-6), 52.1 (1C, NCH_2_Ph), 55.9 (1C, OCH_3_), 56.1 (0.6C, C-11), 57.3 (0.4C, C-11), 76.0 (0.6C, C-8), 76.5 (0.4C, C-8), 110.0 (1C, C-3Ar), 118.9 (1C, C-6Ar), 122.9 (1C, C-4Ar), 123.8 (2C, C-3_4-NO2Ph_, C-5_4-NO2Ph_), 127.5 (1C, C-1_Ar_), 129.0 (2C, C-2_4-NO2Ph_, C-6_4-NO2Ph_), 130.7 (1C, C-5_Ar_), 145.6 (1C, C-2_Ar_), 147.2 (1C, C-1_4-NO2Ph_), 148.3 (1C, C-4_4-NO2Ph_), 153.5 (1C, C=O). *anti,anti*-**4g**:*anti,syn*-**4g** = 6:4. Purity (HPLC): 98.0% (*t_R_* = 20.04 min).

#### 5.3.8. [(8-*anti*-11-*anti* and 8-*anti*-11*-syn*)-11-[(4-Dimethylamino)benzylamino]-[4.3.3]propellan-8-yl] *N*-(2-methoxy-5-methylphenyl)carbamate (8-*anti*-4h)

NaBH(OAc)_3_ (0.12 g, 0.57 mmol) was added to a solution of **4v** (0.10 g, 0.28 mmol) and 4-(dimethylamino)benzaldehyde (50 mg, 0.34 mmol) in 1,2-dichloroethane (5 mL, dried over molecular sieves 4 Å). The mixture was stirred at rt for 20 h. Then NaOH (1 M) was added (pH 8–10), the mixture was extracted with CH_2_Cl_2_ (3×) and the combined organic layers were washed with brine (1×), dried (Na_2_SO_4_), filtered, the filtrate was concentrated in vacuo and the residue was purified by fc (1 cm, cyclohexane:ethyl acetate:methanol = 5.5:3.5:1, 10 mL, R*_f_* = 0.30) to obtain a mixture of diastereoisomeric aminocarbamates *anti,anti-***4h** and *anti,syn*-**4h** as yellow oil, yield 70 mg (50%). C_30_H_41_N_3_O_3_ (491.7). Exact mass (APCI): *m/z* = 492.3228 (calcd. 492.3221 for C_30_H_42_N_3_O_3_ [M+2H]^+^). FT-IR (ATR, film): ν (cm^−1^) = 3429 (ν *N-H*), 2927 (ν *C-H* aliphatic), 1724 (ν *C=O*), 1612 (δ *N-H*). ^1^H NMR (400 MHz, CDCl_3_): δ (ppm) = 1.28–1.60 (m, 8H, 2-CH_2_, 3-CH_2_, 4-CH_2_, 5-CH_2_), 1.61–1.88 (m, 3H, 7-H, 9-H, 10-H(0.5H), 12-H(0.5H)), 1.91–2.25 (m, 5H, 7-H, 9-H, 10-H(1.5H), 12-H(1.5H)), 2.28 (s, 3H, CH_3_), 2.89 (s, 3H, NCH_3_), 2.91 (s, 3H, NCH_3_), 3.39–3.54 (m, 1H, 2 × 0.5 H, 11-H), 3.71 (s, 2H, NCH_2_Ph), 3.82 (s, 3 × 0.5H, OCH_3_), 3.84 (s, 3 × 0.5H, OCH_3_), 5.17–5.31 (m, 2 × 0.5H, 8-H), 6.66–6.71 (m, 2H, 3-H_4-diMePh_, 5-H_4-diMePh_), 6.73–6.79 (m, 2H, 3-H_Ar_, 4-H_Ar_), 7.12 (s, 0.5H, NH), 7.22–7.26 (m, 2H, 2-H_4-diMePh_, 6-H_4-diMePh_), 7.29 (s, 0.5H, NH), 7.92 (s, 1H, 6-H_Ar_). A signal for the NH proton is not seen in the spectrum. ^13^C NMR (100 MHz, CDCl_3_): δ (ppm) = 20.9, 21.1, 21.4 (3C, C-3, C-4, CH_3_), 32.5, 32.8 (2C, C-2, C-5), 40.7, 40.8 (2C, N(*CH_3_*)_2_), 44.8, 45.0 (4C, C-7, C-9, C-10, C-12), 49.6, 50.3 (2C, C-1, C-6), 55.9 (0.5C, C-11), 56.3 (1C, OCH_3_), 60.5 (0.5C, C-11), 75.9 (0.5C, C-8), 76.4 (0.5, C-8), 110.0 (1C, C-3_Ar_), 112.7 (2C, C-3_4-diMePh_, C-5_4-diMePh_), 119.1 (1C, C-6_Ar_), 122.8 (1C, C-4_Ar_), 127.7 (1C, C-1_Ar_), 129.9 (2C, C-2_4-diMePh_, C-6_4-diMePh_), 130.1 (1C, C-1_4-diMePh_), 130.7 (1C, C-5_Ar_), 145.8 (1C, C-2_Ar_), 150.2 (1C, C-4_4-diMePh_), 153.7 (1C, C=O). *anti,anti*-**4h**:*anti,syn*-**4h** = 1:1. Purity (HPLC): 98.4% (*t_R_* = 17.89 min).

#### 5.3.9. [(8-*anti*-11-*anti* and 8-*anti*-11-*syn*)-11-[(Furan-2-yl-mehtyl)amino][4.3.3]propellan-8-yl] *N*-(2-methoxy-5-methylphenyl)carbamate (8-*anti*-4i)

NaBH(OAc)_3_ (0.36 g, 1.67 mmol) was added to a solution of ketone *anti*-**7** (0.15 g, 0.42 mmol), furfurylamine (0.15 g, 0.55 mmol) and acetic acid (24 μL, 0.42 mmol) in 1,2-dichloroethane (10 mL, dried over molecular sieves 4 Å). The mixture was stirred at rt for 48 h. Then NaOH (1 M) was added (pH 8–10), the mixture was extracted with CH_2_Cl_2_ (3×) and the combined organic layers were washed with brine (1×), dried (Na_2_SO_4_), filtered, the filtrate was concentrated in vacuo and the residue was purified by fc (1 cm, cyclohexane:ethyl acetate:methanol = 5.5:3.5:1, 10 mL, R*_f_* = 0.59) to obtain a mixture of diastereoisomeric aminocarbamates *anti,anti-***4i** and *anti,syn*-**4i** as dark yellow oil, yield 0.15 mg (79%). C_26_H_34_N_2_O_4_ (438.6). MS (ESI): *m/z* = 439 [M+H]^+^. Exact mass (APCI): *m/z* = 439.2605 (calcd. 439.2591 for C_26_H_35_N_2_O_4_ [M+H]^+^). FT-IR (ATR, film): ν (cm^−1^) = 3425 (ν *N-H*), 2931 (ν *C-H* aliphatic), 1724 (ν *C=O*), 1597 (δ *N-H*). ^1^H NMR (400 MHz, CDCl_3_): δ (ppm) = 1.29–1.54 (m, 8H, 2-CH_2_, 3-CH_2_, 4-CH_2_, 5-CH_2_), 1.61 (dd, *J* = 13.3/6.8 Hz, 2 × 0.5H, 10-H, 12-H), 1.67–1.73 (m, 2H, 7-H, 9-H), 1.90 (dd, *J* = 14.4/4.9 Hz, 2 × 0.5H, 10-H, 12-H), 1.96–2.07 (m, 2H, 7-H, 9-H), 2.13–2.26 (m, 2H, 7-H, 9-H), 2.28 (m, 3H, CH_3_), 3.35–3.49 (m, 2 × 0.5H, 11-H), 3.78 (s, 2H, NCH_2_Furyl), 3.84 (s, 3H, OCH_3_), 5.16–5.32 (m, 2 × 0.5H, 8-H), 6.18–6.24 (m, 1H, 3-H_Furan_), 6.29–6.33 (m, 1H, 4-H_Furan_), 6.71–6.79 (m, 2H, 3-H_Ar_, 4-H_Ar_), 7.13 (s, 0.5H, NH), 7.20 (s, 0.5H, NH), 7.32–7.38 (m, 1H, 5-H_Furan_), 7.92 (s, 1H, 6-H_Ar_). A signal for the NH proton is not seen in the spectrum. Signals for the OH and NH protons are not seen in the spectrum. ^13^C NMR (100 MHz, CDCl_3_): δ (ppm) = 20.9, 21.1 (2C, C-3, C-4), 21.5 (1C, CH_3_), 32.4, 32.8 (2C, C-2, C-5), 44.5, 44.9, 45.1, 45.2 (5C, C-7, C-9, C-10, C-12, NCH_2_Furan), 49.7, 50.4 (2C, C-1, C-6), 55.5 (0.5C, C-11), 55.9 (1C, OCH_3_), 56.8 (0.5C, C-11), 76.0 (0.5C, C-8), 76.4 (0.5C, C-8), 107.2, 107.6 (1C, C-3_Furan_), 109.9 (1C, C-3_Ar_), 110.4, 110.6 (1C, C-4_Furan_), 118.9 (1C, C-6_Ar_), 122.9(1C, C-4_Ar_), 127.5 (1C, C-1_Ar_), 130.6 (1C, C-5_Ar_), 142.0, 142.3 (1C, C-5_Furan_), 145.6 (1C, C-2_Ar_), 146.0 (1C, C-2_Furan_), 153.6 (1C, C=O). *anti,anti*-**4i**:*anti,syn*-**4i** = 6:4. Purity (HPLC): 91.4% (*t_R_* = 19.33 min).

#### 5.3.10. [(8-syn-11-syn and 8-*syn*-11-*anti*)-11-[(Furan-2-yl-mehtyl)amino][4.3.3]propellan-8-yl] *N*-(2-methoxy-5-methylphenyl)carbamate (8-*syn*-4i)

NaBH(OAc)_3_ (0.3 g, 1.42 mmol) was added to a solution of ketone *syn*-**7** (0.10 g, 0.28 mmol), furfurylamine (29 mg, 0.34 mmol) and acetic acid (16 μL, 0.28 mmol) in 1,2-dichloroethane (5 mL, dried over molecular sieves 4 Å). The mixture was stirred at rt for 5 days. Then NaOH (1 M) was added (pH 8–10), the mixture was extracted with CH_2_Cl_2_ (3×) and the combined organic layers were washed with brine (1×), dried (Na_2_SO_4_), filtered, the filtrate was concentrated in vacuo and the residue was purified by fc (1 cm, cyclohexane:ethyl acetate:methanol = 5.5:3.5:1 to 1:1, 10 mL, R*_f_* = 0.18) to obtain a mixture of diastereoisomeric aminocarbamates *syn,syn-***4i** and *anti,syn*-**4i** as yellow oil, yield 84 mg (66%). C_26_H_34_N_2_O_4_ (438.6). MS (ESI): *m/z* = 439 [M+H]^+^. Exact mass (APCI): *m/z* = 439.2631 (calcd. 439.2591 for C_26_H_35_N_2_O_4_ [M+H]^+^). FT-IR (ATR, film): ν (cm^−1^) = 3425 (ν *N-H*), 2931 (ν *C-H* aliphatic), 1720 (ν *C=O*), 1597 (δ *N-H*). ^1^H NMR (600 MHz, CDCl_3_): δ (ppm) = 1.33–1.58 (m, 8H, 2-CH_2_, 3-CH_2_, 4-CH_2_, 5-CH_2_), 1.70 (dd, *J* = 13.5/7.3 Hz, 2 × 0.5H, 7-H, 9-H), 1.80–1.90 (m, 3H), 1.97–2.07 (m, 3H), 2.29 (s, 3H, CH_3_), 2.29–2.35 (m, 2 × 0.5H, 7-H, 9-H), 3.29–3.35 (m, 0.5H, 11-H), 3.35–3.41 (m, 0.5H, 11-H), 3.80–3.85 (m, 5H, OCH_3_, NCH_2_Furyl), 5.20 (tt, *J* = 8.3/4.7 Hz, 0.5H, 8-H), 5.31 (tt, *J* = 8.4/4.2 Hz, 0.5H, 8-H), 6.30 (dd, *J* = 6.2/3.5 Hz, 1H, 3-H_Furan_), 6.33 (d, *J* = 2.8/2.0 Hz, 1H, 4-H_Furan_), 6.73 (dd, *J* = 8.2/3.2 Hz, 1H, 3-H_Ar_), 6.77 (dd, *J* = 7.1/5.0 Hz, 1H, 4-H_Ar_), 7.12 (s, 1H, NH), 7.38 (ddd, *J* = 4.0/1.8/0.9 Hz, 1H, 5-H_Furan_), 7.92 (s, 1H, 6-H_Ar_). A signal for the NH proton is not seen in the spectrum. ^13^C NMR (150 MHz, CDCl_3_): δ (ppm) = 21.1, 21.2, 21.3 (3C, C-3, C-4, CH_3_), 32.1, 33.1 (2C, C-2, C-5), 44.3, 44.6, 45.0, 45.0 (5C, C-7, C-9, C-10, C-12, NCH_2_Furyl), 49.5, 50.0 (2C, C-1, C-6), 55.7 (0.5C, C-11), 55.9 (1C, OCH_3_), 56.4 (0.5C, C-11), 76.0 (0.5C, C-8), 76.1 (0.5C, C-8), 108.5 (1C, C-3_Furan_), 110.0 (1C, C-3_Ar_), 110.6 (1C, C-4_Furan_), 119.0 (1C, C-2_Furan_), 122.8 (C-4_Ar_), 127.5 (C-1_Ar_), 130.7 (1C, C-5_Ar_), 142.4 (1C, C-5_Furan_), 145.6 (1C, C-2_Ar_), 153.5 (1C, C=O). *syn,syn*-**4i**:*syn,anti*-**4i** = 1:1. Purity (HPLC): 97.6% (*t_R_* = 19.35 min).

#### 5.3.11. [(8-anti-11-anti and 8-anti-11-*syn*)-11-[2-(Indol-3-yl)ethylamino][4.3.3]propellan-8-yl] *N*-(2-methoxy-5-methylphenyl)carbamate (8-*anti*-4k)

NaBH(OAc)_3_ (62 mg, 0.29 mmol) was added to a solution of ketone *anti*-**7** (35 mg, 0.10 mmol), tryptamine (20 mg, 0.12 mmol) and acetic acid (6 μL, 0.10 mmol) in 1,2-dichloroethane (5 mL, dried over molecular sieves 4 Å). The mixture was stirred at rt for 96 h. Then NaOH (1 M) was added (pH 8–10), the mixture was extracted with CH_2_Cl_2_ (3×) and the combined organic layers were washed with brine (1×), dried (Na_2_SO_4_), filtered, the filtrate was concentrated in vacuo and the residue was purified by fc (1 cm, cyclohexane:ethyl acetate:methanol = 5.5:3.5:1, 10 mL, R*_f_* = 0.18,) to obtain a mixture of diastereoisomeric aminocarbamates *anti,anti-***4k** and *anti,syn*-**4k** as brown oil, yield 20 mg (41%). C_31_H_39_N_3_O_3_ (501.7). Exact mass (APCI): *m/z* = 502.3190 (calcd. 502.3064 for C_31_H_40_N_3_O_3_ [M+H]^+^). FT-IR (ATR, film): ν (cm^−1^) = 3429 (ν *N-H*), 2927 (ν *C-H* aliphatic), 1716 (ν *C=O*), 1597 (δ *N-H*). ^1^H NMR (600 MHz, CDCl_3_): δ (ppm) = 1.25–1.49 (m, 8H, 2-CH_2_, 3-CH_2_, 4-CH_2_, 5-CH_2_), 1.56 (dd, *J* = 14.5/5.0 Hz, 2 × 0.5H, 7-H, 9-H), 1.98–2.18 (m, 7H, 7-CH_2_(1H), 9-CH_2_(1H), 10-CH_2_, 12-CH_2_), 2.27 (s, 3H, CH_3_), 3.17–3.25 (m, 2H, N*CH_2_*CH_2_Indole), 3.39–3.46 (m, 2H, NCH_2_*CH_2_*Indole), 3.59–3.66 (m, 0.6H, 11-H), 3.70 (tt, *J* = 9.4/7.3 Hz, 0.4H, 11-H), 3.77 (s, 3 × 0.4H, OCH_3_), 3.78 (s, 3 × 0.6H, OCH_3_), 5.15 (tt, *J* = 7.9/5.0 Hz, 0.6H, 8-H), 5.23 (tt, *J* = 8.1/4.9 Hz, 0.4H, 8-H), 6.71 (d, *J* = 8.2 Hz, 1H, 3-H_Ar_), 6.74–6.78 (m, 1H, 4-H_Ar_), 7.04–7.07 (m, 2H, 5-H_Indole_, 6-H_Indole_), 7.12–7.17 (m, 1H, 2-H_Indole_), 7.33 (ddd, *J* = 8.1/2.1/1.0 Hz, 1H, 7-H_Indole_), 7.59 (s, 1H, NH_carbamate_) 7.63 (ddd, *J* = 8.2/2.5/1.0 Hz, 1H, 4-H_Indole_), 7.86–7.89 (s, 1H, 6-H_Ar_), 8.41 (s, 1H, NH_Indole_). A signal for the NH proton is not seen in the spectrum. ^13^C NMR (150 MHz, CDCl_3_): δ (ppm) = 20.4 (1C, CH_3_), 20.7, 21.1 (2C, C-3, C-4), 22.6 (1C, NCH_2_*CH_2_*Indole), 32.3, 32.8 (2C, C-2, C-5), 40.9, 41.6 (2C, C-10, C-12), 44.4, 45.1 (2C, C-7, C-9), 47.8 (1C, N*CH_2_*CH_2_Indole), 49.4, 49.8 (2C, C-1, C-6), 55.9 (1C, OCH_3_), 56.7 (0.4C, C-11), 58.0 (0.6C, C-11), 75.5 (0.4C, C-8), 76.2 (0.6C, C-8), 110.0 (1C, C-3_Ar_), 110.5, 111.6 (2 × 0.5C, C-7_Indole_), 118.7 (1C, C-4_Indole_), 119.7 (1C, C-6_A_r), 119.8 (1C, C-5_Indole_), 122.4 (1C, C-2_Indole_), 123.1 (2C, C-4_Ar_, C-6_Indole_), 126.9 (1C, C-3a_Indole_), 127.7 (1C, C-1_Ar_), 130.6 (1C, C-5_Ar_), 136.5 (1C, C-7a_Indole_), 146.3 (1C, C-2_Ar_), 153.8 (1C, C=O). *anti,anti*-**4k**:*anti,syn*-**4k** = 6:4. Purity (HPLC): 66.3%, light sensitive (*t_R_* = 20.8 min).

#### 5.3.12. [(8-*syn*-11-*syn* and 8-*syn*-11-*anti*)-11-[(2-(Indol-3-yl)ethylamino][4.3.3]propellan-8-yl] *N-*(2-methoxy-5-methylphenyl)carbamate (8-*syn*-**4k**)

NaBH(OAc)_3_ (0.3 g, 1.42 mmol) was added to a solution of ketone *syn*-**7** (0.10 g, 0.28 mmol), tryptamine (52 mg, 0.34 mmol) and acetic acid (16 μL, 0.28 mmol) in 1,2-dichloroethane (5 mL, dried over molecular sieves 4 Å). The mixture was stirred at rt for 5 days. Then NaOH (1 M) was added (pH 8–10), the mixture was extracted with CH_2_Cl_2_ (3×) and the combined organic layers were washed with brine (1×), dried (Na_2_SO_4_), filtered, the filtrate was concentrated in vacuo and the residue was purified by fc (1 cm, cyclohexane:ethyl acetate:methanol = 5.5:3.5:1, 10 mL, R*_f_* = 0.15,) to obtain a mixture of diastereoisomeric aminocarbamates *syn,syn-***4k** and *syn,anti*-**4k** as brown oil, yield 60 mg (41%). C_31_H_39_N_3_O_3_ (501.7). Exact mass (APCI): *m/z* = 502.3078 (calcd. 502.3064 for C_31_H_40_N_3_O_3_ [M+H]^+^). FT-IR (ATR, film): ν (cm^−1^) = 3421 (ν *N-H*), 2931 (ν *C-H* aliphatic), 1716 (ν *C=O*), 1597 (δ *N-H*). ^1^H NMR (400 MHz, CDCl_3_): δ (ppm) = 1.26–1.58 (m, 8H, 2-CH_2_, 3-CH_2_, 4-CH_2_, 5-CH_2_), 1.70–1.97 (m, 6H, 7-CH_2_(1H), 9-CH_2_(1H), 10-CH_2_, 12-CH_2_), 2.28 (s, 3H, CH_3_), 2.30–2.43 (m, 2H, 7-H, 9-H), 3.11–3.21 (m, 2H, N*CH_2_*CH_2_Indolyl), 3.31–3.39 (m, 2H, NCH_2_*CH_2_*Indole), 3.57–3.66 (m, 1H, 11-H), 3.83 (m, 3H, OCH_3_), 5.02–5.10 (m, 0.4H, 8-H), 5.22 (tt, *J* = 8.1/4.0 Hz, 0.6H, 8-H), 6.73 (d, *J* = 8.3 Hz, 1H, 3-H_Ar_), 6.78 (dd, *J* = 8.3/2.1 Hz, 1H, 4-H_Ar_), 7.02–7.17 (m, 4H, 2-H_Indole_,5-H_Indole_, 6-H_Indole_, NH_Carbamate_), 7.35 (d, *J* =, 1H, 7-H_Indole_), 7.61 (d, *J* = 7.9 Hz, 1H, 4-H_Indole_), 7.89 (s, 1H, 6-H_Ar_), 8.74 (s, 1H, NH_Indole_). A signal for the NH proton is not seen in the spectrum. ^13^C NMR (100 MHz, CDCl_3_): δ (ppm) = 20.7, 21.1 (3C, C-3, C-4, CH_3_), 22.9 (1C, NCH_2_*CH_2_*Indole), 31.9, 32.9 (2C, C-2, C-5), 41.5, 41.6 (2C, C-10, C-12), 44.8, 45.0 (2C, C-7, C-9), 47.8 (1C, N*CH_2_*CH_2_Indole), 49.1, 49.8 (2C, C-1, C-6), 55.9 (1C, OCH_3_), 56.7 (0.4C, C-11), 57.6 (0.6C, C-11), 75.6 (0.4C, C-8), 76.1 (0.6C, C-8), 110.0 (1C, C-3_Ar_), 110.8, 111.6 (2 × 0.5C, C-7_Indole_), 118.7 (1C, C-4_Indole_), 119.1 (1C, C-6_A_r), 119.6 (1C, C-5_Indole_), 122.3 (1C, C-2_Indole_), 123.1 1 (2C, C-4_Ar_, C-6_Indole_), 127.0 (1C, C-3a_Indole_), 127.5 (1C, C-1_Ar_), 130.7 (1C, C-5_Ar_), 136.5 (1C, C-7a_Indole_), 145.7 (1C, C-2_Ar_), 153.4 (1C, C=O). *syn,syn*-**4k**:*syn,anti*-**4k** = 4:6. Purity (HPLC): 97.8% (*t_R_* = 20.97 min).

#### 5.3.13. [(8-*anti*-11-*anti* and 8-*anti*-11-*syn*)-11-(Phenylamino)[4.3.3]propellan-8-yl] *N-*(2-methoxy-5-methylphenyl)carbamate (8-*anti*-**4l**)

NaBH(OAc)_3_ (0.30 g, 1.42 mmol) was added to a solution of ketone *anti*-**7** (0.10 g, 0.28 mmol), aniline (31.3 mg, 0.34 mmol) and acetic acid (16 μL, 0.28 mmol) in 1,2-dichloroethane (5 mL, dried over molecular sieves 4 Å). The mixture was stirred at rt for 48 h. Then NaOH (1 M) was added (pH 8–10), the mixture was extracted with CH_2_Cl_2_ (3×) and the combined organic layers were washed with brine (1×), dried (Na_2_SO_4_), filtered, the filtrate was concentrated in vacuo and the residue was purified by fc (3 cm, cyclohexane:ethyl acetate = 8:2, 5 mL, R*_f_* = 0.82, cyclohexane:ethyl acetate = 7:3) to obtain a mixture of diastereoisomeric aminocarbamates *anti,anti-***4l** and *anti,syn*-**4l** as pale yellow oil, yield 84 mg (69%). C_27_H_34_N_2_O_3_ (434.6). Exact mass (APCI): *m/z* = 435.2599 (calcd.435.2642 for C_27_H_35_N_2_O_3_ [M+H]^+^). FT-IR (ATR, film): ν (cm^−1^) = 3394 (ν *N-H*), 2931 (ν *C-H* aliphatic), 1720 (ν *C=O*), 1600 (δ *N-H*). ^1^H NMR (400 MHz, CDCl_3_): δ (ppm) = 1.32–1.54 (m, 8H, 2-CH_2_, 3-CH_2_, 4-CH_2_, 5-CH_2_), 1.66 (dd, *J* = 13.6/6.0 Hz, 2 × 0.5H, 10-H, 12-H), 1.72–1.82 (m, 2H, 7-H, 9-H), 1.90 (dd, J = 14.4/4.9 Hz, 2 × 0.5H, 10-H, 12-H), 2.21–2.35 (m, 7H, CH_3_, 7-H, 9-H, 10-H, 12-H), 3.85 (s, 3H, OCH_3_), 3.99–4.07 (m, 0.5H, 11-H), 4.07–4.12 (m, 0.5H, 11-H), 5.29 (tt, *J* = 8.5/4.8 Hz, 1H, 8-H), 6.61 (dd, *J* = 8.3/3.2 Hz, 2H, 2-H_Ph_, 6-H_Ph_), 6.70 (td, J = 7.2/1.3 Hz, 1H, 4H_Ph_), 6.75 (d, *J* = 8.3 Hz, 1H, 3-H_Ar_), 6.78 (d, *J* = 8.2 Hz, 1H, 4-H_Ar_), 7.13–7.22 (m, 3H, 3-H_Ph_, 5-_Ph_, NH), 7.94 (s, 1H, 6-H_Ar_). ^13^C NMR (100 MHz, CDCl_3_): δ (ppm) = 20.9 (2 × 0.5C, C-3, C-4), 21.1 (1C, CH_3_), 21.4 (2 × 0.5C, C-3, C-4), 32.3, 32.7 (2C, C-2, C-5), 44.6, 44.9 (4C, C7, C-9, C-10, C-12), 49.8, 50.4 (2C, C-1, C-6), 55.9 (1C, OCH_3_), 60.5 (1C, C-11), 75.9, 76.3 (2 × 0.5C, C-8), 110.0 (1C, C-3_Ar_), 119.0 (1C, C-4_Ph_), 119.1 (1C, C-6_Ar_), 123.0 (1C, C-4_Ar_), 129.5 (4C, C-2_Ph_, C-3_Ph_, C-5_Ph_, C-6_Ph_), 130.7 (2C, C-1_Ar_, C-5_Ar_), 145.6 (1C, C-2_Ar_), 145.8 (1C, C-1_Ph_), 153.6 (1C, C=O). *anti,anti*-**4l**:*anti,syn*-**4l** = 1:1. Purity (HPLC): 98.2% (*t_R_* = 20.55 min).

#### 5.3.14. [(8-*anti*-11-*anti* and 8-*anti*-11-*syn*)-11-(4-Methoxyphenylamino)[4.3.3]propellan-8-yl] *N-*(2-methoxy-5-methylphenyl)carbamate (8-*anti*-**4m**)

NaBH(OAc)_3_ (0.47 g, 2.24 mmol) was added to a solution of ketone *anti*-**7** (0.20 g, 0.56 mmol), 4-methoxyphenylamine (87 mg, 0.67 mmol) and acetic acid (32 μL, 0.56 mmol) in 1,2-dichloroethane (10 mL, dried over molecular sieves 4 Å). The mixture was stirred at rt for 72 h. Then NaOH (1 M) was added (pH 8–10), the mixture was extracted with CH_2_Cl_2_ (3×) and the combined organic layers were washed with brine (1×), dried (Na_2_SO_4_), filtered, the filtrate was concentrated in vacuo and the residue was purified by fc (3 cm, cyclohexane:Et_2_O = 8:2 to 1:1, 10 mL, R*_f_* = 0.56, cyclohexane:ethylacetate = 7:3) to obtain a mixture of diastereoisomeric aminocarbamates *anti,anti-***4m** and *anti,syn*-**4m** as brown oil, yield 0.19 g (74%). C_28_H_36_N_2_O_4_ (464.6). Exact mass (APCI): *m/z* = 465.2721 (calcd.465.2748 for C_28_H_35_Cl_2_N_2_O_4_ [M+H]^+^). FT-IR (ATR, film): ν (cm^−1^) = 3390 (ν *N-H*), 2931 (ν *C-H* aliphatic), 1724 (ν *C=O*), 1597 (δ *N-H*). ^1^H NMR (400 MHz, CDCl_3_): δ (ppm) = 1.30–1.53 (m, 8H, 2-CH_2_, 3-CH_2_, 4-CH_2_, 5-CH_2_), 1.64 (dd, *J* = 13.6/6.1 Hz, 2 × 0.5H, 7-H, 9-H), 1.70–1.80 (m, 2H, 10-H, 12-H), 1.90 (dd, *J* = 14.4/4.8 Hz, 2 × 0.5H, 7-H, 9-H), 2.17–2.32 (m, 7H, CH_3_, 7-H, 9-H, 10-H, 12-H), 3.74 (s, 3H, OCH_3 4-OMephenyl_), 3.85 (s, 3H, OCH_3 Ar_), 3.94–4.08 (m, 2 × 0.5H, 11-H), 5.23–5.33 (m, 2 × 0.5H, 8-H), 6.56–6.62 (m, 2H, 3-H_Ar_, 4-H_Ar_), 6.73–6.80 (m, 4H, 2-H_4-OMephenyl_, 3-H_4-OMephenyl_,, 5-H_4-OMephenyl_, 6-H_4-OMephenyl_), 7.18 (s, 1H, NH), 7.93 (s, 1H, 6-H_Ar_). A signal for the NH proton is not seen in the spectrum. ^13^C NMR (100 MHz, CDCl_3_): δ (ppm) = 21.2, 21.2, 21.6 (3C, C-3, C-4, CH_3_), 32.3, 32.7 (2C, C-2, C-5), 44.7, 44.8, 45.5, 45.9 (4C, C-7, C-9, C-10, C-12), 50.0, 50.6 (2C, C-1, C-6), 52.9, 53.0 (2 × 0.5C, C-11), 55.9, 56.0 (2C, 2 × OCH_3_), 76.0, 76.3 (2 × 0.5C, C-8), 110.0 (1C, C-3_Ar_), 115.0 (4C, C-2_4-OMephenyl_, C-3_4-OMephenyl_, C-5_4-OMephenyl_, C-6_4-OMephenyl_), 118.9 (1C, C-6_Ar_), 122.9 (1C, C-4_Ar_), 130.7, 130.7 (2C, C-1_Ar_, C-5_Ar_), 145.6 (1C, C-2_Ar_), 146.0 (1C, C-1_4-OMephenyl_), 152.3 (1C, C-4_4-OMephenyl_), 153.6 (1C, C=O). *anti,anti*-**4m**:*anti,syn*-**4m** = 1:1. Purity (HPLC): 96.4% (*t_R_* = 20.53 min).

#### 5.3.15. [(8-*anti*-11-*anti* and 8-*anti*-11-*syn*)-11-(3-Chloro-4-methoxyphenylamino)-[4.3.3]propellan-8-yl] *N-*(2-methoxy-5-methylphenyl)carbamate (8-*anti*-**4n**)

NaBH(OAc)_3_ (0.3 g, 1.42 mmol) was added to a solution of ketone *anti*-**7** (0.10 g, 0.28 mmol), 3-chloro-4-methoxyaniline (53 mg, 0.34 mmol) and acetic acid (16 μL, 0.28 mmol) in 1,2-dichloroethane (5 mL, dried over molecular sieves 4 Å). The mixture was stirred at rt for 72 h. Then NaOH (1 M) was added (pH 8–10), the mixture was extracted with CH_2_Cl_2_ (3×) and the combined organic layers were washed with brine (1×), dried (Na_2_SO_4_), filtered, the filtrate was concentrated in vacuo and the residue was purified by fc (3 cm, cyclohexane:Et_2_O = 7:3 to 1:1, 10 mL, R*_f_* = 0.51, cyclohexane:ethylacetate = 7:3) to obtain a mixture of diastereoisomeric aminocarbamates *anti,anti-***4n** and *anti,syn*-**4n** as brown oil, yield 0.13 g (63%). C_28_H_35_ClN_2_O_4_ (499.0). Exact mass (APCI): *m/z* = 499.2403 (calcd.499.2358 for C_28_H_36_ClN_2_O_4_ [M+H]^+^). FT-IR (ATR, film): ν (cm^−1^) = 3390 (ν *N-H*), 2931 (ν *C-H* aliphatic), 1720 (ν *C=O*), 1597 (δ *N-H*). ^1^H NMR (400 MHz, CDCl_3_): δ (ppm) = 1.33–1.50 (m, 8H, 2-CH_2_, 3-CH_2_, 4-CH_2_, 5-CH_2_), 1.62–1.92 (m, 4H, 7-H, 9-H, 10-H, 12-H), 2.19–2.32 (m, 7H, 7-H, 9-H, 10-H, 12-H, CH_3_), 3.82 (s, 3H, OCH_3_), 3.85 (s, 3H, OCH_3_), 3.90–4.04 (m, 2 × 0.5H, 11-H), 5.23–5.33 (m,2 × 0.5H, 8-H), 6.49–6.58 (m, 1H, 2-H_4-Cl-5-MeOPhen_), 6.67–6.83 (m, 4H, 5-H_4-Cl-5-MeOPhen_, 6-H_4-Cl-5-MeOPhen_, 3-H_Ar_, 4-H_Ar_), 7.18 (s, 1H, NH_carbamate_), 7.93 (s, 1H, 6-H_Ar_). A signal for the NH proton is not seen in the spectrum. ^13^C NMR (100 MHz, CDCl_3_): δ (ppm) = 20.8 (1C, CH_3_), 21.1 (2C, C-3, C-4), 32.3, 32.7 (2C, C-2, C-5), 44.5, 44.9 (4C, C-7, C-9, C-10, C-12), 49.8, 50.3 (2C, C-1, C-6), 55.9 (2C, 2 × OCH_3_), 56.9 (2 × 0.5C, C-11), 75.7, 76.3 (2 × 0.5C, C-8), 110.0 (1C, C-3_Ar_), 113.8 (1C, C-6_4-Cl-5-MeOPhen_), 114.9 (1C, C-2_4-Cl-5-MeOPhen_), 118.9 (1C, C-6_Ar_), 119.2 (1C, C-5_4-Cl-5-MeOPhen_), 123.0 (1C, C-4_Ar_), 123.6 (1C, C-3_4-Cl-5-MeOPhen_), 127.5 (1C, C-1_Ar_), 130.7 (1C, C-5_Ar_), 138.2 (1C, C-1_4-Cl-5-MeOPhen_), 145.7, 147.7, (2C, C-2_Ar_, C-4_4-Cl-5-MeOPhen_) 153.6 (1C, C=O). *anti,anti*-**4n**:*anti,syn*-**4n** = 1:1. Purity (HPLC): 95.2% (*t_R_* = 21.27 min).

#### 5.3.16. [(8-*anti*-11-*anti* and 8-*anti*-11-*syn*)-11-(4-Aminophenylamino)[4.3.3]propellan-8-yl] *N-*(2-methoxy-5-methylphenyl)carbamate (8-*anti*-**4o**)

NaBH(OAc)_3_ (0.47 g, 2.24 mmol) was added to a solution of ketone *anti*-**7** (0.20 g, 0.56 mmol), *p*-phenylenediamine (97 mg, 0.90 mmol) and acetic acid (32 μL, 0.56 mmol) in 1,2-dichloroethane (10 mL, dried over molecular sieves 4 Å). The mixture was stirred at rt for 72 h. Then NaOH (1 M) was added (pH 8–10), the mixture was extracted with CH_2_Cl_2_ (3×) and the combined organic layers were washed with brine (1×), dried (Na_2_SO_4_), filtered, the filtrate was concentrated in vacuo and the residue was purified by fc (3 cm, ethyl acetate:methanol = 9.5:0.5–8:2, 10 mL, R*_f_* = 0.11, cyclohexane:ethylacetate = 7:3) to obtain a mixture of diastereoisomeric aminocarbamates *anti,anti-***4o** and *anti,syn*-**4o** as violet oil, yield 0.19 g (76%). C_27_H_35_N_2_O_3_ (449.6). Exact mass (APCI): *m/z* = 450.2720 (calcd. 450.2751 for C_27_H_36_N_3_O_3_ [M+H]^+^). FT-IR (ATR, film): ν (cm^−1^) = 3417 (ν *N-H_2_*), 3290 (ν *N-H*), 2931 (ν *C-H* aliphatic), 1716 (ν *C=O*), 1604 (δ *N-H*). ^1^H NMR (400 MHz, CDCl_3_): δ (ppm) = 1.37–1.51 (m, 8H, 2-CH_2_, 3-CH_2_, 4-CH_2_, 5-CH_2_), 1.63–92 (m, 4H, 7-H, 9-H, 10-H, 12-H), 2.17–2.28 (m, 4H, 7-H, 9-H, 10-H, 12-H), 2.29 (s, 3H, CH_3_), 3.85 (s, 3H, OCH_3_), 3.92–4.05 (m, 2 × 0.5H, 11-H), 5.22–5.32 (m, 2 × 0.5H, 8-H), 6.53–6.66 (m, 4H, 2-H_4-aminophenyl_, 3-H_4-aminophenyl_, 5-H_4-aminophenyl_, 6-H_4-aminophenyl_), 6.74 (dd, *J* = 8.3/1.4 Hz, 1H, 4-H_Ar_), 6.78 (d, *J* = 8.5 Hz, 1H, 3-H_Ar_), 7.14 (s, 0.5H, NH), 7.21 (s, 0.5H, NH), 7.28 (s, 1H, NH_carbamate_), 7.93 (s, 1H, 6-H_Ar_). A signal for the NH protons is not seen in the spectrum. ^13^C NMR (100 MHz, CDCl_3_): δ (ppm) = 21.1, 21.4 (3C, C-3, C-4, CH_3_), 32.0, 32.7 (2C, C-2, C-5), 45.0, 45.4 (4C, C-7, C-9, C-10, C-12), 50.0 (2C, C-1, C-6), 55.9 (2C, C-11, OCH_3_), 77.4 (2 × 0.5C, C-8), 110.1(1C, C-3_Ar_), 119.0 (1C, C-6_Ar_), 123.1 (5C, C-2_4-aminophenyl_, C-3_4-aminophenyl_, C-5_4-aminophenyl_, C-6_4-aminophenyl_, C-4_Ar_), 130.6, 131.3 (3C, C-1_4-aminophenyl_, C-4_4-aminophenyl_,C-5_Ar_), 145.7 (1C, C-2_Ar_), 151.8 (1C, C=O). *anti,anti*-**4o**:*anti,syn*-**4o** = 1:1. Purity (HPLC): 97.8% (*t_R_* = 19.33 min).

#### 5.3.17. [(8-*anti*-11-*anti* and 8-*anti*-11-*syn*)-11-(Phenethylamino)[4.3.3]propellan-8-yl] *N-*(2-methoxy-5-methylphenyl)carbamate (8-*anti*-**4p**)

NaBH(OAc)_3_ (0.67 g, 3.20 mmol) was added to a solution of ketone *anti*-**7** (0.23 g, 0.64 mmol), 2-phenylethanamine (0.1 g, 0.85 mmol) and acetic acid (38 μL, 0.64 mmol) in 1,2-dichloroethane (10 mL, dried over molecular sieves 4 Å). The mixture was stirred at rt for 6 days. Then NaOH (1 M) was added (pH 8–10), the mixture was extracted with CH_2_Cl_2_ (3×) and the combined organic layers were washed with brine (1×), dried (Na_2_SO_4_), filtered, the filtrate was concentrated in vacuo and the residue was purified by fc (3 cm, methanol:ethyl acetate = 9:1, 5 mL, R*_f_* = 0.27, cyclohexane:ethyl acetate = 7:3) to obtain a mixture of diastereoisomeric aminocarbamates *anti,anti-***4p** and *anti,syn*-**4p** as pale yellow oil, yield 0.23 g (76%). C_29_H_38_N_2_O_3_ (462.6). Exact mass (APCI): *m/z* = 463.2952 (calcd.463.2955 for C_29_H_39_N_2_O_3_ [M+H]^+^). FT-IR (ATR, film): ν (cm^−1^) = 3379 (ν*N-H*), 2931 (ν *C-H* aliphatic), 1732 (ν*C=O*), 1597 (δ *N-H*). ^1^H NMR (400 MHz, CDCl_3_): δ (ppm) = 1.27–1.45 (m, 6H, 2-CH_2_(1H), 3-CH_2_, 4-CH_2_, 5-CH_2_(1H)), 1.51 (m, 2 × 0.5H, 2-H, 5-H), 1.60 (m, 2 × 0.5H, 2-H, 5-H), 1.67 (dd, *J* = 14.4/5.1 Hz, 2 × 0.5H, 7-H, 9-H), 1.98–2.08 (m, 4H, 10-CH_2_, 12-CH_2_), 2.13–2.24 (m, 3H, 7-CH_2_(1.5H), 9-CH_2_(1.5H)), 2.29 (s, 3H, CH_3_), 3.00–3.21 (m, 4H, N*CH_2_CH_2_*Ph), 3.50–3.60 (m, 0.5H, 11-H), 3.64 (q, *J* = 9.6/9.0 Hz, 0.5H, 11-H), 3.81 (s, 3 × 0.5H, OCH_3_), 3.83 (s, 3 × 0.5H, OCH_3_), 5.20 (tt, *J* = 7.7/5.1 Hz, 0.5H, 8-H), 5.28 (tt, *J* = 8.4/4.1 Hz, 0.5H, 8-H), 6.73 (dd, *J* = 8.2/2.1 Hz, 1H, 3-H_Ar_), 6.77 (dd, *J* = 8.2/2.1 Hz, 1H, 4-H_Ar_), 7.13 (s, 1H, NH), 7.19–7.42 (m, 5H, Ph), 7.91 (s, 1H, 6-H_Ar_). A signal for the NH proton is not seen in the spectrum. ^13^C NMR (100 MHz, CDCl_3_): δ (ppm) = 20.6 (1C, CH_3_), 21.0, 21.1 (2C, C-3, C-4), 32.5, 32.9 (2C, C-2, C5), 34.4 (1C, NCH_2_*CH_2_*Ph), 44.6 (2C, C-10, C-12), 45.2 (2C, C-7, C-9), 49.5 (1C, N*CH_2_*CH_2_Ph), 50.0 (2C, C-1, C-6), 55.8, 55.9 (1C, OCH_3_), 56.6, 57.9 (2 × 0.5 C, C-11), 76.2 (1C, C-8), 109.9 (1C, C-3_Ar_), 119.3 (1C, C-6_Ar_), 122.9 (1C, C-4_Ar_), 126.8 (1C, C-4_Ph_), 127.3 (2C, C-3_Ph_, C-5_Ph_), 127.6 (2C, C-2_Ph_, C-6_Ph_), 128.9 (1C, C-1_Ar_), 130.6 (1C, C-5_Ar_), 137.9 (1C, C-1_Ph_), 145.6, 145.9 (1C, C-2_Ar_), 153.4, 153.7 (1C, C=O). *anti,anti*-**4p**:*anti,syn*-**4p** = 1:1. Purity (HPLC): 96.8% (*t_R_* = 21.20 min).

#### 5.3.18. [(8-*anti*-11-*anti* and 8-*anti*-11-*syn*)-11-(3-Phenylpropylamino)[4.3.3]propellan-8-yl] *N-*(2-methoxy-5-methylphenyl)carbamate (8-*anti*-**4q**)

NaBH(OAc)_3_ (0.3 g, 1.42 mmol) was added to a solution of ketone *anti*-**7** (0.13 g, 0.36 mmol), 3-phenypropan-1-amine (64 mg, 0.47 mmol) and acetic acid (30 μL, 0.36 mmol) in 1,2-dichloroethane (5 mL, dried over molecular sieves 4 Å). The mixture was stirred at rt for 72 h. Then NaOH (1 M) was added (pH 8–10), the mixture was extracted with CH_2_Cl_2_ (3×) and the combined organic layers were washed with brine (1×), dried (Na_2_SO_4_), filtered, the filtrate was concentrated in vacuo and the residue was purified by fc (3 cm, cyclohexane:Et_2_O = 8:2–1:1, 10 mL, R*_f_* = 0.56, cyclohexane:ethylacetate = 7:3) to obtain a mixture of diastereoisomeric aminocarbamates *anti,anti-***4q** and *anti,syn*-**4q** as pale yellow oil, yield 0.12 g (63%). C_30_H_40_N_2_O_3_ (476.7). Exact mass (APCI): *m/z* = 477.3134 (calcd. 477.3112 for C_30_H_41_N_2_O_3_ [M+H]^+^). FT-IR (ATR, film): ν (cm^−1^) = 3267 (ν*N-H*), 2931 (ν *C-H* aliphatic), 1720 (ν*C=O*), 1597 (δ *N-H*). ^1^H NMR (400 MHz, CDCl_3_): δ (ppm) = 1.13–1.65 (m, 9H, 2-CH_2_, 3-CH_2_, 4-CH_2_, 5-CH_2_, 7-CH_2_(0.5H), 9-CH_2_(0.5H)), 1.92–2.28 (m, 12H, 7-CH_2_(1.5H), 9-CH_2_(1.5H), 10-CH_2_, 12-CH_2_, CH_3_, NCH_2_*CH_2_*CH_2_Ph), 2.58–2.66 (m, 2H, NCH_2_CH_2_*CH_2_*Ph), 2.81–2.88 (m, 2H, N*CH_2_*CH_2_CH_2_Ph), 3.52–3.68 (m, 2 × 0.5H, 11-H), 3.79 (s, 3 × 0.5H, OCH_3_), 3.84 (s, 3 × 0.5 H, OCH_3_), 5.20 (m, 2 × 0.5H, 8-H), 6.70–6.80 (m, 2H, 3-H_Ar_, 4-H_Ar_), 7.13–7.18 (m, 4H, 2-H_Ph_, 3-H_Ph_, 5-H_Ph_, 6-H_Ph_), 7.22–7.26 (m, 1H, 4-H_Ph_), 7.35 (s, 1H, NH_carbamate_), 7.62 (s, 1H. NH), 7.92 (d, *J* = 2.0 Hz, 1H, 6-H_Ar_). A signal for the NH proton is not seen in the spectrum. ^13^C NMR (100 MHz, CDCl_3_): δ (ppm) = 20.4 (1C, CH_3_), 20.8, 21.1 (2C, C-3, C-4), 29.4 (NCH_2_*CH_2_*CH_2_Ph), 32.4 (NCH_2_CH_2_*CH_2_*Ph), 32.7, 33.1 (2C, C-2, C-5), 40.9, 41.6, 44.3, 45.1 (4C, C-7, C-9, C-10, C-12), 47.1 (N*CH_2_*CH_2_CH_2_Ph), 49.3, 49.7 (2C, C-1, C-6), 55.9, 56.0 (2 × 0.5C, C-11), 56.7, 58.0 (2 × 0.5C, OCH_3_), 76.0 (1C, C-8), 110.2 (1C, C-3_Ar_), 119.5 (1C, C-6_Ar_), 123.0 (1C, C-4_Ar_), 126.5 (1C, C-4), 128.5 (4C, C-2_Ph_, C-3_Ph_, C-5_Ph_, C-6_Ph_), 128.7(1C, C-1_Ar_), 130.7 (1C, C-5_Ar_), 140.0 (1C, C-1_Ph_), 146.1 (1C, C-2_Ar_), 153.7 (1C; C=O). *anti,anti*-**4q**:*anti,syn*-**4q** = 1:1. Purity (HPLC): 97.9% (*t_R_* = 21.61 min).

#### 5.3.19. [(8-*anti*-11-*anti* and 8-*anti*-11-*syn*)-11-[(3-Aminopropyl)amino][4.3.3]propellan-8-yl] *N-*(2-methoxy-5-methylphenyl)carbamate (8-*anti*-**4r**)

NaBH(OAc)_3_ (0.3 g, 1.42 mmol) was added to a solution of ketone *anti*-**7** (0.10 g, 0.28 mmol), propane-1,3-diamine (62 mg, 0.34 mmol) and acetic acid (16 μL, 0.28 mmol) in 1,2-dichloroethane (5 mL, dried over molecular sieves 4 Å). The mixture was stirred at rt for 96 h. Then NaOH (1 M) was added (pH 8–10), the mixture was extracted with CH_2_Cl_2_ (3×) and the combined organic layers were washed with brine (1×), dried (Na_2_SO_4_), filtered, the filtrate was concentrated in vacuo and the residue was purified by fc (1 cm, cyclohexane:ethyl acetate:methanol = 5.5:3.5:1–1:1, 10 mL, R*_f_* = 0.18) to obtain a mixture of diastereoisomeric aminocarbamates *anti,anti-***4r** and *anti,syn*-**4r** as colorless solid, mp 74–76 °C, yield 80 mg (66%). C_24_H_37_N_3_O_3_ (415.6). Exact mass (APCI): *m/z* = 417.2856 (calcd. 417.2991 for C_24_H_39_N_3_O_3_ [M+2H]^+^). FT-IR (ATR, film): ν (cm^−1^) = 3429 (ν *N-H*), 2927 (ν *C-H* aliphatic), 1720 (ν *C=O*), 1597 (δ *N-H*). ^1^H NMR (600 MHz, CDCl_3_): δ (ppm) = 1.26–1.51 (m, 10H, 2-CH_2_, 3-CH_2_, 4-CH_2_, 5-CH_2_, NCH_2_*CH_2_*CH_2_NH_2_), 1.63–2.25 (m, 10H, 7-CH2, 9-CH_2_, 10-CH_2_, 12-CH_2_, NCH_2_CH_2_*CH_2_*NH_2_), 2.24 (s, 3 × 0.5H, CH_3_), 2.28 (s, 3 × 0.5H, CH_3_), 2.82–2.88 (m, 2H, N*CH_2_*CH_2_CH_2_NH_2_), 3.39–3.53 (m, 2 × 0.5H, 11-H), 3.79 (s, 3 × 0.5H, OCH_3_), 3.86 (s, 3 × 0.5H, OCH_3_), 5.17–5.30 (m, 2 × 0.5H, 8-H), 6.69–6.78 (m, 2H, 3-H_Ar_, 4-H_Ar_), 7.14 (s, 1H, NH), 7.89 (s, 1H, 6-H_Ar_). Signals for the NH protons are not seen in the spectrum. ^13^C NMR (150 MHz, CDCl_3_): δ (ppm) = 20.7 (1C, CH_3_), 21.1, 21.4 (2C, C-3, C-4), 32.0, 32.3, 32.3, 32.8 (3C, C-2, C-5, NCH_2_*CH_2_*CH_2_NH_2_), 44.5, 44.9 (4C, C-7, C-9, C-10, C-12), 47.6 (1C, NCH_2_CH_2_*CH_2_*NH_2_), 49.6 (1C, N*CH_2_*CH_2_CH_2_NH_2_), 50.2 (2C, C-1, C-6), 55.9 (1C, OCH_3_), 58.3 (0.5C, C-11), 59.1 (0.5C, C-11), 75.6 (0.5C, C-8), 76.0 (0.5C, C-8), 110.1 (1C, C-3_Ar_), 119.3 (1C, C-6_Ar_), 123.0 (C-4_Ar_), 127.6 (1C, C-1_Ar_), 130.6 (1C, C-5_Ar_), 145.9 (1C, C-2_Ar_), 153.6 (1C, C=O). *anti,anti*-**4r**:*anti,syn*-**4r** = 1:1. Purity (HPLC): 92.3% (*t_R_* = 20.97 min).

#### 5.3.20. [(8-*anti*-11-*anti* and 8-*anti*-11-*syn*)-11-(2-Hydroxyethyl-1-amino)[4.3.3]propellan-8-yl] *N-*(2-methoxy-5-methylphenyl)carbamate (8-*anti*-**4s**)

NaBH(OAc)_3_ (0.3 g, 1.42 mmol) was added to a solution of ketone *anti*-**7** (0.10 g, 0.28 mmol), 2-aminoethanol (20 mg, 0.34 mmol) and acetic acid (16 μL, 0.28 mmol) in 1,2-dichloroethane (5 mL, dried over molecular sieves 4 Å). The mixture was stirred at rt for 7 days. Then NaOH (1 M) was added (pH 8–10), the mixture was extracted with CH_2_Cl_2_ (3×) and the combined organic layers were washed with brine (1×), dried (Na_2_SO_4_), filtered, the filtrate was concentrated in vacuo and the residue was purified by fc (1 cm, ethyl acetate:methanol = 8:2–1:1, 10 mL, R*_f_* = 0.11, cyclohexane:ethyl acetate:methanol = 5.5:3.5:1) to obtain a mixture of diastereoisomeric aminocarbamates *anti,anti-***4s** and *anti,syn*-**4s** as yellow oil, yield 35 mg (33%). C_23_H_34_N_2_O_4_ (402.5). Exact mass (APCI): *m/z* = 403.2626 (calcd. 403.2591 for C_23_H_35_N_2_O_4_ [M+H]^+^). FT-IR (ATR, film): ν (cm^−1^) = 3429 (ν *N-H*), 2927 (ν *C-H* aliphatic), 1724 (ν*C=O*), 1597 (δ *N-H*). ^1^H NMR (400 MHz, CDCl_3_): δ (ppm) = 1.28–1.53 (m, 8H, 2-CH_2_, 3-CH_2_, 4-CH_2_, 5-CH_2_), 1.59–1.77 (m, 2H, 7-H, 9-H), 1.87–2.27 (m, 6H, 7-CH_2_(1H), 9-CH_2_(1H), 10-CH_2_, 12-CH_2_), 2.28 (m, 3H, CH_3_), 2.76–2.84 (m, 2H, N*CH_2_*CH_2_OH), 3.37–3.53 (m, 2 × 0.5H, 11-H), 3.66–3.71 (m, 2H, NCH_2_*CH_2_*OH), 3.84 (s, 3H, OCH_3_), 5.18–5.31 (m, 2 × 0.5H, 8-H), 6.73 (dd, *J* = 8.3/2.1 Hz, 1H, 4-H_Ar_), 6.77 (d, *J* = 8.3 Hz, 1H, 3-H_Ar_), 7.14 (s, 1H, NH), 7.92 (s, 1H, 6-H_Ar_). Signals for the OH and NH protons are not seen in the spectrum. ^13^C NMR (100 MHz, CDCl_3_): δ (ppm) = 20.9 (1C, CH_3_), 21.1, 21.4 (2C, C-3, C-4), 32.4, 32.9 (2C, C-2, C-5), 44.5, 44.8, 45.0 (4C, C-7, C-9, C-10, C-12), 49.6 (2C, C-1, C-6), 50.2, 50.4 (2 × 0.5C, N*CH_2_*CH_2_OH), 55.9 (1C, OCH_3_), 56.5 (0.5C, C-11), 57.7 (0.5C, C-11), 60.8 (1C, NCH_2_*CH_2_*OH), 75.9 (0.5C, C-8), 76.4 (0.5C, C-8), 109.9(1C, C-3_Ar_), 118.9 (1C, C-6_Ar_), 122.9 (1C, C-4_Ar_), 127.5 (1C, C-1_Ar_), 130.6 (1C, C-5_Ar_), 145.6 (1C, C-2_Ar_), 153.4 (1C, C=O). *anti,anti*-**4s**:*anti,syn*-**4s** = 1:1. Purity (HPLC): 92.5% (*t_R_* = 17.37 min).

#### 5.3.21. [(8-*anti*-11-*anti* and 8-*anti*-11-*syn*)-11-(5-Hydroxypentylamino)[4.3.3]propellan-8-yl] *N-*(2-methoxy-5-methylphenyl)carbamate (8-*anti*-**4t**)

NaBH(OAc)_3_ (0.3 g, 1.42 mmol) was added to a solution of ketone *anti*-**7** (0.10 g, 0.28 mmol), 5-aminopentan-1-ol (35 mg, 0.34 mmol) and acetic acid (16 μL, 0.28 mmol) in 1,2-dichloroethane (5 mL, dried over molecular sieves 4 Å). The mixture was stirred at rt for 11 days. Then NaOH (1 M) was added (pH 8–10), the mixture was extracted with CH_2_Cl_2_ (3×) and the combined organic layers were washed with brine (1×), dried (Na_2_SO_4_), filtered, the filtrate was concentrated in vacuo and the residue was purified by fc (3 cm, ethyl acetate:methanol = 8:2–1:1, 20 mL, R*_f_* = 0.23, ethyl acetate:methanol = 1:1) to obtain a mixture of diastereoisomeric aminocarbamates *anti,anti-***4t** and *anti,syn*-**4t** as yellow oil, yield 25 mg (21%). C_26_H_40_N_2_O_4_ (444.6). Exact mass (APCI): *m/z* = 445.3066 (calcd. 445.3061 for C_26_H_40_N_2_O_4_ [M+H]^+^ ). FT-IR (ATR, film): ν (cm^−1^) = 3425 (ν *N-H*), 3330 (ν*O-H*), 2931 (ν *C-H* aliphatic), 1724 (ν*C=O*), 1597 (δ *N-H*). ^1^H NMR (400 MHz, CDCl_3_): δ (ppm) = 1.25–1.48 (m, 10H, 2-CH_2_, 3-CH_2_, 4-CH_2_, 5-CH_2_, NCH_2_CH_2_*CH_2_*CH_2_CH_2_O), 1.51–1.71 (m, 6H, 7-H, 9-H, 10-H, 12-H, NCH_2_*CH_2_*CH_2_CH_2_CH_2_O), 1.95–2.25 (m, 6H, 7-H, 9-H, 10-H, 12-H, NCH_2_CH_2_CH_2_*CH_2_*CH_2_O), 2.27 (s, 3 × 0.5H, CH_3_), 2.29 (s, 3 × 0.5H, CH_3_), 2.93 (m, 2H, N*CH_2_*CH_2_CH_2_CH_2_CH_2_O), 3.65–3.80 (m, 3H, 11-H, *CH_2_*O), 3.81 (s, 3 × 0.5H, OCH_3_), 3.84 (m, 3 × 0.5H, OCH_3_), 5.17–5.31 (m, 2 × 0.5H, 8-H), 6.70–6.80 (m, 2H, 3-H_Ar_, 4-H_Ar_), 7.16 (s, 1H, NH), 7.88 (s, 1H, 6-H_Ar_). A signal for the NH proton is not seen in the spectrum. Signals for the OH and NH protons are not seen in the spectrum. ^13^C NMR (100 MHz, CDCl_3_): δ(ppm) = 20.6 (2C, C-3, C-4), 21.1 (1C, CH_3_), 23.3 (1C, NCH_2_CH_2_*CH_2_*CH_2_CH_2_O), 25.8 (1C, NCH_2_CH_2_CH_2_*CH_2_*CH_2_O), 31.3 (1C, NCH_2_*CH_2_*CH_2_CH_2_CH_2_O), 32.7 (2C, C-2, C-5), 41.0 (2C, C-10, C-12), 45.1 (2C, C-7, C-9), 47.2 (1C, N*CH_2_*CH_2_CH_2_CH_2_CH_2_O), 49.6 (2C, C-1, C-6), 55.9 (1C, OCH_3_), 56.8 (0.5C, C-11), 58.2 (0.5C, C-11), 61.9 (1C, NCH_2_CH_2_CH_2_CH_2_*CH_2_*O), 75.5 (0.5C, C-8), 76.2 (0.5C, C-8), 110.2 (1C, C-3_Ar_), 119.5 (1C, C-6_Ar_), 123.1 (1C, C-4_Ar_), 127.5 (1C, C-1_Ar_), 130.6 (1C, C-5_Ar_), 146.0 (1C, C-2_Ar_), 153.5 (1C, C=O). *anti,anti*-**4t**:*anti,syn*-**4t** = 1:1. Purity (HPLC): 94.4% (*t_R_* = 22.40 min).

#### 5.3.22. [(8-*anti*-11-*anti* and 8-*anti*-11-*syn*)-11-(Isobutylamino)[4.3.3]propellan-8-yl] *N-*(2-methoxy-5-methylphenyl)carbamate (8-*anti*-**4u**)

NaBH(OAc)_3_ (0.3 g, 1.42 mmol) was added to a solution of ketone *anti*-**7** (0.10 g, 0.28 mmol), isobutylamine (25 mg, 0.34 mmol) and acetic acid (16 μL, 0.28 mmol) in 1,2-dichloroethane (5 mL, dried over molecular sieves 4 Å). The mixture was stirred at rt for 6 days. Then NaOH (1 M) was added (pH 8–10), the mixture was extracted with CH_2_Cl_2_ (3×) and the combined organic layers were washed with brine (1×), dried (Na_2_SO_4_), filtered, the filtrate was concentrated in vacuo and the residue was purified by fc (3 cm, cyclohexane:ethyl acetate:methanol = 5.5:3.5:1, 20 mL, R*_f_* = 0.23, ethyl acetate:methanol = 1:1) to obtain a mixture of diastereoisomeric aminocarbamates *anti,anti-***4u** and *anti,syn*-**4u** as yellow oil, yield 0.11 g (92%). C_25_H_38_N_2_O_3_ (414.6). Exact mass (APCI): *m/z* = 415.2969 (calcd. 415.2955 for C_25_H_39_N_2_O_3_ [M+H]^+^). FT-IR (ATR, film): (ν (cm^−1^) = 3429 ν *N-H*), 2927 (ν*C-H* aliphatic), 1724 (ν *C=O*), 1597 (δ*N-H*). ^1^H NMR (600 MHz, CDCl_3_): δ (ppm) = 1.00 (d, *J* = 6.6 Hz, 3H, NCH_2_CH(*CH_3_*)_2_), 1.01 (d, *J* = 6.6 Hz, 3H, NCH_2_CH(*CH_3_*)_2_), 1.30–1.43 (m, 6H, 2-CH_2_(1H), 3-CH_2_, 4-CH_2_, 5-CH_2_(1H)), 1.49–1.62 (m, 2H, 2-H, 5-H), 1.70–1.76 (m, 1H, NCH_2_*CH*(CH_3_)_2_), 1.95–2.23 (m, 8H 7-CH_2_, 9-CH_2_, 10-CH_2_, 12-CH_2_), 2.28 (s, 3 × 0.5H, CH_3_), 2.29 (s, 3 × 0.5H, CH_3_), 2.55 (d, *J* = 6.9 Hz, 1H, N*CH_2_*CH(CH_3_)_2_), 2.58 (d, *J* = 6.9 Hz, 1H, N*CH_2_*CH(CH_3_)_2_), 3.48–3.54 (m, 0.5H, 11-H), 3.55–3.62 (m, 0.5H, 11-H), 3.82 (s, 1.5H, OCH_3_), 3.84 (s, 1.5H, OCH_3_), 5.21 (tt, *J* = 8.0/5.5 Hz, 0.5H, 8-H), 5.26 (tt, *J* = 8.4/4.2 Hz, 0.5H, 8-H), 6.71–6.79 (m, 2H, 3-H_Ar_, 4-H_Ar_), 7.14 (s, 1H, NH), 7.91 (s, 1H, 6-H_Ar_). A signal for the NH proton is not seen in the spectrum. Signals for the OH and NH protons are not seen in the spectrum. ^13^C NMR (150 MHz, CDCl_3_): δ (ppm) = 21.0, 21.1, 21.1, 21.2 (5C, C-3, C-4, CH_3_, NCH_2_CH*(CH_3_)_2_*), 32.6, 33.0 (2C, C-2, C-5), 44.7, 45.3 (5C, C-7, C-9, C-10, C-12, NCH_2_*CH*(CH_3_)_2_), 49.3, 50.0 (2C, C-1, C-6), 55.9 (2C, OCH_3_, N*CH_2_*CH(CH_3_)_2_), 56.8 (0.5C, C-11), 58.2 (0.5C, C-11), 75.7 (0.5C, C-8), 76.3 (0.5C, C-8), 110.1, 119.1, 122.9, 127.6, 130.7, 145.8, 153.6. *anti,anti*-**4u**:*anti,syn*-**4u** = 1:1. Purity (HPLC): 95.8% (*t_R_* = 20.07 min).

#### 5.3.23. [(8-anti-11-anti and 8-anti-11-syn)-11-(Benzylamino)[4.3.3]propellan-8-ol (11-anti-13)

NaBH(OAc)_3_ (0.33 g, 1.55 mmol) was added to a solution of hydroxyketone *anti*-**11** (0.10 g, 0.52 mmol), benzylamine (72 mg, 0.68 mmol) and acetic acid (30 μL, 0.52 mmol) in 1,2-dichloroethane (5 mL, dried over molecular sieves 4 Å). The mixture was stirred at rt for 8 days. Then NaOH (1 M) was added (pH 8–10), the mixture was extracted with CH_2_Cl_2_ (3×) and the combined organic layers were washed with brine (1×), dried (Na_2_SO_4_), filtered, the filtrate was concentrated in vacuo and the residue was purified by fc (1 cm, cyclohexane:ethyl acetate:methanol = 5.5:3.5:1, 10 mL, R*_f_* = 0.28) to obtain a mixture of diastereoisomeric aminoalcohols *anti,anti-***13** and *anti,syn*-**13** as colorless solid, mp 121–123 °C, yield 86 mg (61%). C_19_H_27_NO (285.4). MS (ESI): *m/z* = 286 [M+H]^+^. Exact mass (APCI): *m/z* = 286.2145 (calcd.286.2165 for C_19_H_28_NO [M+H]^+^). FT-IR (ATR, film): ν (cm^−1^) = 3313 (δ *O-H*), 2931 (ν *C-H* aliphatic). ^1^H NMR (600 MHz, CDCl_3_): δ (ppm) = 1.24–1.40 (m, 6H, 2-CH_2_(1H), 3-CH_2_, 4-CH_2_, 5-CH_2_(1H)), 1.40–1.45 (m, 2H, 7-H, 9-H, 10-H, 12-H), 1.50–1.62 (m, 2H, 2-CH_2_(1H), 5-CH_2_(1H)), 1.96 (dd, *J* = 13.8/7.1 Hz, 2 × 0.3H, 10-H, 12-H), 2.03–2.16 (m, 4H, 7-H, 9-H, 10-H, 12-H), 2.30–2.36 (m, 2 × 0.3H, 7-H, 9-H), 3.45–3.51 (m, 0.3H, 11-H), 3.60–3.66 (m, 0.7H, 11-H), 3.93 (s, 2 × 0.3H, NCH_2_Ph), 3.95 (s, 2 × 0.7H, NCH_2_Ph), 4.32 (tt, *J* = 7.1/5.3 Hz, 0.7H, 8-H), 4.40 (tt, *J* = 8.0/4.2 Hz, 0.3H, 8-H), 7.31–7.40 (m, 3H, 3-H_Ph_, 4-H_Ph_, 5-H_Ph_), 7.59 (dd, *J* = 6.7/1.3 Hz, 2H, 2-H_Ph_, 6-H_Ph_). Signals for the NH and OH protons are not observed in the spectrum ^13^C NMR (150 MHz, CDCl_3_): δ (ppm) = 19.5 (2C, C-3, C-4), 32.8 (2C, C-2, C-5), 41.1 (2C, C-10, C-12), 48.9 (2C, C-7, C-9), 49.9 (2C, C-1, C-6), 50.6 (1C, NCH_2_Ph), 54.4 (0.7C, C-11), 56.0 (0.3C, C-11), 71.6 (0.7C, C-8), 72.6 (0.3C, C-8), 129.0 (1C, C-4_Ph_), 129.3 (2C, C-3_Ph_, C-5_Ph_), 130.5(2C, C-2_Ph_, C-6_Ph_). *anti,anti*-**13**:*anti,syn*-**13** = 7:3. Purity (HPLC): 93.3% (*t_R_* = 13.61, 15.15 min).

#### 5.3.24. [(8-syn-11-syn and 8-syn-11-anti)-11-(Benzylamino)[4.3.3]propellan-8-ol (11-syn-13)

NaBH(OAc)_3_ (0.33 g, 1.55 mmol) was added to a solution of hydroxyketone *syn*-**11** (0.10 g, 0.52 mmol), benzylamine (72 mg, 0.68 mmol) and acetic acid (30 μL, 0.52 mmol) in 1,2-dichloroethane (5 mL, dried over molecular sieves 4 Å). The mixture was stirred at rt for 8 days. Then NaOH (1 M) was added (pH 8–10), the mixture was extracted with CH_2_Cl_2_ (3×) and the combined organic layers were washed with brine (1×), dried (Na_2_SO_4_), filtered, the filtrate was concentrated in vacuo and the residue was purified by fc (1 cm, cyclohexane:ethyl acetate:methanol = 5.5:3.5:1, 10 mL, R*_f_* = 0.28) to obtain a mixture of diastereoisomeric aminoalcohols *syn,syn-***13** and *syn,anti*-**13** as colorless oil, yield 90 mg (64%). C_19_H_27_NO (285.4). MS (ESI): *m/z* = 286 [M+H]^+^. Exact mass (APCI): *m/z* = 286.2215 (calcd.286.2165 for C_19_H_28_NO [M+H]^+^). FT-IR (ATR, film): ν (cm^−1^) = 3290 (ν *O-H*), 2927 (ν *C-H* aliphatic). ^1^H NMR (600 MHz, CDCl_3_): δ (ppm) = 1.2–61.39 (m, 2H, 3-CH_2_(1H), 4-CH_2_(1H)), 1.44–1.61 (m, 6H, 2-CH_2_, 3-CH_2_(1H), 4-CH_2_(1H), 5-CH_2_), 1.62–1.88 (m, 6H, 7-CH_2_(1H), 9-CH_2_(1H), 10-CH_2_, 12-CH_2_), 1.93 (dd, *J* = 13.3/8.3 Hz, 2 × 0.5H, 7-H, 9-H), 2.15 (dd, *J* = 13.8/7.5 Hz, 2 × 0.5H, 7-H, 9-H), 3.28 (q, *J* = 8.2 Hz, 0.5H, 11-H), 3.35 (q, *J* = 8.4 Hz, 0.5H, 11-H), 3.82 (s, 2H, NCH_2_Ph), 4.33 (tt, *J* = 7.5/5.5 Hz, 0.5H, 8-H), 4.54 (tt, *J* = 7.5/5.4 Hz, 0.5H, 8-H), 7.27–7.32 (m, 1H, 4-H_Ph_), 7.32–7.36 (m, 2H, 3-H_Ph_, 5-H_Ph_), 7.39–7.43 (m, 2H, 2-H_Ph_, 6-H_Ph_). Signals for the NH and OH protons are not observed in the spectrum. ^13^C NMR (150 MHz, CDCl_3_): δ (ppm) = 20.2, 20.6 (2C, C-3, C-4), 32.1, 33.4 (2C, C-2, C-5), 43.6, 47.9 (4C, C-7, C9, C-10, C-12), 49.3 (2C, C-1, C-6), 51.0 (1C, NCH_2_Ph), 55.3 (0.5C, C-11), 55.6 (0.5C, C-11), 71.3 (0.5C, C-8), 72.1 (0.5C, C-8), 128.3 (1C, C-4_Ph_), 128.7, 128.8 (2C, C-3_Ph_, C-5_Ph_), 129.3, 129.5 (2C, C-2_Ph_, C-6_Ph_), 134.4 (1C, C-1_Ph_). *syn,syn*-**13**:*syn,anti*-**13** = 1:1. Purity (HPLC): 93.1% (*t_R_* = 12.42, 13.90 min).

#### 5.3.25. (11′-*anti*)-Spiro-([1,3]dioxolane-2,8′-(*N*-benzyl[4.3.3]propellan))-11′-amine (*anti*-**14**) and (11′-*syn*)-spiro-([1,3-dioxolane-2,8′-(*N*-benzyl[4.3.3]propellan))-11′-amine (*syn*-**14**)

Under N_2_, NaBH(OAc)_3_ (1.35 g, 6.37 mmol) was added to a solution of monoketal **12** (0.5 g, 2.12 mmol), benzylamine (0.45 g, 4.24 mmol) and acetic acid (0.12 mL, 2.12 mmol) in 1,2-dichloroethane (15 mL, dried over molecular sieves 4 Å). The mixture was stirred at rt for 24 h. Then NaOH (1 M) was added (pH 8–10), the mixture was extracted with CH_2_Cl_2_ (3×) and the combined organic layers were washed with brine (1×), dried (Na_2_SO_4_), filtered, the filtrate was concentrated in vacuo and the residue was purified by fc (3 cm, cyclohexane:Et_2_O: NEt_3_ = 5.5:3.5:1, 20 mL).

*syn-***14** (R*_f_* = 0.43) Pale yellow oil, yield 0.19 g (28%). C_21_H_29_NO_2_ (327.5). MS (ESI): *m/z* = 328 [M+H]^+^. Exact mass (APCI): *m/z* = 328.2295 (328.2271 calcd. for C_21_H_30_NO_2_ [M+H]^+^). FT-IR (ATR, film): ν (cm^−1^) = 2927 (ν *C-H* aliphatic). ^1^H NMR (400 MHz, Toluene-*d*_8_): δ (ppm) = 1.60–1.65 (m, 4H, 3′-CH_2_, 4′-CH_2_,), 1.67–1.83 (m, 6H, 2′-CH_2_, 5′-CH_2_, 10′-CH_2_(1H), 12′-CH_2_(1H)), 2.11 (d, *J* = 14.0 Hz, 2H, 7′-H, 9′-H), 2.16–2.24 (m, 4H; 7′-H, 9′-H, 10′-H, 12′-H), 3.49 (tt, *J* = 8.5/5.6 Hz, 1H, 8′-H), 3.67–3.69 (m, 4H, OCH_2_CH_2_O), 3.76 (s, 2H, NCH_2_Ph), 7.23 (m, 1H, 4-H_Ph_), 7.38 (dd, 2H, *J* = 8.7/6.4 Hz, 2H, 3-H_Ph_, 5-H_Ph_), 7.49 (m, 2H, 2-H_Ph_, 6-H_Ph_). A signal for the NH proton is not observed in the spectrum. ^13^C NMR (100 MHz, Toluene-*d*_8_): δ (ppm) = 21.9 (2C, C-3′, C-4′), 32.4 (2C, C-2′, C-5′), 44.9 (2C, C-10′, C-12′), 49.2 (2C, C-7′, C-9′), 49.6 (2C, C-1′, C-6′), 53.4 (1C, NCH_2_Ph), 56.7 (C-11′), 63.9 (1C, C-8′), 64.0 (2C, OCH_2_CH_2_O), 105,1 (1C, C-4_Ph_), 126.9 (2C, C-3_Ph_, C-5_Ph_), 128.4 (2C, C-2_Ph_, C-6_Ph_), 141.1 (1C, C-1_Ph_). Purity (HPLC): 83.8% (*t_R_* = 15.67 min).

*anti*-**14** (R*_f_* = 0.38) Pale yellow oil, yield 0.21 g (30%). C_21_H_29_NO_2_ (327.5). MS (ESI): *m/z* = 328 [M+H]^+^. Exact mass (APCI): *m/z* = 328.2270 (328.2271 calcd. for C_21_H_30_NO_2_ [M+H]^+^). FT-IR (ATR, film): ν (cm^−1^) = 2927 (ν *C-H* aliphatic). ^1^H NMR (400 MHz, Toluene-*d*_8_): ν (ppm) = 1.39–1.52 (m, 4H, 3′-CH_2_, 4′-CH_2_,), 1.55–1.61 (m, 2H, 2′-H, 5′-H), 1.67–1.73 (m, 2H, 2′-H, 5′-H), 1.87 (dd, *J* = 13.2/6.3 Hz. 2H, 10′-H_anti_, 12′-H_anti_), 2.05 (dd, *J* = 13.2/8.5 Hz, 2H, 10′-H_syn_, 12′-H_syn_), 2.26 (d, *J* = 14.1 Hz, 2H, 7′-H, 9′-H), 2.40 (d, *J* = 14.1 Hz, 2H, 7′-H, 9′-H), 3.44 (tt, *J* = 8.3/6.3 Hz, 1H, 11′-H), 3.70 (s, 4H, OCH_2_CH_2_O), 3.78 (s, 2H, NCH_2_Ph), 7.26 (m, 1H, 4-H_Ph_), 7–38 (m, 2H, 3-H_Ph_, 5-H_Ph_), 7.49 (d, 2H, *J* = 7.6 Hz, 2-H_Ph_, 6-H_Ph_). A signal for the NH proton is not observed in the spectrum. ^13^C NMR (100 MHz, Toluene-*d*_8_): δ (ppm) = 21.9 (2C, C-3′, C-4′), 31.9 (2C, C-2′, C-5′), 45.1 (2C, C-10′, C-12′), 49.4 (2C, C-7′, C-9′), 49.8 (2C, C-1′, C-6′), 53.3 (1C, NCH_2_Ph), 56.9 (1C, C-11′), 63.9 (1C, C-8′), 64.0 (2C, OCH_2_CH_2_O), 117.5 (1C, C-4_Ph_), 126.9 (2C, C-3_Ph_, C-5_Ph_), 128.4 (2C, C-2_Ph_, C-6_Ph_), 141.6 (1C, C-1_Ph_). Purity (HPLC): 98.2% (*t_R_* = 16.24 min).

#### 5.3.26. *syn*- and *anti*-*N-*[2-(Indol-3-yl)ethyl]-[4.3.3]propellan-8-amine (**15**)

Under N_2_, NaBH(OAc)_3_ (0.3 g, 1.42 mmol) was added to a solution of monoketone **10** (0.1 g, 0.56 mmol), tryptamine (0.14 g, 8.41 mmol) and acetic acid (0.32 μL, 0.56 mmol) in 1,2-dichloroethane (5 mL, dried over molecular sieves 4 Å). The mixture was stirred at rt for 5 days. Then NaOH (1 M) was added (pH 8–10), the mixture was extracted with CH_2_Cl_2_ (3×) and the combined organic layers were washed with brine (1×), dried (Na_2_SO_4_), filtered, the filtrate was concentrated in vacuo and the residue was purified by fc (1 cm, cyclohexane:ethyl acetate:methanol = 8:1:1 5 mL, R*_f_* = 0.35) to obtain a mixture of diastereoisomeric amines *syn-***15** and *anti*-**15** as a brown solid, mp 109–111 °C, yield 60 mg (33%). C_22_H_30_N_2_ (322.5). MS (ESI): *m/z* = 323 [M+H]^+^. Exact mass (APCI): *m/z* = 323.2506 (calcd. 323.2482 for C_22_H_31_N_2_ [M+H]^+^). FT-IR (ATR, film): ν (cm^−1^) = 3429 (ν*N-H*), 2927 (ν *C-H* aliphatic). ^1^H NMR (600 MHz, CDCl_3_): δ (ppm) = 1.40–1.90 (m, 8H, 2-CH_2_, 3-CH_2_, 4-CH_2_, 5-CH_2_), 1.54–1.64 (m, 5H, 10-CH_2_, 11-CH_2_(1H), 12-CH_2_), 1.68–1.71 (m, 3H, 7-CH_2_(1H), 9-CH_2_(1H), 11-CH_2_(1H)), 1.83–1.86 (m, 2 × 0.5H, 7-H, 9-H), 1.89–1.93 (2 × 0.5H, 7-H, 9-H), 2.77 (s, broad, 1H, NH), 3.06–3.10 (m, 2H, N*CH_2_*CH_2_), 3.16–3.19 (m, 2H, NCH_2_*CH_2_*), 3.45 (m, 2 × 0.5H, 8-H), 7.06–7.10 (m, 2H, 5-H_indole_, 6-H_indole_), 7.15 (s, 1H, 2-H_indole_), 7.35 (d, *J* = 8.1 Hz, 1H, 7-H_indole_), 7.56 (d, *J* = 7.2 Hz, 1H, 4-H_indole_), 8.95 (s, broad, 1H, NH_indole_). ^13^C NMR (150 MHz, CDCl_3_): δ (ppm) = 20.5, 20.9 (2C, C-3, C-4), 21.4, 22.3 (2 × 0.5 C, C-11), 23.2 (1C, NCH_2_*CH_2_*), 31.9, 32.6 (2C, C-2, C-5), 37.6, 39.1 (2C, C-10, C-12), 41.9, 42.2 (2C, C-7, C-9), 47.5 (1C, N*CH_2_*CH_2_), 49.9, 50.5 (2C, C-1, C-6), 56.5, 57.5 (2 × 0.5C, C-8), 111.4 (1C, C-3_indole_), 111.5 (1C, C-7_indole_), 118.7 (1C, C-4_indole_), 119.7 (1C, C-5_indole_), 122.3 (1C, C-2_indole_), 122.9 (1C, C-6_indole_), 127.1 (1C, C-3a_indole_), 136.56 (1C, C-7a_indole_). *syn*-**15**:*anti*-**15** = 1:1. Purity (HPLC): 96.8% (*t_R_* = 19.28 min).

#### 5.3.27. *syn*- and *anti*-*N*-(4-(Dimethylaminobenzyl)-[4.3.3]propellan-8-amine (**17**)

Under N_2_, NaBH(OAc)_3_ (89 mg, 0.42 mmol) was added to a solution of propellanamine *syn*-**16**/*anti*-**16** (50 mg, 0.28 mmol) and 4-(dimethylamino)benzaldehyde (44 mg, 0.29 mmol) in 1,2-dichloroethane (5 mL, dried over molecular sieves 4 Å). The mixture was stirred at rt for 72 h. Then NaOH (1 M) was added (pH 8–10), the mixture was extracted with CH_2_Cl_2_ (3×) and the combined organic layers were washed with brine (1×), dried (Na_2_SO_4_), filtered, the filtrate was concentrated in vacuo and the residue was purified by fc (1 cm, cyclohexane:ethyl acetate:methanol = 8:1:1 20 mL, R*_f_* = 0.32) to obtain a mixture of diastereoisomeric amines *syn-***17** and *anti*-**17** as a yellow solid, mp 89–91 °C, yield 86 mg (98%). C_21_H_32_N_2_ (312.5). MS (ESI): *m/z* = 313 [M+H]^+^. Exact mass (APCI): *m/z* = 313.2673 (calcd. 313.2638 for C_21_H_33_N_2_ [M+H]^+^). ^1^H NMR (600 MHz, CDCl_3_): δ(ppm) = 1.27–1.50 (m, 8H, 2-CH_2_, 3-CH_2_, 4-CH_2_, 5-CH_2_), 1.54–1.78 (m, 7H, 7-CH_2_(1H), 9-CH_2_(1H), 10-CH_2_, 11-CH_2_(1H), 12-CH_2_), 1.79–1.81 (m, 1H, 11-CH_2_(1H), 1.86 (dd, *J* = 13.1/8.0 Hz, 2 × 0.5H, 7-H, 9-H), 1.92 (dd, *J* = 13.1/8.2, 2 × 0.5H, 7-H, 9-H), 2.04 (s, 1H, NH), 2.91 (s, 6H, N(CH_3_)_2_), 3.31–3.38 (m, 2 × 0.5H, 8-H), 3.70 (s, 2H, NCH_2_Ar), 6.69 (d, *J* = 8.6 Hz, 2H, 3-H_Ar_, 5-H_Ar_), 7.25 (d, *J* = 8.6 Hz, 2H, 2-H_Ar_, 6-H_Ar_). ^13^C NMR (150 MHz, CDCl_3_): δ (ppm) = 20.7, 20.9 (2C, C-3, C-4), 21.2 (1C, C-11), 31.9, 32.6 (2C, C-2, C-5), 37.6, 38.9 (2C, C-10, C-12), 40.6 (2C, N(CH_3_)_2_), 43.6, 44.1 (2C, C-7, C-9), 49.9, 50.4 (2C, C-1, C-6), 51.1 (1C, N*C*H_2_Ar), 55.0, 55.9 (2 × 0.5C, C-8), 112.5 (2C, C-3_Ar_, C-5_Ar_), 129.9 (3C, C-1_Ar_, C-2_Ar_, C-6_Ar_), 150.1 (1C, C-4_Ar_). *syn*-**17**:*anti*-**17** = 1:1. Purity (HPLC): 96.1% (*t_R_* = 15.12 min).

#### 5.3.28. *syn*- and *anti*-*N*-Cyclohexylmethyl[4.3.3]propellan-8-amine (**18**)

Under N_2_, NaBH(OAc)_3_ (71 mg, 0.34 mmol) was added to a solution of propellanamine *syn-***16/***anti*-**16** (30 mg, 0.17 mmol) and cyclohexanecarbaldehyde (15 μL, 0.19 mmol) in 1,2-dichloroethane (5 mL, dried over molecular sieves 4 Å). The mixture was stirred at rt for 12 h. Then NaOH (1 M) was added (pH 8–10), the mixture was extracted with CH_2_Cl_2_ (3×) and the combined organic layers were washed with brine (1×), dried (Na_2_SO_4_), filtered, the filtrate was concentrated in vacuo and the residue was purified by fc (1 cm, cyclohexane:ethyl acetate:methanol = 8:1:1 20 mL, R*_f_* = 0.53) to obtain a mixture of diastereoisomeric amines *syn-***18** and *anti*-**18** as a colorless solid, mp 108–111 °C, yield 42 mg (91%). C_19_H_33_N (275.5). MS (ESI): *m/z* = 276 [M+H]^+^. Exact mass (APCI): *m/z* = 276.2710 (calcd. 276.2686 for C_19_H_34_N [M+H]^+^). ^1^H NMR (600 MHz, CDCl_3_): δ (ppm) = 0.89–0.98 (m, 2H, 2-CH_2Cy_(1H), 6-CH_2Cy_(1H), 1.16 (ttd, *J* = 12.5/3.3/1.2 Hz, 1H, 4-CH_2Cy_(1H)), 1.21–1.42 (m, 8H, 2-CH_2_(1H), 3-CH_2_, 4-CH_2_, 5-CH_2_(1H), 3-CH_2Cy_(1H), 5-CH_2Cy_(1H)), 1.45–1.87 (m, 16H, 2-CH_2_(1H), 5-CH_2_(1H), 7-CH_2_(1H), 9-CH_2_(1H), 10-CH_2_, 11-CH_2_, 12-CH_2_, 1-H_Cy_, 2-CH_2Cy_(1H), 3-CH_2Cy_(1H), 4-CH_2Cy_(1H), 5-CH_2Cy_(1H), 6-CH_2Cy_(1H)), 1.89 (dd, *J* = 13.1/8.0 Hz, 2 × 0.5H, 7-H, 9-H), 1.96 (dd, *J* = 13.1/8.3 Hz, 2 × 0.5H, 7-H, 9-H), 2.57 (dd, *J* = 6.8/1.5 Hz, 2H, NCH_2_Cy), 3.37–3.45 (m, 2 × 0.5 H, 8-H). A signal for the NH proton is not seen in the spectrum. ^13^C NMR (150 MHz, CDCl_3_): δ (ppm) = 20.5, 20.9 (2C, C-3, C-4), 21.5, 22.5 (2 × 0.5C, C-11), 25.8 (2C, C-3_Cy_, C-5_Cy_), 26.4 (1C, C-4_Cy_), 31.5 (2C, C-2_Cy_, C-6_Cy_), 32.1, 32.8 (2C, C-2, C-5), 36.0, 36.1 (2 × 0.5C, C-1_Cy_), 37.8, 39.5 (2C, C-10, C-12), 42.9, 43.4 (2C, C-7, C-9), 49.8, 50.4 (2C, C-1, C-6), 54.2 (1C, NCH_2_Cy), 56.9, 57.8 (2 × 0.5C, C-8). ). *syn-***18**:*anti-**18*** = 1:1. Purity (LC-MS): 97.4% (*t_R_* = 6.957 min).

#### 5.3.29. 11-*syn*-11-Benzylamino[4.3.3]propellan-11-one (*syn*-20) and 11-*anti*-11-Benzylamino[4.3.3]propellan-8-one (*anti*-**20**)

Under N_2_, NaBH(OAc)_3_ (4.40 g, 20.8 mmol) was added to a solution of diketone **19** (2.0 g, 10.4 mmol), benzylamine (1.34 g, 12.5 mmol) and acetic acid (0.60 mL, 10.40 mmol) in 1,2-dichloroethane (30 mL, dried over molecular sieves 4 Å). The mixture was stirred at rt for 96 h. Then NaOH (1 M) was added (pH 8–10), the mixture was extracted with CH_2_Cl_2_ (3×) and the combined organic layers were washed with brine (1×), dried (Na_2_SO_4_), filtered, the filtrate was concentrated in vacuo and the residue was purified by fc (5 cm, cyclohexane:ethyl acetate: NEt_3_ = 6.95:2.95:0.1 to 2.95:6.95:0.1, 20 mL).

*syn-***20** (R*_f_* = 0.29, cyclohexane: ethyl acetate: NEt_3_ = 69.5:29.5:1) Pale yellow solid, mp 77–79 °C, yield 0.46 g (16%). C_19_H_25_NO (283.4). Exact mass (APCI): *m/z* = 284.1973 (calcd. 284.2009 for C_19_H_26_NO [M+H]^+^). FT-IR (ATR, film): ν (cm^−1^) = 2927 (ν *C-H* aliphatic), 1724 (ν *C=O*). ^1^H NMR (400 MHz, CDCl_3_): δ (ppm) = 1.31–1.39 (m, 4H, 3-CH_2_, 4-CH_2_), 1.44–1.50 (m, 2H, 2-CH_2_(1H), 5-CH_2_(1H)), 1.61–1.66 (m, 2H, 2-CH_2_(1H), 5-CH_2_(1H)), 1.71 (dd, *J* = 13.7/5.6 Hz, 2H, 10-H_syn_, 12-H_syn_), 1.89 (dd, *J* = 13.7/8.6 Hz, 2H, 10-H_anti_, 12-H_anti_), 2.06 (d, *J* = 19.1 Hz, 2H, 7-H_anti_, 9-H_anti_), 2.18 (d, *J* = 19.1 Hz, 2H, 7-H_syn_, 9-H_syn_), 3.37 (tt, *J* = 8.6/5.6 Hz, 1H, 11-H), 3.67 (s, 2H, NCH_2_Ph), 7.16–7.20 (m, 1H, 4-H_Ph_), 7.23–7.28 (m, 4H, 2-H_Ph_, 3-H_Ph_, 5-H_Ph_, 6-H_Ph_). A signal for the NH proton is not observed in the spectrum. ^13^C NMR (100 MHz, CDCl_3_): δ (ppm) =21.9 (2C, C-3, C-4), 32.7 (2C, C-2, C-5), 44.5 (2C, C-10, C-12), 47.8 (2C, C-1, C-6), 50.0 (2C, C-7, C-9), 53.1 (1C, NCH_2_Ph), 56.1 (1C, C-11), 127.2 (1C, C-4_Ph_), 128.4 (2C, C-3_Ph_, C-5_Ph_), 128.7 (2C, C-2_Ph_, C-6_Ph_), 140.5 (1C, C-1_Ph_), 219.7 (1C, C=O). Purity (HPLC): 74.7% (*t_R_* = 13.83 min).

*anti-***20** (R*_f_* = 0.19, cyclohexane:ethyl acetate: NEt_3_ = 6.95:2.95:0.1) Pale yellow oil, yield 0.24 g (9%). C_19_H_25_NO (283.4). Exact mass (APCI): *m/z* = 284.2069 (calcd. 284.2009 for C_19_H_26_NO [M+H]^+^ ). FT-IR (ATR, film): ν (cm^−1^) = 2927 (ν *C-H* aliphatic), 1724 (δ *C=O*). ^1^H NMR (400 MHz, CDCl_3_): δ (ppm) = 1.31–1.50 (m, 8H, 2-CH_2_, 3-CH_2_, 4-CH_2_, 5-CH_2_), 1.54 (dd, *J* = 13.6/5.9 Hz, 2H, 10-H_anti_, 12-H_anti_), 2.16 (dd, *J* = 13.6/8.5 Hz, 2H, 10-H_syn_, 12-H_syn_), 2.27 (d, *J* = 19.5 Hz, 2H, 7-H_syn_, 9-H_syn_), 2.44 (d, *J* = 19.5 Hz, 2H, 7-H_anti_, 9-H_anti_), 3.51 (tt, *J* = 8.5/5.9 Hz, 1H, 11-H), 3.70 (s, 2H, NCH_2_Ph), 7.22–7.33 (m 5H, Ph). A signal for the NH proton is not observed in the spectrum. ^13^C NMR (100 MHz, CDCl_3_): δ (ppm) = 21.6 (2C, C-3, C-4), 32.2 (2C, C-2, C-5), 44.7 (2C, C-10, C-12), 47.9 (2C, C-1, C-6), 50.6 (2C, C-7, C-9), 52.9 (1C, NCH_2_Ph), 56.3 (1C, C-11), 127.1 (1C, C-4_Ph_), 128.3 (2C, C-3_Ph_, C-5_Ph_), 128.5 (2C, C-2_Ph_, C-6_Ph_), 140.5 (1C, C-1_Ph_), 219.9 (1C, C=O). Purity (HPLC): 73.8% (*t_R_* = 14.05 min).

#### 5.3.30. 1 -Benzyl-3-(2-methoxy-5-methylphenyl)-1-(*syn*-11-oxo-[4.3.3]propellan-8-yl)urea (*syn*-**21**)

According to the General Procedure A, amine *syn-***20** (0.20 g, 0.70 mmol), 2-methoxy-5-methylphenyl isocyanate (0.14 g, 0.84 mmol) and Bu_2_Sn(OAc)_2_ (26 mg, 0.07 mmol) were dissolved in THF (15 mL) and the mixture was stirred at rt for 18 h. The crude product was purified by fc (3 cm, cyclohexane:ethyl acetate = 7:3, 20 mL, R*_f_* = 0.44). Colorless solid, mp 168–170 °C, yield 0.22 g (70%). C_28_H_34_N_2_O_3_ (446.3). Exact mass (APCI): *m/z* = 447.2666 (calcd. 447.2642 for C_28_H_35_N_2_O_3_ [M+H]^+^). FT-IR (ATR, film): ν (cm^−1^) = 3390 (ν *N-H*), 2927 (ν *C-H* aliphatic), 1732 (ν *C=O* ketone), 1658 (ν*C=O* urea), 1597 (δ *N-H*). ^1^H NMR (400 MHz, CDCl_3_): δ (ppm) = 1.38–1.51 (m, 6H, 2-CH_2_(1H), 3-CH_2_, 4-CH_2_, 5-CH_2_(1H)), 1.57–1.66 (m, 2H, 2-CH_2_(1H), 5-CH_2_(1H)), 1.92 (dd, *J* = 13.8/9.1 Hz, 2H, 7-H, 9-H), 2.03 (dd, *J* = 13.8/9.3 Hz, 2H, 7-H, 9-H), 2.25 (s, 3H, CH_3_), 2.28 (d, *J* = 18.4 Hz, 2H, 10-H, 12-H), 2.33 (d, *J* = 18.4 Hz, 2H, 10-H, 12-H), 3.46 (s, 3H, OCH_3_), 4.60 (s, 2H, NCH_2_Ph), 5.33 (p, *J* = 9.2 Hz, 1H, 8-H), 6.59 (d, *J* = 8.2 Hz, 1H, 3-H_Ar_), 6.68 (dd, *J* = 8.2/2.1 Hz, 1H, 4-H_Ar_), 6.95 (s, 1H, NH), 7.31–7.44 (m, 5H, Ph), 7.97 (d, *J* = 2.1 Hz, 1H, 6-H_Ar_). ^13^C NMR (100 MHz, CDCl_3_): δ (ppm) = 21.1 (1C, CH_3_), 21.8 (2C, C-3, C-4), 33.4 (2C, C-2, C-5), 40.8 (2C, C-7, C-9), 46.4 (2C, C-1, C-6), 47.5 (1C, NCH_2_Ph), 50.0 (2C, C-10, C-12), 53.8 (1C, C-8), 55.8 (1C, OCH_3_), 109.9 (1C, C-3_Ar_), 119.7 (1C, C-6_Ar_), 122.2 (1C, C-4_Ar_), 126.5 (1C, C-4_Ph_), 127.9 (4C, C-2_Ph_, C-3_Ph_, C-5_Ph_, C-6_Ph_), 128.8 (1C, C-1_Ar_), 130.7 (1C, C-5_Ar_), 137.7 (1C, C-1_Ph_), 145.7 (1C, C-2_Ar_), 156.1 (1C,C=O urea), 218.6 (1C,C=O ketone). Purity (HPLC): 95.0% (*t_R_* = 22.67 min).

#### 5.3.31. 1 -Benzyl-3-(2-methoxy-5-methylphenyl)-1-(*anti*-11-oxo-[4.3.3]propellan-8-yl)urea (*anti*-**21**)

According to the General Procedure A, amine *anti-***20** (0.22 g, 0.78 mmol), 2-methoxy-5-methylphenyl isocyanate (0.15 g, 0.94 mmol) and Bu_2_Sn(OAc)_2_ (27 mg, 0.08 mmol) were dissolved in THF (20 mL) and the mixture was stirred at rt for 18 h. The crude product was purified by fc (3 cm, cyclohexane:ethyl acetate = 7:3, 20 mL, R*_f_* = 0.40). Pale yellow solid, mp 133–136 °C, yield 0.28 g (82%). C_28_H_34_N_2_O_3_ (446.3). Exact mass (APCI): *m/z* = 447.2639 (calcd. 447.2642 for C_28_H_35_N_2_O_3_ [M+H]^+^). FT-IR (ATR, film): ν (cm^−1^) = 3425 (ν *N-H*), 2935 (ν *C-H* aliphatic), 1735 (ν*C=O* ketone), 1654 (ν *C=O* urea), 1597 (δ *N-H*). ^1^H NMR (400 MHz, CDCl_3_): δ (ppm) = 1.35–1.52 (m, 6H, 2-CH_2_(1H), 3-CH_2_, 4-CH_2_, 5-CH_2_(1H)), 1.58–1.66 (m, 2H, 2-CH_2_(1H), 5-CH_2_(1H)), 1.81 (dd, *J* = 13.5/9.6 Hz, 2H, 7-H_anti_, 9-H_anti_), 2.15 (dd, *J* = 13.5/9.1 Hz, 2H, 7-H_syn_, 9-H_syn_), 2.26 (s, 3H, CH_3_), 2.32 (s, 4H, 10-CH_2_, 12-CH_2_), 3.45 (s, 3H, OCH_3_), 4.51 (s, 2H, NCH_2_Ph), 5.33 (q, *J* = 9.4 Hz, 1H, 8-H), 6.59 (d, *J* = 8.2 Hz, 1H, 3-H_Ar_), 6.68 (dd, *J* = 8.2/1.6 Hz, 1H, 4-H_Ar_), 6.94 (s, 1H, NH), 7.29–7.43 (m, 5H, Ph), 7.97 (d, *J* = 2.1 Hz, 1H, 6-H_Ar_). ^13^C NMR (100 MHz, CDCl_3_): δ (ppm) = 21.2 (1C, CH_3_), 21.5 (2C, C-3, C-4), 32.7 (2C, C-2, C-5), 41.1 (2C, C-7, C-9), 46.7 (2C, C-1, C-6), 48.2 (1C, NCH_2_Ph), 51.1 (2C, C-10, C-12), 55.2 (1C, C-8), 55.9 (1C, OCH_3_), 110.0 (1C, C-3_Ar_), 119.7 (1C, C-6_Ar_), 122.3 (1C, C-4_Ar_), 126.5 (1C, C-4_Ph_), 128.0 9 (4C, C-2_Ph_, C-3_Ph_, C-5_Ph_, C-6_Ph_), 128.9 (1C, C-1_Ar_), 129.3, 130.8 (1C, C-5_Ar_), 137.4 (1C, C-1_Ph_), 145.8 (1C, C-2_Ar_), 156.1 (1C,C=O urea), 219.1 (1C,C=O ketone). Purity (HPLC): 80.8% (*t_R_* = 22.66 min).

#### 5.3.32. 1 -Benzyl-1-{(8-*syn*,11-*anti*)-11-hydroxy[4.3.3]propellan-8-yl}-3-(2-methoxy-5-methylphenyl)urea (*syn*,*anti*-**22**) and 1-benzyl-1-{(8-*syn*,11-*syn*)-11-hydroxy[4.3.3]propellan-8-yl}-3-(2-methoxy-5-methylphenyl)urea (*syn*,*syn*-**22**)

NaBH_4_ (30 mg, 0.79 mmol) was added to a solution of the ketone *syn*-**21** (0.35 g, 0.78 mmol) in a mixture of THF and methanol (9:1, 15 mL). The mixture was stirred at rt for 30 min, then water (1 mL) was added and stirred for additional 10 min. After evaporation of the organic solvent under vacuum, ethyl acetate (10 mL) was added. The mixture was washed with NaOH (1 M, 5 mL) and brine (5 mL), filtered, the filtrate was concentrated in vacuo and the residue was purified by fc (4 cm, petroleum ether:ethyl acetate = 8:2 to 6.5:3.5, 20 mL).

*syn,anti***-22** (R*_f_* = 0.38, cyclohexane:ethyl acetate = 7:3): Colorless solid, mp 177–179 °C, yield 0.15 g (43%). C_28_H_36_N_2_O_3_ (448.6). Exact mass (APCI): *m/z* = 449.2870 (calcd. 449.2799 for C_28_H_37_N_2_O_3_ [M+H]^+^ ). FT-IR (ATR, film): ν (cm^−1^) = 3383 (ν *O-H*), 2927 (ν *C-H* aliphatic), 1639 (ν *C=O*_urea_), 1535 (δ*N-H*). ^1^H NMR (400 MHz, CDCl_3_): 1.34–1.44 (m, 8H, 2-CH_2_, 3-CH_2_, 4-CH_2_, 5-CH_2_), 1.67–1.73 (m, 4H, 7-H_anti_, 9-H_anti_, 10-H_anti_, 12-H_anti_), 1.88 (dd, *J* = 12.9/7.4 Hz, 2H, 7-H_syn_, 9-H_syn_), 2.02 (dd, *J* = 13.6/7.0 Hz, 2H, 10-H_syn_, 12-H_syn_), 2.26 (s, 3H, CH_3_), 3.52 (s, 3H, OCH_3_), 4.34 (tt, *J* = 7.0/5,9 Hz, 1H, 11-H), 4.56 (s, 2H, NCH_2_Ph), 5.17 (tt, *J* = 10.9/7.4 Hz, 1H, 8-H), 6.61 (d, *J* = 8.2 Hz, 1H, 3-H_Ar_), 6.68 (ddd, *J* = 8.2/2.1/0.8 Hz, 1H, 4-H_Ar_), 7.06 (s, broad, 1H, NH), 7.26–7.40 (m, 5H, Ph), 8.04 (d, *J* = 2.1 Hz, 1H, 6-H_Ar_). ^13^C NMR (100 MHz, CDCl_3_): δ (ppm) = 18.7 (1C, CH_3_), 21.2 (2C, C-3, C-4), 32.9 (2C, C-2, C-5), 41.2 (2C, C-7, C-9), 46.8 (1C, NCH_2_Ph), 47.3 (2C, C-1, C-6), 50.2 (2C, C-10, C-12), 52.7(1C, C-8), 55.8 (1C, OCH_3_), 71.8 (1C, C-11), 109.9 (1C, C-3_Ar_), 119.6 (1C, C-6_Ar_), 122.0 (1C, C-4_Ar_), 126.6 (1C, C-4_Ph_), 127.5 (2C, C-3_Ph_, C-5_Ph_), 128.9 (2C, C-2_Ph_, C-6_Ph_), 129.0(1C, C-1_Ar_), 130.7 (1C, C-5_Ar_), 138.5 (1C, C-1_Ph_), 145.7 (1C, C-2_Ar_), 156.1 (1C, C=O). Purity (HPLC): 94.9% (*t_R_* = 22.69 min).

*syn,syn***-22** (R*_f_* = 0.36, cyclohexane:ethyl acetate = 7:3): Pale yellow solid, mp 148–151 °C, yield 0.10 g (28%). C_28_H_36_N_2_O_3_ (448.6). Exact mass (APCI): *m/z* = 449.2770 (calcd. 449.2799 for C_28_H_37_N_2_O_3_ [M+H]^+^). FT-IR (ATR, film): ν (cm^−1^) = 3437 (ν *O-H*), 2931 (ν *C-H* aliphatic), 1643 (ν *C=O*_urea_), 1535 (δ *N-H*). ^1^H NMR (400 MHz, CDCl_3_): δ (ppm) = 1.39–1.51 (m, 8H, 2-CH_2_, 3-CH_2_, 4-CH_2_, 5-CH_2_), 1.62–1.82 (m, 6H, 7-CH_2_, 9-CH_2,_ 10-H_syn_, 12-H_syn_), 2.03 (dd, *J* = 13.9/7.0 Hz, 2H, 10-H_anti_, 12-H_anti_), 2.25 (s, 3H, CH_3_), 3.48 (s, 3H, OCH_3_), 4.51 (s, 2H, NCH_2_Ph), 4.54 (tt, *J* = 7.1/6.2, Hz, 1H, 11-H), 5.01 (tt, *J* = 10.6/7.9 Hz, 1H, 8-H), 6.59 (d, *J* = 8.2 Hz, 1H, 3-H_Ar_), 6.68 (dd, *J* = 8.2, 2.1 Hz, 1H, 4-H_Ar_), 7.05 (s, broad, 1H, NH), 7.27–7.42 (m, 5H, Ph), 7.99 (d, *J* = 2.1 Hz, 1H, 6-H_Ar_). ^13^C NMR (100 MHz, CDCl_3_): δ (ppm) = 20.1 (1C, CH_3_), 21.1 (2C; C-3, C-4), 34.0 (2C, C-2, C-5), 42.3 (2C, C-7, C-9), 48.0 (2C, C-1, C-6), 46.9 (1C, NCH_2_Ph) 49.4 (2C, C-10, C-12), 53.6 (1C, C-8), 55.8 (1C, OCH_3_), 72.6 (1C, C-11), 109.9 (1C, C-3_Ar_), 119.7 (1C, C-6_Ar_), 122.2 (1C, C-4_Ar_), 126.7 (1C, C-4_Ph_), 127.8 (2C, C-3_Ph_, C-5_Ph_), 128.8 (2C, C-2_Ph_, C-6_Ph_), 129.0 (1C, C-1_Ar_), 130.7 (1C, C-5_Ar_), 137.8 (1C, C-1_Ph_), 145.8 (1C, C-2_Ar_), 156.0 (1C, C=O). Purity (HPLC): 97.7% (*t_R_* = 22.15 min).

#### 5.3.33. 1 -Benzyl-1-{(8-*anti*,11-*anti*)-11-hydroxy[4.3.3]propellan-8-yl}-3-(2-methoxy-5-methylphenyl)urea (*anti*,*anti*-**22**) and 1-benzyl-1-{(8-*anti*,11-*syn*)-11-hydroxy[4.3.3]propellan-8-yl}-3-(2-methoxy-5-methylphenyl)urea (*anti*,*syn*-**22**)

NaBH_4_ (9 mg, 0.24 mmol) was added to a solution of the ketone *anti*-**21** (0.15 g, 0.23 mmol) in a mixture of THF and methanol (9:1, 15 mL). The mixture was stirred at rt for 30 min, then water (1 mL) was added and stirred for additional 10 min. After evaporation of the organic solvent under vacuum, ethyl acetate (10 mL) was added. The mixture was washed with NaOH (1 M, 5 mL) and brine (5 mL), filtered, the filtrate was concentrated in vacuo and the residue was purified by fc (4 cm, petroleum ether:ethyl acetate = 8:2 to 6.5:3.5, 20 mL).

*anti,anti***-22** (R*_f_* = 0.23, cyclohexane:ethyl acetate = 7:3): Colorless solid, mp 126–128 °C, yield 50 mg (33%). C_28_H_36_N_2_O_3_ (448.6). Exact mass (APCI): *m/z* = 449.2793 (calcd. 449.2799 for C_28_H_37_N_2_O_3_ [M+H]^+^ ). FT-IR (ATR, film): ν (cm^−1^) = 3394 (ν*O-H*), 2924 (ν*C-H* aliphatic), 1635 (ν *C=O*_urea_), 1535 (δ *N-H*). ^1^H NMR (400 MHz, CDCl_3_): δ (ppm) = 1.24–1.54 (m, 8H, 2-CH_2_, 3-CH_2_, 4-CH_2_, 5-CH_2_), 1.67 (dd, *J* = 14.0/4.6 Hz, 2H, 10-H_anti_, 12-H_anti_), 1.95 (m, 4H, 7-CH_2_, 9-CH_2_), 2.07 (dd, *J* = 14.0/7.6 Hz, 2H, 10-H_syn_, 12-H_syn_), 2.25 (s, 3H, CH_3_), 3.45 (s, 3H, OCH_3_), 4.46 (tt, *J* = 7.6/4.6 Hz, 1H, 11-H), 4.57 (s, 2H, NCH_2_Ph), 5.25 (p, *J* = 9.5 Hz, 1H, 8-H), 6.58 (d, *J* = 8.2 Hz, 1H, 3-H_Ar_), 6.66 (dd, *J* = 7.9/1.3 Hz, 1H, 4-H_Ar_), 6.96 (s, 1H, NH), 7.26–7.40 (m, 5H, Ph), 8.01 (d, *J* = 2.1 Hz, 1H, 6-H_Ar_). ^13^C NMR (100 MHz, CDCl_3_): δ (ppm) = 20.8 (1C, CH_3_), 21.1 (2C, C-3, C-4), 33.1 (2C, C-2, C-5), 43.1 (2C, C-7, C-9), 46.9 (1C, NCH_2_Ph), 48.4 (2C, C-10, C-12), 49.3 (2C, C-1, C-6), 55.0 (1C, C-8), 55.8 (1C, OCH_3_), 73.3 (1C, C-11), 109.9 (1C, C-3_Ar_), 119.6 (1C, C-6_Ar_), 122.0 (1C, C-4_Ar_), 126.6 (1C, C-4_Ph_), 127.6 (2C, C-3_Ph_, C-5_Ph_), 129.0 (2C, C-2_Ph_, C-6_Ph_), 129.1 (1C, C-1_Ar_), 130.7 (1C, C-5_Ar_), 138.1 (1C, C-1_Ph_), 145.7 (1C, C-2_Ar_), 156.1 (1C, C=O). Purity (HPLC): 95.5% (*t_R_* = 21.94 min).

*anti,syn***-22** (R*_f_* = 0.26, cyclohexane:ethyl acetate = 7:3): Pale yellow solid, mp 149–151 °C, yield 58 mg (38%). C_28_H_36_N_2_O_3_ (448.6). Exact mass (APCI): *m/z* = 449.2782 (calcd. 449.2799 for C_28_H_37_N_2_O_3_ [M+H]^+^ ). FT-IR (ATR, film): ν (cm^−1^) = 3379 (ν *O-H*), 2927 (ν *C-H* aliphatic), 1643 (ν *C=O*_urea_), 1531 (δ *N-H*). ^1^H NMR (400 MHz, CDCl_3_): δ (ppm) = 1.46–1.56 (m, 8H, 2-CH_2_, 3-CH_2_, 4-CH_2_, 5-CH_2_), 1.59 (dd, *J* = 12.9/10.9 Hz, 2H, 7-H_anti_, 9-H_anti_), 1.69 (dd, *J* = 13.5/6.4 Hz, 2H, 10-H_syn_, 12-H_syn_), 1.89 (d, *J* = 13.0/8.0 Hz, 2H, 7-H_syn_, 9-H_syn_), 1.94 (dd, *J* = 13.5/7.2 Hz, 2H, 10-H_anti_, 12-H_anti_), 2.25 (m, 3H, CH_3_), 3.47 (s, 3H, OCH_3_), 4.39 (p, *J* = 6.9 Hz, 1H, 11-H), 4.50 (s, 2H, NCH_2_Ph), 5.08 (tt, *J* = 10.7/7.9 Hz, 1H, 8-H), 6.59 (d, *J* = 8.2 Hz, 1H, 3-H_Ar_), 6.67 (ddd, *J* = 8.2/2.1/0.8 Hz, 1H, 4-H_Ar_), 6.97 (s, broad, 1H, NH), 7.27–7.41 (m, 5H, Ph), 8.00 (d, *J* = 2.0 Hz, 1H, 6-H_Ar_). ^13^C NMR (100 MHz, CDCl_3_): δ (ppm) = 19.6 (1C, CH_3_), 21.1 (2C, C-3, C-4), 32.3 (2C, C-2, C-5), 43.7 (2C, C-7, C-9), 46.7 (1C, NCH_2_Ph), 47.6 (2C, C-10, C-12), 47.9 (2C, C-1, C-6), 53.9 (1C, C-8), 55.8 (1C, OCH_3_), 71.5 (1C, C-11), 109.9 (1C, C-3_Ar_), 119.6 (1C, C-6_Ar_), 122.0 (1C, C-4_Ar_), 126.5 (1C, C-4_Ph_), 127.7 (2C, C-3_Ph_, C-5_Ph_), 128.9 (2C, C-2_Ph_, C-6_Ph_), 129.0 (1C, C-1_Ar_), 130.7(1C, C-5_Ar_), 138.1(1C, C-1_Ph_), 145.7 (1C, C-2_Ar_), 156.1 (1C,C=O). Purity (HPLC): 93.5% (*t_R_* = 21.57 min).

#### 5.3.34. 1 -Phenyl-3-(*syn*- and *anti*-[4.3.3]propellan-8-yl)urea (**23a**)

According to General Procedure A, propellanamine **16** (90 mg, 0.50 mmol), phenyl isocyanate (72 mg, 0.6 mmol) and Bu_2_Sn(OAc)_2_ (35 mg, 0.1 mmol) were dissolved in THF (5 mL) and the mixture was stirred at rt for 30 h. The crude product was purified by fc (1 cm, cyclohexane:ethyl acetate = 8:2, 5 mL, R*_f_* = 0.36) to obtain a mixture of diastereoisomeric urea *syn-***23a** and *anti*-**23a** as a brown solid, mp 134–138 °C, yield 30 mg (22%). C_19_H_26_N_2_O (298.4). MS (ESI): *m/z* = 299 [M+H]^+^. Exact mass (APCI): *m/z* = 299.2131 (calcd. 299.2118 for C_19_H_27_N_2_O [M+H]^+^). FT-IR (ATR, film): ν (cm^−1^) = 3429 (ν *N-H*), 2927 (ν *C-H* aliphatic), 1647 (ν *C=O*), 1597 (δ*N-H*). ^1^H NMR (600 MHz, CDCl_3_): δ (ppm) = 1.30–1.35 (m, 4H, 2-CH_2_, 5-CH_2_), 1.37–1.44 (m, 5H, 3-CH_2_, 4-CH_2_, 7-CH_2_(0.5H), 9-CH_2_(0.5H)), 1.48 (dd, *J* = 13.3/7.3 Hz, 2 × 0.5H, 7-H, 9-H), 1.51–1.54 (m, 2H, 10-CH_2_(1H), 12-CH_2_(1H)), 1.57–1.66 (m, 3H, 10-CH_2_(1H), 11-CH_2_(1H), 12-CH_2_(1H)), 1.68–1.75 (m, 1H, 11-CH_2_(1H)), 2.03 (dd, *J* = 13.4/8.3 Hz, 2 × 0.5H, 7-H, 9-H), 2.11 (dd, *J* = 13.4/8.7 Hz, 2 × 0.5H, 7-H, 9-H), 4.25–4.31 (m, 2 × 0.5H, 8-H), 7.07–7.16 (m, 1H, 4-H_phenyl_), 7.26–7.28 (m, 2H, 3H_phenyl_, 5-H_phenyl_), 7.29–7.33 (M, 2H, 2-H_phenyl_, 6-H_phenyl_). Signals for the NH protons are not observed in the spectrum. ^13^C NMR (150 MHz, CDCl_3_): δ (ppm) = 20.8, 21.0 (2C, C-3, C-4), 21.2, 21.9 (2 × 0.5C, C-11), 31.7, 32.5 (2C, C-2, C-5), 37.6, 38.5 (2C, C-10, C-12), 46.0, 45.5 (2C, C-7, C-9), 49.4 (2C, C-1, C-6), 50.1, 50.7 (2 × 0.5C, C-8), 120.8, 121,7 (2C, C-3_phenyl_, C-5_phenyl_), 129.2 (1C, C-4_phenyl_), 129.6, 129.9 (2C, C-2_phenyl_, C-6_phenyl_), 137.3 (1C, C-1_phenyl_), 156.3 (1C, C=O). *syn*-**23a**:*anti*-**23a** = 1:1. Purity (HPLC): 97.4% (*t_R_* = 21.03 min).

#### 5.3.35. 1 -Cyclohexyl-3-(*syn*- and *anti*-[4.3.3]propellan-8-yl)urea (**23b**)

According to General Procedure A, propellanamine **16** (90 mg, 0.50 mmol), cyclohexyl isocyanate (75 mg, 0.6 mmol) and Bu_2_Sn(OAc)_2_ (35 mg, 0.1 mmol) were dissolved in THF (5 mL) and the mixture was stirred at rt for 24 h. The crude product was purified by fc (1 cm, cyclohexane:ethyl acetate = 7:3, 5 mL, R*_f_* = 0.40) to obtain a mixture of diastereoisomeric urea *syn-***23b** and *anti*-**23b** as a colorless solid, mp 187–193 °C, yield 73 mg (47%). C_19_H_32_N_2_O (304.5). MS (ESI): *m/z* = 305 [M+H]^+^. Exact mass (APCI): *m/z* = 305.2587 (calcd. 305.2587 for C_19_H_27_N_2_O [M+H]^+^). FT-IR (ATR, film): ν (cm^−1^) = 3302 (ν *N-H*), 2924 (ν *C-H* aliphatic), 1616 (ν *C=O*), 1558 (δ *N-H*). ^1^H NMR (600 MHz, CDCl_3_): δ (ppm) = 1.10–1.18 (m, 2H, 2-CH_2Cy_(1H), 6-CH_2Cy_(1H)), 1.31–1.44 (m, 11H, 2-CH_2_, 3-CH_2_, 4-CH_2_, 5-CH_2_, 7-CH_2_(0.5H), 9-CH_2_(0.5H), 4-CH_2Cy_), 1.48 (d, *J* = 13.5/7.0 Hz, 2 × 0.5 H, 7-H, 9-H), 1.51–1.65 (m, 7H, 10-CH_2_, 11-CH_2_(1H), 12-CH_2_, 3-CH_2Cy_(1H), 5-CH_2Cy_(1H)), 1.68–176 (m, 3H, 11-CH_2_(1H), 3-CH_2Cy_(1H), 5-CH_2Cy_(1H)), 1.92 (m, 2H, 2-CH_2Cy_(1H), 6-CH_2Cy_(1H)) 2.00 (dd, *J* = 13.3/8.4 Hz, 2 × 0.5H, 7-H, 9-H), 2.08 (dd, *J* = 13.3/8.6 Hz, 2 × 0.5H, 7-H, 9-H), 3.51 (tt, *J* = 9.3/4.8 Hz, 2 × 0.5H, 1-H_Cy_), 4.09–4.15 (m, 2 × 0.5H, 8-H). Signals for the NH protons are not observed in the spectrum. ^13^C NMR (150 MHz, CDCl_3_): δ (ppm) = 20.8, 20.9 (2C, C-3, C-4), 21.6 (1C, C-11), 24.8, 25.5 (2C, C-3_Cy_, C-5_Cy_), 26.9 (1C, C-4_Cy_), 31.9, 32.3 (2C, C-2, C-5), 33.8 (2C, C-2_Cy_, C-6_Cy_), 37.9, 38.2 (2C, C-10, C-12), 45.5, 46.0 (2C, C-7, C-9), 49.8 (1C, C-1_Cy_), 50.0 (1C, C-8), 50.5 (2C, C-1, C-6), 157.3 (1C, C=O). *syn*-**23b**:*anti*-**23b** = 1:1. Purity (HPLC): 98.3% (*t_R_* = 21.62 min).

#### 5.3.36. 1 -(2-Methoxy-5-methylphenyl)-3-(*syn*- and *anti*-[4.3.3]propellan-8-yl)urea (**23c**)

According to General Procedure A, propellanamine **16** (90 mg, 0.50 mmol), 2-methoxy-5-methylphenyl isocyanate (98 mg, 0.6 mmol) and Bu_2_Sn(OAc)_2_ (35 mg, 0.1 mmol) were dissolved in THF (5 mL) and the mixture was stirred at rt for 30 h. The crude product was purified by fc (1 cm, cyclohexane:ethyl acetate = 7:3, 5 mL, R*_f_* = 0.55) to obtain a mixture of diastereoisomeric urea *syn-***23c** and *anti*-**23c** as a beige solid, mp 215–220 °C, yield 160 mg (91%). C_21_H_30_N_2_O_2_ (342.5). MS (ESI): *m/z* = 343 [M+H]^+^. Exact mass (APCI): *m/z* = 343.2387 (calcd. 343.2380 for C_21_H_31_N_2_O_2_ [M+H]^+^). FT-IR (ATR, film): ν (cm^−1^) = 3332 (ν *N-H*), 2927 (ν*C-H* aliphatic), 1639 (ν *C=O*), 1546 (δ *N-H*). ^1^H NMR (600 MHz, CDCl_3_): δ(ppm) = 1.32–1.46 (m, 9H, 2-CH_2_, 3-CH_2_, 4-CH_2_, 5-CH_2_, 7-CH_2_(0.5H), 9-CH_2_(0.5H)), 1.45 (dd, *J* = 13.3/7.3 Hz, 2 × 0.5H, 7-H, 9-H), 1.53–1.74 (m, 6H, 10-CH_2_, 11-CH_2_, 12-CH_2_), 2.06 (dd, *J* = 13.3/8.4 Hz, 2 × 0.5H, 7-H, 9-H), 2.14 (dd, *J* = 13.3/8.6 Hz, 2 × 0.5H, 7-H, 9-H), 2.27 (s, 3H, CH_3_), 3.81 (s, 3H, OCH_3_), 4.28–4.31 (m, 2 × 0.5H, 8-H), 6.72 (d, *J* = 8.2 Hz, 1H, 3-H_Ar_), 6.76 (d, *J* = 8.0 Hz, 1H, 4-H_Ar_), 6.80 (s, broad, 1H, NH), 7.90 (s, 1H, 6-H_Ar_). ^13^C NMR (150 MHz, CDCl_3_): δ (ppm) = 20.9, 21.1 (2C, C-3, C-4), 21.1 (1C, CH_3_), 21.2, 21.9 (2 × 0.5C, C-11), 31.8, 32.5 (2C, C-2, C-5), 37.6, 38.6 (2C, C-10, C-12), 45.6, 46.2 (2C, C-7, C-9), 49.2, 49.9 (2 × 0.5C, C-8), 50.1, 50.6 (2C, C-1, C-6), 55.9 (1C, OCH_3_), 110.1 (1C, C-3_Ar_), 120.2 (1C, C-6_Ar_), 122.5 (1C, C-4_Ar_), 128.6 (1C, C-1_Ar_), 130.9 (1C, C-5_Ar_), 146.0 (1C, C-2_Ar_), 155.2 (1C,C=O). *syn*-**23c**:*anti*-**23c** = 1:1. Purity (HPLC): 93.8% (*t_R_* = 22.21 min).

#### 5.3.37. 1 -(3,4-Difluorophenyl)-3-(*syn*- and *anti*-[4.3.3]propellan-8-yl)urea (**23d**)

According to the General Procedure A, propellanamine **16** (90 mg, 0.50 mmol), 3,4-difluorophenyl isocyanate (93 mg, 0.6 mmol) and Bu_2_Sn(OAc)_2_ (35 mg, 0.1 mmol) were dissolved in THF (5 mL) and the mixture was stirred at rt for 48 h. The crude product was purified by fc (1 cm, cyclohexane:ethyl acetate = 7:3, 5 mL, R*_f_* = 0.56) to obtain a mixture of diastereoisomeric urea *syn-***23d** and *anti*-**23d** as a pale orange solid, mp 168–175 °C, yield 100 mg (63%). C_19_H_24_F_2_N_2_O (334.4). MS (ESI): *m/z* = 335 [M+H]^+^. Exact mass (APCI): *m/z* = 335.1932 (calcd. 335.1929 for C_19_H_25_F_2_N_2_O [M+H]^+^ ). FT-IR (ATR, film): ν (cm^−1^) = 3325 (ν *N-H*), 2927 (ν *C-H* aliphatic), 1643 (ν *C=O*), 1558 (δ *N-H*). ^1^H NMR (600 MHz, CDCl_3_): δ(ppm) = 1.27–1.43 (m, 9H, 2-CH_2_, 3-CH_2_, 4-CH_2_, 5-CH_2_, 7-CH_2_(0.5H), 9-CH_2_(0.5)), 1.45–1.64 (m, 6H, 7-CH_2_(0.5H), 9-CH_2_(0.5H), 10-CH_2_, 11-CH_2_(1H), 12-CH_2_), 1.67.172 (m, 1H, 11-CH_2_(1H)), 2.01 (dd, *J* = 13.4/8.4 Hz, 2 × 0.5H, 7-H, 9-H), 2.08 (dd, *J* = 13.8/8.7 Hz, 2 × 0.5H, 7-H, 9-H), 4.22–4.27 (m, 2 × 0.5H, 8-H), 6.89–6.91 (m, 1H, 6-H_Ar_), 6.99–7.06 (m, 1H, 5-H_Ar_), 7.27–7.33 (m, 1H, 2-H_Ar_). Signals for the NH protons are not observed. ^13^C NMR (150 MHz, CDCl_3_): δ (ppm) = 20.9, 21.0 (2C, C-3, C-4), 21.1, 21.8 (2 × 0.5C, C-11), 31.7, 32.5 (2C, C-2, C-5), 37.5, 38.3 (2C, C-10, C-12), 45.5, 46.0 (2C, C-7, C-9), 49.3, 49.97 (2 × 0.5C, C-8), 50.2, 50.7 (2C, C-1, C-6), 110.0 (d, *J* = 19.3 Hz, 1C, C-2_Ar_), 116.0 (d, *J* = 3.9 Hz, 1C, C-6_Ar_), 117.4 (d, *J* = 18.1 Hz, 1C, C-5_Ar_), 135.1 (1C, C-1_Ar_), 150.3 (d, *J* = 260.6 Hz, 2C, C-3_Ar_, C-4_Ar_), 155.9 (1C, C=O). *syn*-**23d**:*anti*-**23d** = 1:1. Purity (HPLC): 98.2% (*t_R_* = 21.73 min).

#### 5.3.38. (11′-*syn* and 11′-*anti*)-Spiro ([1,3]dioxolane-2,8′-[4.3.3]propellan)-11′-amine (**24**)

Pd(OH)_2_/C (20%, 70 mg) was added to a solution of the benzylpropellanamines **14** (0.7 g, 2.14 mmol) and ammonium formate (0.54 g, 8.56 mmol) in methanol (20 mL). The mixture was heated to reflux for 3 h. After evaporation of the solvent under vacuum, ethyl acetate (30 mL) was added. The mixture was washed with NaOH (1 M, 10 mL) and brine (10 mL), filtered, the filtrate was concentrated in vacuo and the residue was purified by fc (3 cm, ethyl acetate:methanol = 1:1, 20 mL, R*_f_* = 0.11(cyclohexane : ethyl acetate : methanol = 6 : 3 : 1)). Pale yellow oil, yield 0.38 g (75%). C_14_H_22_NO_2_ (237.3). MS (ESI): *m/z* = 238 [M+H]^+^. Exact mass (APCI): *m/z* = 238.1834 (calcd. 238.1802 for C_14_H_24_NO_2_ [M+H]^+^ ). FT-IR (ATR, film): ν (cm^−1^) = 2927 (aliphatic ν *C-H*), 1558 (ν *N-H_2_*). ^1^H NMR (600 MHz, CDCl_3_): δ (ppm) = 1.28–1.58 (m, 8H, 2′-CH_2_, 3′-CH_2_, 4′-CH_2_, 5′-CH_2_), 1,63 (dd, *J* = 13.6/6.9 Hz, 2 × 0.5H, 10′-H, 12′-H), 1.71 (dd, *J* = 13.6/7.5 Hz, 2 × 0.5 H, 10′-H, 12′-H), 1.82 (d, *J* = 14.3 Hz, 2H, 7′-H, 9′-H), 1.93 (d, *J* = 14.3 Hz, 2H, 7′-H, 9′-H), 2.05 (dd, *J* = 13.6/8.8 Hz, 2 × 0.5H, 10′-H, 12′-H), 2.10 (dd, *J* = 13.6/8.7 Hz, 2 × 0.5H, 10′-H, 12′-H), 3.61–3.67 (m, 0.5 H, 11′-H), 3.68–3.71 (m, 0.5 H, 11′-H), 3.80 (m, 4H, OCH_2_CH_2_O). A signal for the NH_2_ protons is not observed in the spectrum.

#### 5.3.39. 1 -(3,4-Difluorophenyl)-3-(*syn*- and *anti*-spiro[1,3]dioxolane-2,8’-[4.3.3]propellan-11’-yl)urea

According to General Procedure A, amine **24** (0.35 g, 1.47 mmol), 3,4-difluorophenyl isocyanate (0.27 g, 1.76 mmol) and Bu_2_Sn(OAc)_2_ (52 mg, 0.15 mmol) were dissolved in THF (15 mL) and the mixture was stirred at rt for 48 h. The crude product was purified by fc (3 cm, cyclohexane:ethyl acetate = 7:3, 20 mL, R*_f_* = 0.42) to obtain 0.55 g of a mixture of diastereoisomeric urea and an impurity with the same R*_f_* value. This mixture was used for the next reaction step without further purification.

#### 5.3.40. 1-(3,4-. Difluorophenyl)-3-(8-syn-11-oxo[4.3.3]propellan-8-yl)urea (*syn*-**25**) and 1-(3,4-Difluorophenyl)-3-(8-anti-11-oxo[4.3.3]propellan-8-yl)urea (*anti*-**25**)

The mixture obtained above (0.50 g, 1.37 mmol) and *p*-toluenesulfonic acid monohydrate (26 mg, 0.01 mmol) were dissolved in acetone (20 mL) and the mixture was heated to 60 °C for 2 h. The solvent was removed in vacuo and the residue was purified by fc (3 cm, cyclohexane:ethyl acetate = 9:1 to 6:4, 20 mL).

*syn*-**25** (R*_f_* = 0.23, cyclohexane:ethyl acetate = 7:3): Pale yellow solid, mp 161–173 °C, yield 0.20 g (42%). C_19_H_22_F_2_NO_2_ (348.4). MS (ESI): *m/z* = 349 [M+H]^+^. Exact mass (APCI): *m/z* = 349.1741 (calcd. 349.1722 for C_19_H_23_F_2_N_2_O_2_ [M+H]^+^). FT-IR (ATR, film): ν (cm^−1^) = 3340 (ν *N-H*), 2931 (ν *C-H* aliphatic), 1735 (ν *C=O*_ketone_), 1654 (ν *C=O*_urea_), 1550 (δ *N-H*). ^1^H NMR (400 MHz, CDCl_3_): δ (ppm) = 1.40–1.58 (m, 8H, 2-CH_2_, 3-CH_2_, 4-CH_2_, 5-CH_2_,), (1.75 (dd, *J* = 14.1/5.5 Hz, 2H, 7-H_syn_, 9-H_syn_), 2.15 (d, *J* = 18.8 Hz, 2H, 10-H, 12-H), 2.21 (dd, *J* = 14.1/9.2 Hz, 2H, 7-H_anti_, 9-H_anti_), 2.26 (d, *J* = 18.8 Hz, 2H, 10-H, 12-H), 4.35–4.44 (m, 1H, 8-H), 6.84–6.94 (m, 1H, 6-H_Ar_), 7.01–7.08 (m, 1H, 5-H_Ar_), 7.29–7.35 (m, 1H, 2-H_Ar_). Signals for the NH protons are not observed in the spectrum. ^13^C NMR (100 MHz, CDCl_3_): δ (ppm) = 21.7 (2C, C-3, C-4), 32.5 (2C, C-2, C-5), 44.8 (2C, C-7, C-9), 47.6 (2C, C-10, C-12), 49.1 (1C, C-8), 49.4 (2C, C-1, C-6), 109.8 (d, *J* = 21.1 Hz, 1C, C-2_Ar_), 115.8 (d, *J* = 8.8 Hz, 1C, C-6_Ar_), 117.5 (d, *J* = 18.1 Hz, 1C, C-5_Ar_), 15.2 (1C, C-1_Ar_), 146.8 (dd, *J* = 244.8/13.0 Hz, 1C, C-4_Ar_), 150.4 (dd, *J* = 247.4/13.3 Hz, 1C, C-3_Ar_), 155.3 (1C, N(C=O)N), 218.5 (1C, C=O). Purity (HPLC): 92–6% (*t_R_* = 18.24 min).

*anti*-**25** (R*_f_* = 0.31, cyclohexane:ethyl acetate = 7:3) Pale yellow solid, mp 100–103 °C, yield 0.17 g (35%). C_19_H_22_F_2_NO_2_ (348.4). MS (ESI): *m/z* = 349 [M+H]^+^. Exact mass (APCI): *m/z* = 349.1683 (349.1722 calcd. for C_19_H_23_F_2_N_2_O_2_ [M+H]^+^). FT-IR (ATR, film): ν (cm^−1^) = 3302 (ν *N-H*), 2927 (ν *C-H* aliphatic), 1728 (ν *C=O*_ketone_), 1658 (ν *C=O*_urea_), 1527 (δ *N-H*). ^1^H NMR (400 MHz, CDCl_3_): δ (ppm) = 1.29–1.54 (m, 10H, 2-CH_2_, 3-CH_2_, 4-CH_2_, 5-CH_2_, 7-CH_2_(1H), 9-CH_2_(1H)), 2.19–2.39 (m, 6H, 7-CH_2_(1H), 9-CH_2_(1H), 10-CH_2_, 12-CH_2_), 4.43–4.52 (m, 1H, 8-H), 6.90–6.94 (m, 1H, 6-H_Ar_), 7.04 (dd, *J* = 9.4/8.9 Hz, 1H, 5-H_Ar_), 7.32–7.32–7.40 (m, 1H, 2-H_Ar_). Signals for the NH protons are not observed in the spectrum. ^13^C NMR (100 MHz, CDCl_3_): δ(ppm) = 21.3 (2C, C-3, C-4), 31.8 (2C, C-2, C-5), 44.9 (2C, C-7, C-9), 47.8 (2C, C-1, C-6), 49.1 (1C, C-8), 50.5 (2C, C-10, C-12), 109.4 (d, *J* = 21.1 Hz, 1C, C-2_Ar_), 115.3 (1C, C-6_Ar_), 117.4 (d, *J* = 18.1 Hz, C-5_Ar_), 135.5 (1C, C-1_Ar_) 146.5 (dd, *J* = 247.4/12.0 Hz, 1C, C-4_Ar_), 150.8 (dd, *J* = 245.6/13.3 Hz, 1C, C-3_Ar_), 155.4 (1C, N(C=O)N), 220.3 (1C, C=O). Purity (HPLC): 98.7% (*t_R_* = 18.43 min).

#### 5.3.41. 1 -(3,4-Difluorophenyl)-3-(8-*syn*-11-*syn*- and 8-*syn*-11-*anti*-11-hydroxy[4.3.3]-propellan-8-yl)urea (*syn*,*syn*-**26** and *syn*,*anti*-**26**)

NaBH_4_ (20 mg, 0.53 mmol) was added to a solution of the ketone *syn*-**25** (0.12 g, 0.34 mmol) in a mixture of THF and methanol (9:1, 10 mL). The mixture was stirred at rt for 30 min, then water (1 mL) was added and the mixture was stirred for additional 10 min. After evaporation of the organic solvent under vacuum, ethyl acetate (10 mL) was added. The mixture was washed with NaOH (1M, 5 mL) and brine (5 mL), dried (Na_2_SO_4_), filtered, the filtrate was concentrated in vacuo and the residue was purified by fc (1 cm, cyclohexane:ethyl acetate = 6:4, 5 mL, R*_f_* = 0.49). The mixture could not be separated. Colorless solid, mp 171–178 °C, yield 0.10 g (83%). C_19_H_24_F_2_NO_2_ (350.4). MS (ESI): *m/z* = 351 [M+H]^+^. Exact mass (APCI): *m/z* = 351.1884 (calcd. 351.1879 for C_19_H_25_F_2_N_2_O_2_ [M+H]^+^ ). FT-IR (ATR, film): ν (cm^−1^) = 3298 (ν *O-H*), 2931 (ν *C-H* aliphatic), 1678 (ν *C=O*_urea_), 1558 (δ *N-H*). ^1^H NMR (400 MHz, DMSO-*d*_6_): δ (ppm) = 1.30–1.48 (m, 10H, 2-CH_2_, 3-CH_2_, 4-CH_2_, 5-CH_2_, 7-CH_2_(1H), 9-CH_2_(1H)), 1.50–1.60 (m, 2H, 10-H, 12-H), 1.77–1.90 (m, 3H, 7-CH_2_(0.5H), 9-CH_2_(0.5H), 10-H, 12-H), 2.01 (dd, *J* = 13.2/8.3 Hz, 2 × 0.5H, 7-H, 9-H), 4.00–4.10 (m, 0.5H, 8-H), 4.14–4.27 (m, 1.5H, 8-H(0.5H), 11-H), 6.95–7.01 (m, 1H, 6-H_Ar_), 7.25 (dd, *J* = 10.7/9.2 Hz, 1H, 5-H_Ar_), 7.61 (dddd, *J* = 13.7/7.5/3.5/.26 Hz, 1H, 2-H_Ar_), 8.40 (s, 1H, NH), 8.41 (s, 1H, NH). ^13^C NMR (100 MHz, DMSO-*d*_6_): δ (ppm) = 19.7, 20.4 (2C, C-3, C-4), 31.9, 32.6 (2C, C-2, C-5), 44.6, 45.5 (2C, C-7, C-9), 47.2, 47.7 (2 × 0.5C, C-8), 47.9, 48.2 (2C, C-10, C-12), 48.4, 48.6 (2C, C-1, C-6), 69.6, 69.9 (2 × 0.5C, C-11), 106.3 (d, *J* = 21.9 Hz, 1C, C-2_Ar_), 113.41 (dd, *J* = 5.7/3.1 Hz, 1C, C-6_Ar_), 117.4 (d, *J* = 17.6 Hz, 1C, C-5_Ar_), 137.7 (dd, *J* = 9.5/2.8 Hz, 1C, C-1_Ar_), 143.8 (dd, *J* = 238.4/12.8 Hz, 1C, C-4_Ar_), 149.0 (dd, *J* = 241.7/12.9 Hz, 1C, C-3_Ar_), 154.4 (1C, N(C=O)N). *syn,syn*-**26**:*syn,anti*-**26** = 1:1. Purity (HPLC): 99.2% (*t_R_* = 18.26 min).

#### 5.3.42. 1 -(3,4-Difluorophenyl)-3-(8-*anti*-11-*anti*-11-hydroxy[4.3.3]propellan-8-yl)urea (*anti*,*anti*-**26**) and 1-(3,4-Difluorophenyl)-3-(8-*anti*-11-*syn*-11-hydroxy[4.3.3]propellan-8-yl)urea (*anti*,*syn*-**26**)

NaBH_4_ (20 mg, 0.53 mmol) was added to a solution of the keto urea *anti*-**25** (0.12 g, 0.34 mmol) in a mixture of THF and methanol (9:1, 10 mL). The mixture was stirred at rt for 30 min, then water (1 mL) was added and the mixture was stirred for additional 10 min. After evaporation of the organic solvent under vacuum, ethyl acetate (10 mL) was added. The mixture was washed with NaOH (1 M, 5 mL) and brine (5 mL), dried (Na_2_SO_4_), filtered, the filtrate was concentrated in vacuo and the residue was purified by fc (1 cm, cyclohexane:ethyl acetate = 7:3–5:5, 5 mL).

*anti,anti***-26** (R*_f_* = 0.22, cyclohexane:ethyl acetate = 7:3): Colorless solid, mp 201–203 °C, yield 32 mg (27%). C_19_H_24_F_2_NO_2_ (350.4). MS (ESI): *m/z* = 351 [M+H]^+^. Exact mass (APCI): *m/z* = 351.1901 (calcd. 351.1879 for C_19_H_25_F_2_N_2_O_2_ [M+H]^+^). FT-IR (ATR, film): ν (cm^−1^) = 3232 (ν *O-H*), 2924 (ν *C-H* aliphatic), 1670 (ν *C=O*_urea_), 1566 (δ *N-H*). ^1^H NMR (400 MHz, CDCl_3_): 1.11–1.25 (m, 8H, 2-CH_2_, 3-CH_2_, 4-CH_2_, 5-CH_2_), 1.57 (dd, *J* = 14.0/4.7 Hz, 2H, 10-H_anti_, 12-H_anti_), 1.62 (dd, *J* = 13.6/6.9 Hz, 2H, 7-H_anti_, 9-H_anti_), 1.98 (dd, *J* = 14.0/8.2 Hz, 2H, 10-H_syn_,, 12-H_syn_), 2.03 (dd, *J* = 13.6/9.1 Hz, 2H, 7-H_syn_, 9-H_syn_), 4.23 (tt, *J* = 9.0/6.9 Hz, 1H, 8-H), 4.34 (tt, *J* = 8.1/4.7 Hz, 1H, 11-H), 6.76–6.81 (m, 1H, 6-H_Ar_), 6.85 (dd, *J* = 9.9/8.7 Hz, 1H, 5-H_Ar_), 7.40 (ddd, *J* = 13.1/7.3/2.5 Hz, 1H, 2-H_Ar_). Signals for the NH protons and the OH proton are not observed in the spectrum. ^13^C NMR (100 MHz, CDCl_3_): δ (ppm) = 21.1 (2C, C-3, C-4), 32.1 (2C, C-2, C-5), 45.9 (2C, C-7, C-9), 47.7 (2C, C-10, C-12), 48.8 (1C, C-8), 50.4 (2C, C-1, C-6), 71.9 (1C, C-11), 107.2 (d, *J* = 21.9 Hz, 1C, C-2_Ar_), 113.1 (dd, *J* = 5.6/3.3 Hz, 1C, C-6_Ar_), 116.5 (d, *J* = 17.9 Hz, 1C, C-5_Ar_), 137.0 (dd, *J* = 9.3/2.8 Hz, C-1_Ar_), 144.9 (dd, *J* = 240.6/13.1 Hz, 1C, C-4_Ar_), 149.8 (dd, *J* = 242.8/13.1 Hz, 1C, C-3_Ar_), 155.2 (1C, N(C=O)N). Purity (HPLC): 96.9% (*t_R_* = 18.49 min).

*anti,syn***-26** (R*_f_* = 0.13, cyclohexane:ethyl acetate = 7:3): Colorless solid, mp 186–189 °C, yield 10 mg (8%). C_19_H_24_F_2_NO_2_ (350.4). MS (ESI): *m/z* = 351 [M+H]^+^. Exact mass (APCI): *m/z* = 351.1907 (calcd. 351.1879 for C_19_H_25_F_2_N_2_O_2_ [M+H]^+^ ). FT-IR (ATR, film): ν (cm^−1^) = 3383 (ν *O-H*), 2924 (ν *C-H* aliphatic), 1654 (δ *C=O*_urea_), 1570 (δ *N-H*). ^1^H NMR (400 MHz, DMSO-*d*_6_): 1.35 (dd, *J* = 13.3/8.1 Hz, 2H, 7-H_anti_, 9-H_anti_), 1.39–1.50 (m, 8H, 2-CH_2_, 3-CH_2_, 4-CH_2_, 5-CH_2_), 1.56 (dd, *J* = 13.4/5.4 Hz, 2H, 10-H_syn_, 12-H_syn_), 1.87 (dd, *J* = 13.6/7.5 Hz, 2H, 10-H_anti_, 12-H_anti_), 1.95 (dd, *J* = 13.3/6.0 Hz, 2H, 7-H_syn_, 9-H_syn_), 4.10 (tt, *J* = 8.1/6.0 Hz, 1H, 8-H), 4.30 (tt, *J* = 7.3/5.4 Hz, 1H, 11-H), 6.34 (d, *J* = 7.5 Hz, 1H, N*H*CONHAr), 6.98 (dddd, *J* = 9.0/4.1/2.5/1.5 Hz, 1H, 6-H_Ar_), 7.25 (dd, *J* = 10.6/9.1 Hz, 1H, 5-H_Ar_), 7.60 (ddd, *J* =13.7/7.5/2.6 Hz, 1H 2-H_Ar_). 8.45 (s, 1H, NHCON*H*Ar). A signal for the OH proton is not observed in the spectrum. ^13^C NMR (100 MHz, DMSO-*d*_6_): δ (ppm) = 19.7 (2C, C-3, C-4), 31.2 (2C, C-2, C-5), 46.2 (2C, C-7, C-9), 47.1 (2C, C-10, C-12), 48.6 (2C, C-1, C-6), 47.7 (1C, C-8), 691 (1C, C-11), 106.3 (d, *J* = 21.8 Hz, 1C, C-2_Ar_), 113.38 (1C, C-6_Ar_), 137.8 (d, *J* = 6.8 Hz, C-1_Ar_), 141.8 (d, *J* = 152.2 Hz, 1C, C-4_Ar_), 149.1 (d, *J* = 242.1 Hz, 1C, C-3_Ar_), 154.5 (1C, N(C=O)N). Purity (HPLC): 93.8% (*t_R_* = 17.82 min).

### 5.4. Computational Details

The starting structure for the σ_1_ receptor was obtained from the RCSB Protein Data Bank (PDB ID 5HK1, https://www.rcsb.org/structure/5HK1 (accessed date: 3 March 2021) [7], of which only the protomer with the more complete sequence was retained for the simulations. The CHARMM-GUI server [54] was used to embed the σ_1_ monomer in a palmitoyl-oleyl-phosphatidyl-choline (POPC, 218 lipid molecules were added) bilayer solvated with explicit TIP3P [55] and water molecules to succeed complete hydration of the membrane and reach a physiological concentration of sodium and chloride ions (0.15 M NaCl). Antechamber program from AMBER20 [56] was used to assign gaff2 [57] atom types to each ligand, while ligand’s partial charges were derived by employing the RESP method offered by the RED server [58]. Docking and classical molecular dynamics simulations on σ_1_ receptor in complex with the new azapropellane derivatives were carried out following a well validated procedure [49,52,59]. Briefly, the system density and volume were relaxed in NPT ensemble maintaining the Berendsen barostat for 20 ns. After this step, 50 ns of unrestrained NVT production simulation was run for each system. Following the MM/PBSA approach [55], each binding free energy values (ΔG) were calculated as the sum of the enthalpic (ΔH) and entropic contributions (-TΔS). The PRBFED analysis was carried out using the molecular mechanics/generalized Boltzmann surface area (MM/GBSA) approach [60] and was based on the same snapshots used in the binding free energy calculation. All images were created by the UCSF Chimera software v1.15 [61], and graphs were produced by GraphPad Prism v8 (GraphPad Software, San Diego, California USA, www.graphpad.com).

### 5.5. X-ray Diffraction

#### 5.5.1. General

Data sets for compounds *syn*-**7**, *anti*-**21**, *syn,anti*-**22**, *anti,syn*-**22** and *syn*-**25** were collected with a Bruker Kappa CCD diffractometer. Programs used: data collection, COLLECT [62], data reduction Denzo-SMN [63]; absorption correction [64]; structure solution SHELXT-2015 [65]; structure refinement SHELXL-2015 [66] and graphics, XP [67]*. R*-values are given for observed reflections, and *w*R^2^ values are given for all reflections.

Exceptions and special features: For compound *syn*-**25** three independent molecules were found in the asymmetric unit. All these three molecules present different groups disordered over two positions. Several restraints (SADI, SAME, ISOR and SIMU) were used in order to improve refinement stability.

#### 5.5.2. X-ray Crystal Structure Analysis of *syn*-**7**

A colorless plate-like specimen of C_21_H_27_NO_4_, approximate dimensions 0.100 mm × 0.200 mm × 0.350 mm, was used for the X-ray crystallographic analysis. The X-ray intensity data were measured. The integration of the data using a monoclinic unit cell yielded a total of 3302 reflections to a maximum θ angle of 67.19° (0.84 Å resolution), of which 3302 were independent (average redundancy 1.000, completeness = 97.6%, R_sig_ = 2.02%) and 3062 (92.73%) were greater than 2σ(F^2^). The final cell constants of a = 6.9404(2) Å, b = 12.2651(4) Å, c = 22.3938(10) Å, β = 98.330(2)° and volume = 1886.15(12) Å^3^ were based upon the refinement of the XYZ-centroids of reflections above 20 σ(I). Data were corrected for absorption effects using the multi-scan method (SADABS). The calculated minimum and maximum transmission coefficients (based on crystal size) are 0.7920 and 0.9330. The structure was solved and refined using the Bruker SHELXTL-2014/7 version Software Package, using the space group *P*2_1_/*c*, with Z = 4 for the formula unit, C_21_H_27_NO_4_. The final anisotropic full-matrix least-squares refinement on F^2^ with 241 variables converged at R1 = 3.98%, for the observed data and wR2 = 9.96% for all data. The goodness-of-fit was 1.054. The largest peak in the final difference electron density synthesis was 0.198 e^−^/Å^3^ and the largest hole was −0.192 e^−^/Å^3^ with an RMS deviation of 0.034 e^−^/Å^3^. On the basis of the final model, the calculated density was 1.259 g/cm^3^ and F(000), 768 e^−^. The hydrogen at N1 atom was refined freely. CCDC number: 2073466.

#### 5.5.3. X-ray Crystal Structure Analysis of *anti*-**21**

A colorless prism-like specimen of C_28_H_34_N_2_O_3_, approximate dimensions 0.060 mm × 0.240 mm × 0.260 mm, was used for the X-ray crystallographic analysis. The X-ray intensity data were measured. The integration of the data using a monoclinic unit cell yielded a total of 4159 reflections to a maximum θ angle of 67.08° (0.84 Å resolution), of which 4159 were independent (average redundancy 1.000, completeness = 96.3%, R_sig_ = 2.90%) and 3509 (84.37%) were greater than 2σ(F^2^). The final cell constants of a = 8.5091(4) Å, b = 15.9061(5) Å, c = 17.8829(6) Å, β = 94.139(3)° and volume = 2414.08(16) Å^3^ were based upon the refinement of the XYZ-centroids of reflections above 20 σ(I). Data were corrected for absorption effects using the multiscan method (SADABS). The calculated minimum and maximum transmission coefficients (based on crystal size) are 0.8530 and 0.9630. The structure was solved and refined using the Bruker SHELXTL Software Package, using the space group *P*2_1_/*n*, with Z = 4 for the formula unit, C_28_H_34_N_2_O_3_. The final anisotropic full-matrix least-squares refinement on F^2^ with 304 variables converged at R1 = 7.05%, for the observed data and wR2 = 20.59% for all data. The goodness-of-fit was 1.041. The largest peak in the final difference electron density synthesis was 0.482 e^−^/Å^3^ and the largest hole was −0.353 e^−^/Å^3^ with an RMS deviation of 0.056 e^−^/Å^3^. On the basis of the final model, the calculated density was 1.229 g/cm^3^ and F(000), 960 e^−^. The hydrogen at N2 atom was refined freely. CCDC number: 2073467.

#### 5.5.4. X-ray Crystal Structure Analysis of *syn,anti-***22**

A colorless plate-like specimen of C_28_H_36_N_2_O_3_, approximate dimensions 0.150 mm × 0.170 mm × 0.270 mm, was used for the X-ray crystallographic analysis. The X-ray intensity data were measured. The integration of the data using a monoclinic unit cell yielded a total of 4152 reflections to a maximum θ angle of 67.18° (0.84 Å resolution), of which 4152 were independent (average redundancy 1.000, completeness = 96.1%, R_sig_ = 2.57%) and 3721 (89.62%) were greater than 2σ(F^2^). The final cell constants of a = 7.5886(3) Å, b = 16.0072(4) Å, c = 20.2378(5) Å, β = 100.377(2)° and volume = 2418.12(13) Å^3^ were based upon the refinement of the XYZ-centroids of reflections above 20 σ(I). Data were corrected for absorption effects using the multi-scan method (SADABS). The calculated minimum and maximum transmission coefficients (based on crystal size) were 0.8490 and 0.9120. The structure was solved and refined using the Bruker SHELXTL Software Package, using the space group *P*2_1_/*c*, with Z = 4 for the formula unit, C_28_H_36_N_2_O_3_. The final anisotropic full-matrix least-squares refinement on F^2^ with 307 variables converged at R1 = 4.36%, for the observed data and wR2 = 11.81% for all data. The goodness-of-fit was 1.031. The largest peak in the final difference electron density synthesis was 0.175 e^−^/Å^3^ and the largest hole was −0.164 e^−^/Å^3^ with an RMS deviation of 0.032 e^−^/Å^3^. On the basis of the final model, the calculated density was 1.232 g/cm^3^ and F(000), 968 e^−^. The hydrogens at N2 and O1 atoms were refined freely. CCDC number: 2073468.

#### 5.5.5. X-ray Crystal Structure Analysis of *anti*,*syn*-22

A colorless needle-like specimen of C_28_H_36_N_2_O_3_, approximate dimensions 0.020 mm × 0.070 mm × 0.270 mm, was used for the X-ray crystallographic analysis. The X-ray intensity data were measured. The integration of the data using a monoclinic unit cell yielded a total of 4164 reflections to a maximum θ angle of 66.92° (0.84 Å resolution), of which 4164 were independent (average redundancy 1.000, completeness = 95.5%, R_sig_ = 3.04%) and 3332 (80.02%) were greater than 2σ(F^2^). The final cell constants of a = 7.9714(2) Å, b = 27.7657(10) Å, c = 11.1029(5) Å, β = 95.285(3)° and volume = 2446.97(15) Å^3^ were based upon the refinement of the XYZ-centroids of reflections above 20 σ(I). Data were corrected for absorption effects using the multiscan method (SADABS). The calculated minimum and maximum transmission coefficients (based on crystal size) are 0.8500 and 0.9880. The structure was solved and refined using the Bruker SHELXTL-2014/7 version Software Package, using the space group *P*2_1_/*c*, with Z = 4 for the formula unit, C_28_H_36_N_2_O_3_. The final anisotropic full-matrix least-squares refinement on F^2^ with 308 variables converged at R1 = 5.00%, for the observed data and wR2 = 13.22% for all data. The goodness-of-fit was 1.031. The largest peak in the final difference electron density synthesis was 0.131 e^−^/Å^3^ and the largest hole was −0.239 e^−^/Å^3^ with an RMS deviation of 0.044 e^−^/Å^3^. On the basis of the final model, the calculated density was 1.218 g/cm^3^ and F(000), 968 e^−^. The hydrogens at N2 and O1 atoms were refined freely. CCDC number: 2073469.

#### 5.5.6. X-ray Crystal Structure Analysis of *syn-*25

A colorless prism-like specimen of C_19_H_22_F_2_N_2_O_2_, approximate dimensions 0.100 mm × 0.140 mm × 0.180 mm, was used for the X-ray crystallographic analysis. The X-ray intensity data were measured. The integration of the data using a triclinic unit cell yielded a total of 9040 reflections to a maximum θ angle of 67.31° (0.84 Å resolution), of which 9040 were independent (average redundancy 1.000, completeness = 96.6%, R_sig_ = 2.87%) and 7415 (82.02%) were greater than 2σ(F^2^). The final cell constants of a = 12.9305(5) Å, b = 13.0504(5) Å, c = 17.0932(4) Å, α = 112.037(2)°, β = 97.638(2)°, γ = 97.0010(10)° and volume = 2603.27(16) Å^3^ were based upon the refinement of the XYZ-centroids of reflections above 20 σ(I). Data were corrected for absorption effects using the multiscan method (SADABS). The calculated minimum and maximum transmission coefficients (based on crystal size) were 0.8630 and 0.9200. The structure was solved and refined using the Bruker SHELXTL Software Package, using the space group *P*-1, with Z = 6 for the formula unit, C_19_H_22_F_2_N_2_O_2_. The final anisotropic full-matrix least-squares refinement on F^2^ with 928 variables converged at R1 = 6.16%, for the observed data and wR2 = 18.77% for all data. The goodness-of-fit was 1.038. The largest peak in the final difference electron density synthesis was 1.261 e^−^/Å^3^ and the largest hole was −0.361 e^−^/Å^3^ with an RMS deviation of 0.044 e^−^/Å^3^. On the basis of the final model, the calculated density was 1.333 g/cm^3^ and F(000), 1104 e^−^. The hydrogens at N1A, N2A, N1B, N2B, N1C and N2C atoms were refined freely. CCDC number: 2073470.

### 5.6. Receptor Binding Studies

The affinity towards σ_1_ and σ_2_ receptors was recorded according to the procedures given in the Appendix A and ref [45,46,47].

## Data Availability

Not applicable.

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
