# Peer review of "Propellanes as Rigid Scaffolds for the Stereodefined Attachment of σ-Pharmacophoric Structural Elements to Achieve σ Affinity"

_ijms, 2021, doi:10.3390/ijms22115685_

Round 1
Reviewer 1 Report
Wünsch et al reported the design and synthesis of 3 sets of propellane derivatives attached to various stereo-defined substituents aiming at obtaining conformationally restricted ligands with potential binding affinity to σ1 receptor. The activity of designed and synthesized compounds was evaluated against sigma 1&2 protein, which revealed the critical structural features which are critical for the binding/inhibitory activity of sigma proteins. Finally, the authors performed an in silico docking study for 2 represented compounds which revealed that the compds bind to the cavity via several hydrophobic and hydrophilic interactions and highlighted the role of the rigid propellane moiety in these interactions. This is very interesting and well designed study. The authors presented a privilege part of synthetic chemistry and stereochemistry. The study is well performed and presents a potential strategy to develop/optimize a rigid ligand. I would recommend the publication of this study after addressing the following concerns:
1- the abstract should be shorten to provide the most important data of this study not discussing detailed activity of compds.
2- please modify the introduction part, its too short, better to provide more detailed about known sigma receptors, selectivity of ligands between sigma 1 and 2.
3- please provide the 2D interactions for compd 9a and anti-5, as it is not clear from the 3D.
4- it would be interesting to show the mode of binding for two isomers, eg syn,anti-5, this would give an essential structural feature for the binding affinity of compds.
5- the authors mentioned that they checked purity by HPLC, was it using chairal columns? please add the chromatogram which show the stereo-purity of compds.
6- the authors mentioned that they have performed 2D-NMR, which is essential to prove the stereo-configuration and purity of synthesized compds, please provide the spectra.
7- for reductive amination of anti-7 and syn-7 , the authors claimed that the ratio of diastereomers obtained was 1:1, how this was confirmed?
8- one disappointing point, that the author designed/synthesized/used/tested diastereomeric mix for compds 4b-u, and even built up a conclusion of activity based on it, which is not right based on their results which showed the critical effect of the stereo-configuration. Accordingly, I think we cannot used this set of compds to make a conclusion about the 4-scaffold.
9- the authors should discuss in depth the SAR study based on the activity of synthesized compds and modelling study, and highlight the structural features that are critical for designing potential rigid binders to σ1 receptor.
10- the authors investigated the activity of synthesized compds against σ1 and σ2 receptors; based on the results the authors should address the way to design a selective binder to σ-receptors?
11- please check the C-NMR for synthesized compds, some peaks are missed.
12- the resolution of C-NMR for compds xxx is very poor, please repeat this analysis with more conc samples or longer scan to the sample.
13- it would be beneficial if the authors presented the binding mode of compd 18, since it showed potential activity.
14- in scheme 4 legend the authors should mention the stereo-selectivity of performed reactions?
15- I'm curious to know, based on this study, would it be possible to design a covalent inhibitor based on the potential activity of obtained compds and based on computational study?
Author Response
Reviewer #1
Wünsch et al reported the design and synthesis of 3 sets of propellane derivatives attached to various stereo-defined substituents aiming at obtaining conformationally restricted ligands with potential binding affinity to σ1 receptor. The activity of designed and synthesized compounds was evaluated against sigma 1&2 protein, which revealed the critical structural features which are critical for the binding/inhibitory activity of sigma proteins. Finally, the authors performed an in silico docking study for 2 represented compounds which revealed that the compds bind to the cavity via several hydrophobic and hydrophilic interactions and highlighted the role of the rigid propellane moiety in these interactions. This is very interesting and well designed study. The authors presented a privilege part of synthetic chemistry and stereochemistry. The study is well performed and presents a potential strategy to develop/optimize a rigid ligand.
Response: Thank you very much for these very positive comments. We are very happy about this evaluation of our manuscript.
I would recommend the publication of this study after addressing the following concerns:
1- the abstract should be shorten to provide the most important data of this study not discussing detailed activity of compds.
Response: The abstract describes at first the concept of the project. Then, structure-affinity relationships are discussed. Finally, the binding mode of the ligands in the binding pocket determined by molecular dynamics simulations is reported. In the new revised (shortened) abstract, we tried to focus on the essential results. With exception of the s1 affinity of the most active compound of this series of compounds (9a, Ki = 17 nM) the exact affinity values have been removed.
2- please modify the introduction part, its too short, better to provide more detailed about known sigma receptors, selectivity of ligands between sigma 1 and 2.
Response: This manuscript is part of a special s issue. Therefore, we decided to write a focused introduction in order to avoid repeating all the s specific aspects detailed in the introduction of the other manuscripts of this issue. However, the introduction has been increased and revised now with respect to selectivity, i.e. the s1/s2 selectivity of the key spirocyclic and bicyclic compounds is detailed in the revised manuscript. The reviewer should keep in mind that we wrote an introduction for the development of s1 receptor ligands, bur not a review about all aspects of s receptors.
3- please provide the 2D interactions for compd 9a and anti-5, as it is not clear from the 3D.
Response: Although the s1 binding pocket has a 3D structure and its 2D visualization does not reflect its real geometry, for reader’s benefit, we added two new panels in Figure 9 (C and D) in which the 2D interactions for compounds 9a and anti-5 are displayed.
4- it would be interesting to show the mode of binding for two isomers, eg syn,anti-5, this would give an essential structural feature for the binding affinity of compds.
Response: Molecular dynamics simulations have been performed exemplarily for the azapropellanes 9a and anti-5. Altogether, the manuscript presents a huge amount of propellane derivatives, which are very heterogeneous. The detailed analysis of different binding modes of diastereomeric carbamates with the azapropellane scaffold is very tough. Therefore, we decided to focus on the molecular dynamics simulations of azapropellanes 9a and anti-5. Moreover, we added molecular dynamics simulations of the cyclohexylmethylamine 18 to the Supporting Information. Compound 18 belongs to a quite different class of propellanes.
5- the authors mentioned that they checked purity by HPLC, was it using chairal columns? please add the chromatogram which show the stereo-purity of compds.
Response: Due to symmetry, most of the compounds are not chiral. Several compounds (e.g. 4a-u, 8-16, 21-26) contain two pseudochiral centers in 8- and 11-position, which results in four achiral diastereomers. We decided to differentiate the diastereomers with the stereodescriptors syn and anti. Alternative stereodescriptors were r and s. We feel that for the class of propellanes the stereodescriptors syn and anti are easier to understand and to correlate with the structures than the stereodescriptors r and s. However, purity was determined by RP-HPLC, chiral HPLC was not necessary for achiral compounds. Some selected HLPC traces have been added to the Supporting Information.
6- the authors mentioned that they have performed 2D-NMR, which is essential to prove the stereo-configuration and purity of synthesized compds, please provide the spectra.
Response: Some selected 2D-NMR spectra have been added to the Supporting Information. If the signals are separated, 1H NMR spectra are sufficient to determine the relative configuration. However, for some compounds the assignment of the relative configuration was supported by 2D-NMR spectra. In order to confirm the stereochemistry unequivocally, several X-ray crystal structures have been recorded. Altogether, the relative configuration of all synthesized compounds has been determined unequivocally.
7- for reductive amination of anti-7 and syn-7 , the authors claimed that the ratio of diastereomers obtained was 1:1, how this was confirmed?
Response: The ratio of diastereomers was determined by integration of characteristic signals in the 1H NMR spectra; in particular, the signals for the protons at 8- and 11-position were integrated. There are only a few samples, which show a ratio different from 1:1 (5:5). In these cases, the ratio is 6:4 or 7:3. The ratios of diastereomers has been corrected in the Tables. Now the ratios given in the Tables correlate with the ratios given in the Experimental Part. Moreover, for the compounds 8-anti-4i, 8-syn-4i, 8-anti-4k and 8-syn-4k, expansions of the crucial regions of the 1H NMR spectra are added to the Supporting Information to show the principle of the determination of the ratio of diastereomers.
8- one disappointing point, that the author designed/synthesized/used/tested diastereomeric mix for compds 4b-u, and even built up a conclusion of activity based on it, which is not right based on their results which showed the critical effect of the stereo-configuration. Accordingly, I think we cannot used this set of compds to make a conclusion about the 4-scaffold.
Response: The reviewer is correct, when addressing the problem of diastereomeric mixtures. It is problematic to discuss structure affinity relationships with diastereomeric mixtures. However, in our case, the affinity for most of the compounds of type 4 is rather low. If a 1:1-mixture has low affinity the two members of the mixture have low affinity as well. Our description of the affinity data is based on this principle. The idea of this part of the manuscript was to come up with an optimal substitution pattern first, i.e. an optimal amino substituent as for granatane derivative 3, and separate the diastereomers for potent compounds afterwards. It has to be noted that the separation of the enantiomers is very time and material consuming. Therefore, the optimal substituents should be defined first.
9- the authors should discuss in depth the SAR study based on the activity of synthesized compds and modelling study, and highlight the structural features that are critical for designing potential rigid binders to σ1 receptor.
Response: The rigid propellane derivatives were derived from the lead compounds shown in Figure 2. A modelling study has not been performed first. In particular, the granatane derivative 3 shows high similarity to the designed compounds. The concept of rigidity represents a general principle in Medicinal Chemistry: If the substituents of a ligand are appropriately preoriented at a rigid scaffold, the entropic penalty due to freezing a particular conformation within the binding pocket of the receptor is minimized.
10- the authors investigated the activity of synthesized compds against σ1 and σ2 receptors; based on the results the authors should address the way to design a selective binder to σ-receptors?
Response: The “naked” compounds 9a and 18 show very high s1 affinity. We could speculate that only one substituent at the “right” ring is necessary for high s1 affinity and that additional substituents at the other rings are detrimental for high s1 affinity. However, this is only a speculation and should not be given in the manuscript.
11- please check the C-NMR for synthesized compds, some peaks are missed.
Response: Dr. Hector Torres-Gmoez and myself have checked all NMR spectra. We think that all signals are present and signals are not missing. Therefore, it would be helpful, if the reviewer would be more specific and gave us some hints, where signals are missing. Nevertheless, all the compounds are symmetric (they do have a symmetry plain), which results in only half of the 13C signals. Equivalent C-atoms of the molecule give only one signal in the 13C NMR spectrum, which reduces the number of signals.
12- the resolution of C-NMR for compds xxx is very poor, please repeat this analysis with more conc samples or longer scan to the sample.
Response: In a few cases, resolution of 13C NMR spectra is poor. However, for these spectra HSQC spectra were recorded to aid the assignment. These spectra are now included for compounds 8-anti-4f, ,4g, -4n, -4o, .4p, -4r). The signals for the quaternary C-atoms of these compounds appear at very similar chemical shifts. After expansion of the spectra and due to analogy and HSQC spectra it was possible to find and assign all signals in the 13C NMR spectra, even if they have very low intensity.
13- it would be beneficial if the authors presented the binding mode of compd 18, since it showed potential activity.
Response: The initial purpose of the modeling studies was to explain the binding mode of the best compound of the series, the azapropellane 9a, and put it in comparison with its corresponding propellanylcarbamate anti-5. However, since the propellanamine 18 is provided with the second best s1 affinity value, we followed the comment of the reviewer. Molecular dynamics simulations were conducted for compound 18. A brief paragraph about the binding mode of 18 was added in the "Computational studies" section and the corresponding Figure S1 is shown in the Supporting Information.
14- in scheme 4 legend the authors should mention the stereo-selectivity of performed reactions?
Response: In Scheme 4 a 1:1-mixture of diastereomeric primary amines 16 was employed. Therefore, the ureas 23 were obtained as 1:1-mixtures as well.
15- I'm curious to know, based on this study, would it be possible to design a covalent inhibitor based on the potential activity of obtained compds and based on computational study?
Response: It should be possible to design a covalent ligand based on the propellane scaffold. In particular, the propellane scaffold could be decorated in various positions with appropriate functional groups allowing the covalent interaction with the s receptor protein. However, we have a few other compound lasses in hand showing very high s1 affinity or s2 affinity and high selectivity for a particular s receptor subtype. Using these compounds as starting point appears to be a better strategy to come up with a covalent s ligand.
Reviewer 2 Report
In the article, “Propellanes as rigid scaffolds for the stereodefined attachment of σ-pharmacophoric structural elements” Torres-Gomez and colleagues report the synthesis, characterization and evaluation of three series of novel propellanes with affinity for σ receptors. The x-ray structures of 5 compounds were elucidated and molecular dynamics was used to understand the interactions of 2 compounds within the binding site of sigma-1 receptor.
Overall, there is a significant body of work reported in this manuscript (characterization: NMR, HPLC, elucidation of the stereochemistry/structure).
There are some small typos throughout. (Without line numbers, it is difficult to specify, but abstract, introduction, experimentals – need some spaces and to italicize syn/anti and bold the compound numbers, all have typos.) Don’t want to start sentences with numbers (third sentence introduction).
Title – the title was not clear; with “sigma” being used for so many things in this field, I think it would be helpful to say “σ receptor” or even “sigma-1 receptor”
The structures and activity of the most active compounds 4a, 5, 8, 9a and 9b have previously been reported. It seems the discussion of the effects the stereochemistry (4a diastereomers, right after Table 1) has on the activity was in previous papers. Does this need to be included here again? There is also a similar discussion of the activity of 9a/9b (after Table 2). I am not sure this is necessary if these data have already been reported.
In Table 2, the special character is missing after compound 8 indicating that the affinity data have already been reported.
A figure showing the three main scaffolds would be helpful when the modifications are described (end of introduction). The authors may consider moving Figure 4 earlier and combining the structures of 8 and 9 with the structures of 4a and 5 from Figure 2 or using some form of the graphical abstract.
A more complete description of the effect(s) the different modifications had on activity would be helpful. More SAR
Are these molecules agonist/antagonists for the receptors?
The diastereomeric ratio provided in the tables (e.g. 8-anti-4i and 8-anti-4g, 11-anti-13) does not match what is reported in the experimentals.
It is not clear why there are separate entries for the activity of syn,anti-26 and syn,syn-26 (Table 3) when it seems they were obtained as a mixture of diastereomers. Were they separated?
It would be helpful if the Schemes were standardized. Is there a reason the hydroxyl groups are shown as X/Y in Scheme 2, but the different diastereomers are written out in later schemes?
The references have a weird symbol for the sigma (35, 38, 42, 43).
Author Response
Reviewer #2
In the article, “Propellanes as rigid scaffolds for the stereodefined attachment of σ-pharmacophoric structural elements” Torres-Gomez and colleagues report the synthesis, characterization and evaluation of three series of novel propellanes with affinity for σ receptors. The x-ray structures of 5 compounds were elucidated and molecular dynamics was used to understand the interactions of 2 compounds within the binding site of sigma-1 receptor. Overall, there is a significant body of work reported in this manuscript (characterization: NMR, HPLC, elucidation of the stereochemistry/structure).
There are some small typos throughout. (Without line numbers, it is difficult to specify, but abstract, introduction, experimentals – need some spaces and to italicize syn/anti and bold the compound numbers, all have typos.) Don’t want to start sentences with numbers (third sentence introduction).
Response: The typos have been corrected. During the transfer of the manuscript from word to the template, the format (boldface, italics) was gone partially. We have tried to regenerate the lost format, but obviously, we did it not find everything. Now, we have checked the manuscript once more very carefully. Boldface and italics have been regenerated. 20 years … has been changed into Twenty years..
Title – the title was not clear; with “sigma” being used for so many things in this field, I think it would be helpful to say “σ receptor” or even “sigma-1 receptor”
Response: This article should be part of a special s issue. Therefore, we thought that “s-pharmacophoric structural elements” is sufficient here. Nevertheless, the title has been changed and s affinity has been included.
The structures and activity of the most active compounds 4a, 5, 8, 9a and 9b have previously been reported. It seems the discussion of the effects the stereochemistry (4a diastereomers, right after Table 1) has on the activity was in previous papers. Does this need to be included here again? There is also a similar discussion of the activity of 9a/9b (after Table 2). I am not sure this is necessary if these data have already been reported.
Response: It is correct that the given compounds have been reported previously. However, to show the complete picture, the structures and affinity data had to be repeated here. Without these compounds, the discussion of structure affinity relationships appears incomplete. Moreover, the direct comparison of structures and affinity data of compounds 4a-u and 5 and 8 and 9 with each other and with novel compounds would not be possible without mentioning these compounds. Therefore, we decided to repeat the key data very briefly. The corresponding citations have been included, respectively.
In Table 2, the special character is missing after compound 8 indicating that the affinity data have already been reported.
Response: Compound 8 has already been reported. However, the s affinity of 8 has not been reported. Therefore, the special character § had been omitted for compound 8 in Table 2. The reference for the synthesis of compound 8 48 had been given. Moreover, in the manuscript the aspect that the s affinity of compound 8 has not been reported had been detailed in the original manuscript.
A figure showing the three main scaffolds would be helpful when the modifications are described (end of introduction). The authors may consider moving Figure 4 earlier and combining the structures of 8 and 9 with the structures of 4a and 5 from Figure 2 or using some form of the graphical abstract.
Response: Although it is difficult to show the three compound classes in a clear manner, we generated a new Figure 3, which has been included at the end of the introduction. It should be noted that Figure 3 and the Graphical Abstract are not identical.
The lead compounds 4 and 5 represent the introduction of the first compound class. Therefore, they should appear at the beginning, i.e. in Figure 2. Figure 4 represents the introduction of the second compound class. Therefore, it cannot appear earlier. A combination of compounds 4/5 with compounds 8/9 in one Figure would destroy the organization of the manuscript. The manuscript describes at first compounds of type A, then compounds of type B and then compounds of type C.
A more complete description of the effect(s) the different modifications had on activity would be helpful. More SAR
Response: We tried to keep the manuscript short. Therefore, the SAR discussion referring to compounds with moderate affinity was kept very short. However, the most interesting compounds and their SAR are detailed.
Are these molecules agonist/antagonists for the receptors?
Response: The most reliable assay to test agonistic and antagonistic s1 activity so far is the analgesic activity (antiallodynic activity) in the mouse capsaicin assay. This assay will be performed in collaboration for very potent ligands. We decided not to perform this animal assay for our novel propellane derivatives. Therefore, we do not know, whether we have s1 agonists or s1 antagonists in hand. After further optimization, animal assays will be taken into account.
The diastereomeric ratio provided in the tables (e.g. 8-anti-4i and 8-anti-4g, 11-anti-13) does not match what is reported in the experimentals.
Response: All ratios have been checked. Now, the ratios in the Experimental Part fit to the ratios given in the Tables 1-3.
It is not clear why there are separate entries for the activity of syn,anti-26 and syn,syn-26 (Table 3) when it seems they were obtained as a mixture of diastereomers. Were they separated?
Response: Thank you very much for this comment. syn,anti-26 and syn,syn-26 (were obtained as 1:1-mixture of diastereomers only. Table 3 has been corrected accordingly.
It would be helpful if the Schemes were standardized. Is there a reason the hydroxyl groups are shown as X/Y in Scheme 2, but the different diastereomers are written out in later schemes?
Response: In Scheme 2, the diastereomeric alcohols 11-anti-13 and 11-syn-13 were prepared selectively. Both diastereomeric alcohols were tested. Therefore, we have to define the configuration at 11-positon, i.e. the C-atom bearing the OH moiety. The formula in Scheme 2 should reflect the configuration in 11-position. Moreover, the general formula for compounds 10-12 and 8,13,14 contain the naked compounds (X = Y = H) and the ethylene ketals (X = Y = OCH2CH2O) as well. We decided to show these structures with X and Y to present the chemistry in a condensed manner.
In Schemes 3 and 5, the reduction of the ketones leading to the diastereomeric alcohols is the last step, respectively. The Schemes should show all possible four diastereomeric ureas. Therefore, the configuration at C-11 was detailed.
The references have a weird symbol for the sigma (35, 38, 42, 43).
Response: The weird symbol for sigma in the references has been corrected.
Round 2
Reviewer 1 Report
thanks to the authors for addressing and clarifying all points that have been raised by reviewer 2. The manuscript has been significantly modified and improved. Accordingly, I would recommend the publication of this study in the present form.